# DIFFUSION META-PROMPTS FOR FOUNDATION MODELS

## ABSTRACT

Parameter-efficient fine-tuning techniques, such as prompting, are now popular to adapt foundation models to many tasks. In this paper, we introduce a diffusion-based approach to model the distribution of learned foundation models prompts. Specifically, we propose a **Diffusion Meta-Prompt (DMP)** model that generates prompts conditioned on text or prompt embeddings, and can be used to prompt both vision-language models and diffusion models for image synthesis. DMPs have several advantages: improved generalization of learned prompts; memory and runtime efficiency by eliminating the need to store and search over large repositories of prompts or LoRA weights; multiple applications ranging from open-set classification, to personalization or attribute control of image synthesis; support for operations like subject and concept composition, novel subject generation, negative prompting, and editing without explicit training. For open-set classification, DMP improves base-to-new class generalization, achieving upto **3% average gain** across 11 datasets with gains as high as **7.8%/5.4%** on specific datasets such as Eurosat/UCF101 respectively. DMP also enhances domain, cross-dataset and cross-task generalization with ∼**6-12%** improvement for hierarchical classification task. For image synthesis tasks, DMP improves generalization and prompt compliance by **1.4 points** as measured by CLIP score and reduces storage requirements by **91%** while improving runtime efficiency by **92%** over retrieval methods.

## 1 INTRODUCTION

Foundation models (Rombach et al., 2022; Radford et al., 2021) generalize to diverse tasks due to their large model sizes and large-scale training data. They can also be customized to specific downstream tasks, using parameter efficient methods (Hu et al., 2022; Zhang & Agrawala, 2023), such as prompt learning (Zhou et al., 2022b). Learned prompts are small parameter vectors (or tokens) introduced at the input or intermediate layers of the foundation model to improve its performance on specific tasks, typically using few-shot learning. For visual-language models, prompts are commonly introduced either in the visual (Jia et al., 2022) or textual space (Zhou et al., 2022a) or both (Roy & Etemad, 2024; Hao et al., 2025), to improve performance on tasks like fine-grained classification (Helber et al., 2018), enhance class discrimination (Zhou et al., 2022b;a), support taxonomic classification (Wu et al., 2024), overcome domains shifts (Ge et al., 2022), etc. For generative models, prompts are frequently used to customize the foundation model (Gal et al., 2022; Ruiz et al., 2023) to the synthesis of images containing a specific concept, person, or object. Prompts can also be learned to control the strength of fine-grained concepts or attributes (Sridhar & Vasconcelos, 2024), such as age, emotion, or style, allowing users to enhance or diminish these concepts in the generated image. Finally, learned prompts allow fine control over the editing of real images when combined with diffusion-based inversion techniques, such as the LEDITS++ (Brack et al., 2023) method.

Despite their power as a tool for foundation model adaptation and customization, prompts have the limitations and challenges summarized in the left of Figure 1. In open-set classification, learned prompts typically do not generalize well beyond the base classes used for few-shot learning, and requires separate tuning for each label set. Similarly, prompts for attributes such as age or smiling (see Figure 6) have to be learned separately per entity or concept in image editing/generation tasks. Hence, they are inherently task-specific, requiring task-specific data and losses. This is inefficient, as different applications may benefit from the same prompts or adaptation parameters. One solution, also illustrated in the figure, is the creation of shared repositories, where practitioners drop their prompts

Figure 1: **DMP versus existing approaches**: DMP models provide a natural language interface for the adaptation of foundation models, while eliminating the complexities of large prompt repositories, offering better generalization and composition with a minimal runtime overhead due to the feasibility of using small DMP models and fast sampling.

Figure 2: **Left:** Diffusion Meta-Prompt framework for Text-to-Prompt synthesis. **Right:** Prompt Variation synthesis conditioned on learned Textual Inversion Prompts. See Fig. 9.

for common use. Such efforts are already happening. For example, Concept Sliders (Gandikota et al., 2023b) have been quickly embraced by the diffusion model community, with many content creators producing sliders covering thousands of attributes and uploading them to sites like HuggingFace or Civitai (Civitai, 2023). However, these repositories are somewhat chaotic and require regular maintenance and updates. As they grow to the thousands of entries, it becomes cumbersome to know if a specific attribute is covered and how to find it in the repository. The task is so complex that the design of algorithms to locate, download, and apply the needed prompts to a foundation model has become a topic of research in itself (Luo et al., 2024). However, such prompt retrieval methods (Luo et al., 2024) involve time-consuming search, have limited generalization, require storing large prompt repositories in memory, and sophisticated methods to combine prompts.

In this paper, we propose to solve these problems by unifying prompt generation with a new family of diffusion models, denoted *Diffusion Meta-Prompting* (DMP) models. These are generative models of prompt distributions, capable of synthesizing prompts for downstream foundation models, conditioned on a natural language description of the prompting task. As shown in the rightmost panel of Figure 1, DMPs eliminate the complexities of dealing with prompt repositories, offering better generalization and compositionality with a minimal runtime overhead. For example, given a subject name ("Jennifer Aniston") and attributes ("hair" and "age"), a DMP can generate a set of prompts for a foundation diffusion model like Stable Diffusion XL (SDXL) to produce the corresponding images.

We employ diffusion as the generative model since it offers better generalization than alternatives such as autoregressive methods (Radford et al., 2019). This is demonstrated in two ways. First, we show that *DMPs learn to produce a range of sophisticated prompt operations*, such as concept composition, novel subject generation, image inversion, editing and negative prompting *without explicit training*. We also provide a simple theoretical guarantee on the performance of DMPs. Second, we show that *DMPs can learn multiple tasks,* by showing that a single model can be trained to produce prompts for both subject personalization and slider attributes. This replaces the combinatorial complexities of searching separate prompt repositories for the two (and potentially more) tasks into a single DMP model. DMPs are also shown applicable to a diversity of fundamentally different tasks, ranging from image synthesis to open-set classification. For the latter, we show that learning the distribution of prompts across datasets and label sets generalizes to unseen classes better than the standard approach of individual prompt learning per dataset. The fact that this happens *even though the DMP model is trained on the individual prompts of the baseline approach* shows that there is structure in prompt space, which DMPs learn for improved generalization.

We leverage the DMP framework to develop diffusion models for open-set classification, personalized concept generation with attribute control, and synthesis of images with subject variations, yielding three fully automated prompting models: *DMPClass*, *DMPMulti*, and *DMPVariation*. Overall, the paper makes the following key contributions

- We propose the DMP, a new family of diffusion models trained to synthesize foundation model prompts. Beyond eliminating the need to maintain weight or prompt repositories and simplifying large-scale deployments ($\sim$**91%** efficiency improvements), this approach enhances model generalization and enables flexible prompt manipulations.

- We introduce the DMPClass model, which unifies the generation of prompts for open-set classification with CLIP-style models. This is shown to outperform the baseline learned prompts in various challenging generalization tasks: base-to-new classification (upto **3%**), domain generalization, and across levels of taxonomic classification (upto **12%**).

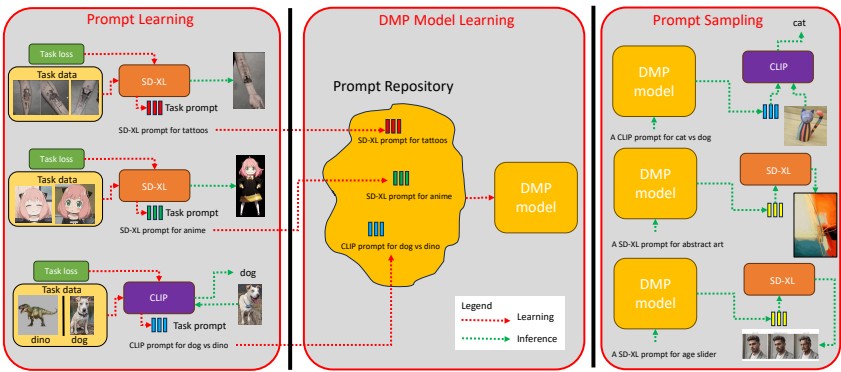

Figure 3: **Pipeline of DMP.** Left: Prompt learning is a technique to specialize a foundation model, such as SD-XL or CLIP, to a specific task, using task data and a task loss to learn a task prompt. This task prompt allows the model to solve the task more effectively. Task examples include personalization of SD-XL to produce images of tattoos or anime characters, or CLIP tuning to the classification of dogs vs dinos. Center: The prompts learned for specific tasks are stored in a prompt repository, which is used to train the DMP model. This does not use any task specific losses nor data. Given enough prompts, the DMP model learns the distribution of prompts for many tasks. Right: After training, the DMP model is used to sample tasks to specialize foundation models, like SD-XL or CLIP to new tasks.

- We introduce the DMPMulti model to unify the generation of personalized subjects and sliders for the control of many attributes in a single model. DMPMulti obtains high identity fidelity and is also shown to support the generation of semantically negative concepts despite not being trained on them. The ability of this model to learn prompts for multiple tasks highlights the potential of DMPs to serve as foundation models for prompts.
- We introduce the DMPVariation model for generating novel variations of a given subject and show that it offers better generalization to unseen subjects (**+1.4** CLIP score), allows composing multiple identities, and supports negative prompting without explicit training.

## 2 RELATED WORK

**Foundation models.** We focus on two classes of foundation models: vision-language representation models (Radford et al., 2021) and text-to-image (T2I) generation models (Rombach et al., 2022). Contrastively trained vision-language models, such as CLIP, are widely used for open-set classification, detection, and segmentation, with prompting techniques enabling adaptation to fine-grained tasks (Zhou et al., 2022b). In T2I models, personalization methods like Textual Inversion (Gal et al., 2022) and DreamBooth (Ruiz et al., 2023) allow generation of images of custom concepts using only a few examples. Building on these, several methods address concept discovery in T2I models (Dalva & Yanardag, 2024; Liu et al., 2023; Gandikota et al., 2023b; Dravid et al., 2024), enabling fine-grained editing or removal of undesirable concepts (Gandikota et al., 2023a; Zhou, 2023). Prompt Sliders (Sridhar & Vasconcelos, 2024) and (Baumann et al., 2024) learn concepts in textual space, either globally or locally. In this work, we propose to unify the prompt learning for foundation models by training a diffusion model to synthesize the prompts conditioned on natural language across tasks.

**Prompt tuning.** Prompt learning, originally developed for language models (Shin et al., 2020; Jiang et al., 2020), applies a fixed or learnable function to input tokens to provide task-specific instructions. In computer vision, prompts can be textual (Zhou et al., 2022a), visual (Jia et al., 2022), or multi-modal (Khattak et al., 2023; Yang et al., 2024; Li et al., 2025b;a). Textual prompt learning, pioneered by CoOp (Zhou et al., 2022b) and CoCoOp (Zhou et al., 2022a), fine-tunes a CLIP model (Radford et al., 2021) for few-shot transfer by optimizing continuous prompt vectors in the language branch. Visual prompt tuning (Jia et al., 2022) introduces task-specific learnable prompts in the visual encoder while keeping the backbone fixed. Multi-modal prompt learning (Roy & Etemad, 2024; Hao et al., 2025) optimizes prompts in both vision and language encoders to improve cross-modal alignment. Bayesian prompt learning (Derakhshani et al., 2023) formulated prompt learning as a variational inference problem and demonstrated its ability to generalize to unseen classes at the expense of base class accuracy. Table 9 in Appendix shows how DMP differs from prior prompt-learning approaches across key axes. Unlike prior approaches, which refine prompts for a single task or single example,

DMP is the only method that enables cross-task, data-free, multi-task prompt generation through a learned prompt distribution, highlighting its novelty and broader applicability.

**Meta learning.** Meta-learning, or learning to learn, enables efficient adaptation to new tasks by leveraging past experience (Ha et al., 2017; Hospedales et al., 2021). It has been applied to learn loss functions (Bechtle et al., 2021), task-specific initialization (Gong et al., 2024), generate weights (Peebles et al., 2022; Zhmoginov et al., 2022; Nava et al., 2022; Zhang et al., 2024), and few-shot learning (Snell et al., 2017). Inspired by this, we propose Diffusion Meta-Prompts (DMP) to *synthesize* foundation model prompts conditioned on natural language for multiple tasks and concepts, to address the generalization limitations of existing prompt learning approaches. Recently, (Du et al., 2024) proposed a diffusion model for *refining* CLIP prompts for classification. It is trained on a *specific example and classification task* to improve CLIP prompts for that task. In contrast, DMP is a generalist, trained to sample prompts across tasks and downstream functionalities. The two approaches are complementary, as task-specific refinement from (Du et al., 2024) could be applied to DMP-generated prompts. Unlike this prior task-specific method, we further demonstrate that DMP generalizes across classes, models, and tasks, by introducing three new DMP variants.

## 3 DIFFUSION META-PROMPTING

We introduce the *Diffusion-based Meta Prompting* (DMP) framework for synthesizing prompts conditioned on task descriptions. Let $\mathcal{C}$ denote a distribution over tasks or concepts $c \sim \mathcal{C}$. For each concept $c$, we assume a textual description $y(c)$, and a repository $\mathcal{R}$ of exemplar prompts $S(c) \in \mathbb{R}^d$ obtained from existing prompt-learning techniques. Figure 2 (left) illustrates the general implementation of the DMP framework. The objective is to learn a conditional generator $p_\theta(S \mid y)$, parameterized by a diffusion model for a downstream task with loss $\mathcal{L}_{\text{task}}(S; c)$, such that prompts $S \in \mathcal{P}$ (prompt space) drawn from it achieve low downstream risk. Figure 3 illustrates the the novel formulation and the pipeline of DMP compared to classical prompt-learning approaches. Unlike prompt learning methods, DMP learns a task-agnostic prompt distribution from a prompt repository and can sample prompts for any task without additional supervision.

**Training.** DMPs are diffusion models (Sohl-Dickstein et al., 2015; Ho et al., 2020a) that synthesize prompts by iteratively denoising a noise seed. DMP training is based on a pair of forward and backward Markov chains. Given a concept $c$, an associated prompt $S(c)$ is retrieved from $\mathcal{R}$ and noised according to a forward process that progressively adds Gaussian noise to $x_0 = S(c)$, according to

$$x_t = \sqrt{\bar{\alpha}_t} x_0 + \sqrt{1 - \bar{\alpha}_t}\, \epsilon_t, \tag{1}$$

where $t$ is a timestep, $\bar{\alpha}_t := \prod_{s=1}^{t}(1 - \beta_s)$, $\{\beta_t\}_{t=1}^{T}$ is a variance schedule, $\epsilon_t \sim N(0, \mathbf{I})$ and $N$ is a Gaussian distribution. In the reverse process, a neural network $\epsilon_\theta$ recurrently denoises $x_t$ to recover $x_0$. This network is trained to predict noise $\epsilon_t$, by minimizing the risk

$$\mathcal{L}_{\text{denoise}}(\theta) = \mathbb{E}_{t, x_0, \epsilon}\left[ \|\epsilon_t - \epsilon_\theta(x_t, t)\|^2 \right]. \tag{2}$$

The process is repeated by sampling over concepts $c$ and associated prompts $S(c)$. The network $\epsilon_\theta(x_t, t)$ is a U-Net (Ronneberger et al., 2015) with self and cross-attention layers (Vaswani et al., 2017). The latter are conditioned by a text prompt $y(c)$ that specifies the concept $c$, e.g. "a personalization prompt for Jennifer Anniston", in the example of Figure 1. A text embedding $\tau_\theta$ maps $y(c)$ into a conditioning vector $\tau_\theta(y)$, where we omit the argument $c$ for brevity. The denoising network is then represented as $\epsilon_\theta(x_t, \tau_\theta(y), t)$. In our implementation, the DMP model is trained with classifier-free guidance, where an empty text or null prompt is used 20% of the time, to allow for better guidance during sampling. After training, a user simply specifies the text prompt $y(c)$. The diffusion model samples a prompt $S(c)$ with $x_{t-1} = \frac{1}{\sqrt{\alpha_t}}\left( x_t - \frac{1 - \alpha_t}{\sqrt{1 - \bar{\alpha}_t}} \epsilon_\theta(x_t, \tau_\theta(y), t) \right) + \sigma_t \epsilon$, where $x_T \sim N(0, \mathbf{I})$ is a noise seed, and $S(c) = x_0$. From a meta-learning perspective, $\theta$ are meta-parameters amortizing the construction of prompts across tasks. Unlike per-task optimization, DMP enables *one-shot prompt synthesis* by sampling from the learned generator.

**Architecture.** To implement the DMP model, we develop a 1D variant of the popular Stable Diffusion model, where 2D operations are replaced by their 1D counterpart (e.g., 1D-conv). Since the size of the prompt embeddings ($d$) is relatively small, we have found that, for most applications, the model can operate directly in the space $\mathcal{P}$ of prompts $S(c) \in \mathbb{R}^d$. However, for applications involving

unusually long prompts($d \geq 2048$), we have also trained a variational autoencoder that maps prompts from $\mathcal{P}$ into a space of lower dimensionality, for faster training and convergence. This is a 1D variant of stable diffusion autoencoder with less than 1M parameters and a latent dimension of 128. See appendix section A.10 for more details.

**Concept Composition.** By framing diffusion models as Energy Based Models, (Liu et al., 2022b) showed that it is possible to compose multiple concepts by conjunction or negation. Given a diffusion model $\epsilon_\theta(x_t, t)$, $n$ concepts are combined by implementing the denoising chain with

$$\hat{\epsilon}(x_t, t) = \epsilon_\theta(x_t, t) + \sum_{i=1}^{n} \eta_i \left( \epsilon_\theta(x_t, \tau_\theta(y(c_i)), t) - \epsilon_\theta(x_t, t) \right), \tag{3}$$

where $\eta_i$ is a hyperparameter corresponding to a temperature scaling of concept $c_i$. If the conditioning is just empty text, this reduces to classifier-free guidance. The standard implementation of concept composition is to run the *downstream* diffusion model $n$ times (once per concept) and average noise predictions with (3). This is significantly more complex than performing the concept composition of (3) in the (much less complex) DMP, which enables the sampling of *single* prompts $x_t$ for the downstream model that combine *all* concepts $c_1, \ldots, c_n$. We show below that DMPs trained on the individual concepts can sample combined prompts (Table 8, Figure 23).

**Concept Negation.** Negative prompting, is commonly used in image diffusion models to direct image generation away from undesired semantic concepts, thus improving quality. Given a diffusion model $\epsilon_\theta(x_t, t)$, the denoising chain is implemented with $\hat{\epsilon}(x_t, t) = \epsilon_\theta(x_t, t) + \eta \left( \epsilon_\theta(x_t, \tau_\theta(y(p)), t) - \epsilon_\theta(x_t, \tau_\theta(y(n)), t) \right)$, where $\eta$ is the hyperparameter that controls the strength of the negation, $p$ is a positive concept and $n$ a negative one. We show that, without specific training, DMPs can sample *single* prompts $x_t$ for the downstream model that oppose concept $p$ to concept $n$ or even purely negative prompts, by using an empty positive prompt $p$ (see Figures 7,22).

**Theoretical Guarantee.** We provide a theoretical guarantee for the performance of DMP. The formal statement is below, with assumptions and proof discussed in Appendix A.1.

**Proposition 1** (DMP performance guarantee)**.** *Assume the denoising loss satisfies $\mathcal{L}_{\mathrm{denoise}}(\theta) \leq \varepsilon$ and the repository contains $n$ i.i.d. prompts for the condition $y$. Then with probability at least $1 - \delta$ over the repository sample,*

$$\mathbb{E}_{S \sim p_\theta(\cdot|y)} \left[ \mathcal{L}_{\mathrm{task}}(S) \right] \leq \mathbb{E}_{S \sim \widehat{p}_{\mathrm{data}}(\cdot|y)} \left[ \mathcal{L}_{\mathrm{task}}(S) \right] \quad + L_{\max} \sqrt{2C\,\varepsilon} + L_{\max} \sqrt{\frac{\log(2/\delta)}{2n}} + o(1) \tag{4}$$

## 3.1 DMP FOR CLASSIFICATION

We train DMP models (DMPClass) for several state-of-the-art prompt tuning methods such as CoOp (Zhou et al., 2022b), CoCoOp (Zhou et al., 2022a), MapLe (khattak et al., 2023),CoPrompt (Roy & Etemad, 2024) and TAC (Hao et al., 2025). These include both unimodal and multimodal methods.

**Prompt Repository.** To create a repository of classification prompts, we train the CLIP ViT-B/16 model following the original CoOp/CoCoOp/MapLe/CoPrompt/TAC methods. Given a classification problem, a prompt set is trained for $N$ epochs where $N$ follows the original settings mentioned in the these papers, using a context vector of size $K$. We consider ImageNet and the standard 10 datasets used in the prompt learning literature (Zhou et al., 2022b;a). We use $40$ different initializations per dataset and save the corresponding prompt embeddings to obtain the training tensor $\mathcal{T} \in \mathbb{R}^{11 \times 40 \times K \times d}$. See Appendix A.10 for more details.

**Training.** Due to the large dimensionality of the prompt embeddings ($K \times d = 2048$), the DMPClass models are trained with an autoencoder. To obtain the text inputs, we concatenate the respective dataset classnames into a text string, whose text embedding is used to condition the DMPClass model. This setup allows us to flexibly mix and match class names across datasets to enable straightforward compositionality (Table 8) without requiring handcrafted or semantically enriched descriptions. Since the CLIP text encoder has a limit of 77 tokens per input (around 50 words), this string can be too long for datasets with many classes (e.g. the 1,000 Imagenet classes). To overcome this, we consider each class name independently, obtain the CLIP embedding for each resulting in $C$ embeddings for $C$ classes, which are then used to prompt DMPClass models.

## 3.2 DMP FOR SUBJECT VARIATION

DMPVariation synthesizes prompts learned via textual inversion (TI) (Gal et al., 2022) to generate variations of a subject when conditioned on its TI prompt.

**Prompt Repository.** TI uses the diffusion loss to learn a text prompt $S^*$ for the downstream diffusion model $\epsilon_\theta^d(x_t, \tau_\theta(y), t)$, from a few example images (typically 3-5 per concept), while keeping the model weights unchanged. This allows the model to synthesize images containing the concept. The prompt is learned through the optimization

$$S^* = \arg\min_S \mathbb{E}_{x \sim \mathcal{E}(x), y, \epsilon \sim \mathcal{N}(0,1), t} \left\| \epsilon_t - \epsilon_\theta^d(x_t, \tau_\theta(y, S), t) \right\|_2^2 \tag{5}$$

where $x_t$ is an image, and $y$ is any additional prompt text, such as "a photo of a". Both $\tau_\theta$ and $\epsilon_\theta$ are fixed during the optimization, which is performed by backpropagation.

To produce a prompt repository $\mathcal{R}$, we split the CelebA dataset into unique identities using the groundtruth labels, and use 3,000 of these as concepts $c$, for which we train personalized prompts $S(c)$ with (5) using 1,000 steps of gradient descent. To obtain variations, we note that earlier optimization steps do not fully encode face attributes, as compared to steps later in the optimization. We save the prompts $S(c)$ from 40 gradient steps (between $200 - 400$ in intervals of 5), per identity. This produces a repository $\mathcal{R}$ of 40 prompt variations per identity to obtain a training tensor $\mathcal{T} \in \mathbb{R}^{3000 \times 40 \times d}$.

**Training.** The DMPVariation model was trained directly in prompt space $\mathcal{P}$, i.e. without autoencoder. Each personalized prompt produced by TI is used as conditional input to DMPVariation, by concatenating it with the noise vector, as illustrated in Figure 2. During training, the model learns to denoise the 40 different variations of the conditional prompt input. At inference, prompts that induce variations of novel subjects can be sampled by simply conditioning on a new subject prompt. We show that *despite only training the model on prompts for faces from the CelebA dataset, the DMP can synthesize variation prompts for very different concepts*, e.g. cats/dogs or statues or different styles (downloaded from the internet repositories). See Figure 17 and Figure 18 in Appendix A.5.

## 3.3 MULTI-TASK DMP

We propose *DMPMulti*, a multi-task DMP model trained to synthesize prompts for personalized subjects (Gal et al., 2022) and prompt sliders (Sridhar & Vasconcelos, 2024). Both tasks are unified within a single framework through the shared CLIP-ViT L/16 text encoder.

**Prompt Slider Repository.** Prompt Sliders is a technique to learn text prompts in the CLIP text embedding space that allow control of particular attributes of a designated concept. Given a target concept $c_t$, a prompt slider $S^*$ is learned to encourage the distribution of images of $c_t$ to exhibit more positive attributes $c^+$ and fewer negative attributes $c^-$. This is implemented by replacing $\epsilon_t$ with $\epsilon_t(\alpha) = \epsilon_\theta^d(x_t, \tau_\theta(y(c_t)), t) + \alpha\eta \sum_{p \in P}(\epsilon_\theta^d(x_t, \tau_\theta(y(c^+, p)), t) - \epsilon_\theta^d(x_t, \tau_\theta(y(c^-, p)), t))$ and $S$ by $\alpha S$ in (5), where $\eta$ is a guidance scale, $\alpha$ a scaling parameter, and $P$ a set of concepts that the attribute manipulation should preserve (for example, race or gender). The positive $c^+$, and negative $c^-$ attributes are sampled from a template predefined for concept $c_t$. To create a slider repository $\mathcal{R}_S$, we trained prompt sliders for 20 different concepts using the SD-XL model and $3,000$ backpropagation steps. For each concept $c$, we save 40 prompts $S(c)$ (from the last 200 steps in intervals of 5) to obtain a training tensor $\mathcal{T} \in \mathbb{R}^{20 \times 40 \times d}$.

**Prompt Identity Repository.** We use a random subset of 20 identities from the 3000 identities in the DMPVariation prompt repository to create a prompt identity repository $\mathcal{R}_I$ with each identity containing 40 prompts obtained from the last 200 steps in intervals of 5 to obtain a training tensor $\mathcal{T} \in \mathbb{R}^{20 \times 40 \times d}$.

**Training.** The DMPMulti model was trained directly in prompt space $\mathcal{P}$, i.e. without autoencoder, using the full set of prompts from both repositories, i.e. $\mathcal{R} = \mathcal{R}_S \cup \mathcal{R}_I$. For each slider concept $c$ (e.g., age, smiling etc.) we use the concept name as the text condition for DMPMulti. For identities, we use "identity-$c$" as the text condition where $c = 1, \ldots, 20$ is associated with each identity $c$ in $\mathcal{R}$.

**Inference.** At inference, DMPMulti can be conditioned with the slider concept name, to generate prompt sliders, or with "identity-$c$" to generate identity prompts. Novel identities can be sampled by specifying a new identity "identity-$c$" with $c > 20$. Slider and identity prompts can then be fed to the downstream model in isolation or together.

Table 1: Memory and complexity requirements.

| Method | Mem (GB) ↓ | Time (s) ↓ |
|---|---|---|
| Stylus | 1.30 | 12.1 |
| DMP | **0.12** | **1** |
| Gain | **(91%↓)** | **(92%↓)** |

Table 2: Performance of different slider methods.

| Method | CLIP-s ↑ | LPIPS↓ |
|---|---|---|
| Concept Slider | 28.90 | 0.086 |
| Prompt Slider | **30.00** | 0.219 |
| DMPMulti | 29.86 | **0.126** |

Table 3: Domain generalization accuracy (%) for prompts learned on ImageNet.

| Method | Source | | Target | | | |
|---|---|---|---|---|---|---|
| | ImageNet | -V2 | -S | -A | -R | Average |
| CoOp | 68.5 | 61.20 | 45.20 | 48.20 | 74.10 | 59.44 |
| DMPCoOp | **68.7** | **62.10** | **46.50** | **50.60** | **75.10** | **60.60** |

Table 4: **Base-to-novel generalization:** Accuracy of CLIP prompted classifier for All, Base, and New classes. See Appendix Table 11 for the full results. The results reported are the average over three seed runs.

| | (a) Avg (11 datasets) | | | | (b) DTD | | | | (c) EuroSAT | | | | (d) UCF101 | | | |
|---|---|---|---|---|---|---|---|---|---|---|---|---|---|---|---|---|
| Method | All | Base | New | HM | All | Base | New | HM | All | Base | New | HM | All | Base | New | HM |
| CoOp (Zhou et al., 2022b) | 68.8 | **82.3** | 70.4 | 75.9 | 51.9 | **80.8** | 51.8 | 63.1 | 63.2 | **90.8** | 72.9 | 80.9 | 67.6 | **83.4** | 65.3 | 73.2 |
| **DMPCoOp** | **70.1** | 80.3 | **73.4** | **76.5** | **53.7** | 75.1 | **55.8** | **64.0** | **65.9** | 85.6 | **80.7** | **83.1** | **71.7** | 79.6 | **70.7** | **74.9** |
| Δ | +1.3 | -2.0 | +3.0 | +0.6 | +1.8 | -5.7 | +4.0 | +0.9 | +2.7 | -5.2 | +7.8 | +2.2 | +4.1 | -3.8 | +5.4 | +1.7 |
| CoCoOp (Zhou et al., 2022a) | 70.1 | **80.7** | 72.5 | 76.0 | 52.3 | **77.5** | 54.8 | 64.2 | **66.0** | **87.9** | 65.6 | 74.9 | **72.7** | **82.2** | 72.1 | 76.8 |
| **DMPCoCoOp** | **70.2** | 79.5 | **74.0** | **76.4** | **52.9** | 75.8 | **58.5** | **66.0** | 65.8 | 87.2 | **66.6** | **75.4** | 72.5 | 80.6 | **76.0** | **78.2** |
| Δ | +0.1 | -1.2 | +1.5 | +0.4 | +0.6 | -1.7 | +3.7 | +1.8 | -0.2 | -0.7 | +1.0 | +0.5 | -0.2 | -1.6 | +3.9 | +1.4 |
| CoPrompt (Roy & Etemad, 2024) | 72.3 | **83.1** | 74.6 | 78.3 | 56.3 | **82.1** | 57.6 | 67.6 | 67.1 | **94.3** | 66.8 | 78.0 | **76.5** | **86.8** | 78.7 | **82.5** |
| **DMPCoPrompt** | **72.9** | 82.5 | **75.4** | **78.5** | **57.4** | 80.5 | **62.1** | **70.1** | **70.7** | 91.1 | **69.6** | **78.6** | **76.5** | 86.4 | **78.8** | 82.4 |
| Δ | +0.6 | -0.6 | +0.8 | +0.2 | +1.1 | -1.6 | +4.5 | +2.5 | +3.6 | -3.2 | +2.8 | +0.6 | 0.0 | -0.4 | +0.1 | -0.1 |
| TAC (Hao et al., 2025) | 74.6 | **85.2** | 77.1 | 80.8 | 59.1 | **83.6** | 62.7 | 71.6 | 76.4 | **94.3** | 80.2 | 86.6 | 78.2 | **87.2** | 81.1 | 84.1 |
| **DMPTAC** | **75.0** | 85.1 | **77.5** | **80.9** | **59.3** | 83.3 | **63.6** | **72.1** | **76.7** | 93.9 | **81.5** | **87.2** | 78.2 | 87.2 | **81.4** | **84.2** |
| Δ | +0.4 | -0.1 | +0.4 | +0.1 | +0.2 | -0.3 | +0.9 | +0.5 | +0.3 | -0.4 | +1.3 | +0.6 | 0.0 | 0.0 | +0.3 | +0.1 |

## 4 EXPERIMENTS

In this section, we discuss experimental results obtained with the three DMPs. For classification prompts, we evaluate accuracy across 11 datasets and OOD-generalization on ImageNet dataset variants: ImageNetV2 (Recht et al., 2019), ImageNet-Sketch (Wang et al., 2019), ImageNet-Rendition (Hendrycks et al., 2020) and ImageNet-Adversarial (Hendrycks et al., 2021). For slider concept prompts, evaluations are based on CLIP/LPIPS-score and for variation prompts, we compute Face ID similarity to groundtruth using a VGGFace2 (Cao et al., 2018) Inception ResNet model.

**Implementation Details.** We conduct all experiments on 24GB (NVIDIA-A10 or 3090-RTX) GPUs using pytorch. DMP models are trained with the standard hyperparameters of (Rombach et al., 2022), a learning rate of $1e^{-6}$, and batch size of 320, for 2,000 epochs. This takes about a day to train on 4 GPUs for 3,000 identities. For classification prompts, the autoencoder is trained for 10,000 epochs, which takes about 10 hours. DMP uses 50 DDIM timesteps to sample one prompt which takes about 1 sec and it can be sped up with faster sampling methods (Liu et al., 2022a). The original CoOp paper reports the full results only for a context length of 16. We use a context length of 4 ($K = 4$) as suggested in CoCoOp. For fair comparisons, we use the codebase of respective methods and run all our experiments under this setup. More details are in the appendix A.10.

**Storage and Runtime Efficiency.** One of the benefits of the DMP framework is its high efficiency. DMP eliminates the need to manage a repository of prompts and associated *metadata*, such as text descriptions. This contrasts with methods like Stylus (Luo et al., 2024), that automatically search, retrieve, and compose LoRAs from a repository. Table 1 compares the storage and processing requirements of DMP and Stylus, when used for the tasks that we consider in this work. DMP is significantly more efficient, reducing storage needs by **91%** and improving inference speed by **92%**. Further, Table 11 shows that DMP achieves better generalization than Stylus for classification, both on new classes (**73.4** vs. 72.8) and across all classes (**70.1** vs. 67.5) over 11 datasets.

### 4.1 DMP FOR CLASSIFICATION

To evaluate DMP-based classification, we conducted Base2New generalization experiments using five different state-of-the-art prompting methods, across 11 diverse datasets. To assess the ability to generalize to unseen classes, only the first half of the classes were used to learn prompts and evaluated for those classes (Base), the remaining classes (New) and all classes (All). Table 4 summarizes the performance of standard learned prompts and those synthesized by DMP. While DMP prompts have slightly lower performance for the base classes, an expected outcome since they are inherently bounded by the accuracy of the learned prompts used to train the DMP - they generalize better, with an average accuracy gain ranging from **+0.5-3.0%** for unseen classes across all prompting methods. In particular, for new classes, DMP outperforms the learned prompts on **11/11** datasets for CoOp,

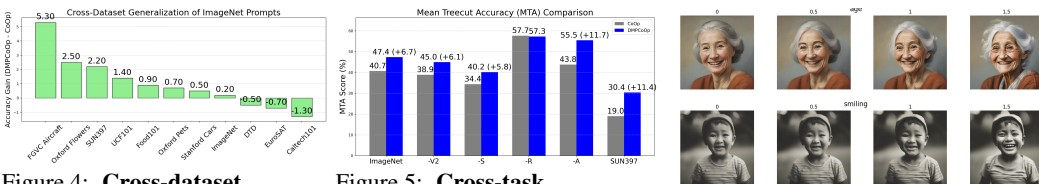

Figure 4: **Cross-dataset generalization across 10 datasets**: Gain of DMPCoOp over CoOp for ImageNet prompts.

Figure 5: **Cross-task generalization**: Comparison of DMPCoOp against CoOp prompts for hierarchical classification.

Figure 6: **DMPMulti Prompt Sliders**: SD-XL images using DMPMulti prompt sliders for 'age' and 'smiling'.

Table 5: Comparison of DMPMulti with baseline Textual Inversion (TI) prompts for identity synthesis.

| Method (SD-RV) | Face-ID↑ | DINO ↓ | CLIP-I ↓ | CLIP-T ↑ |
|---|---|---|---|---|
| Textual Inversion | 0.428 | 0.627 | 0.696 | 0.244 |
| Stylus (Top-1) | **0.434** | 0.645 | 0.706 | 0.246 |
| DMPMulti | **0.434** | **0.558** | 0.653 | 0.245 |
| DMPMulti(20) | 0.429 | 0.595 | **0.599** | **0.285** |

Table 6: Comparison of DMP-Variation with Textual Inversion (TI) for variation synthesis.

| Method (SD-RV) | Face-ID ↓ | CLIP-T ↑ | HPSv2 |
|---|---|---|---|
| TI | 0.428 | 27.0 | 26.6 |
| DMPVariation | **0.299** | **28.4** | **27.3** |
| Δ | -0.13 | +1.4 | +0.7 |

Table 7: Accuracy comparison on combined EuroSAT and Flowers.

| Method | Base | New |
|---|---|---|
| CoOp (ZS) | 50.2 | 59.4 |
| BPL (ZS) | 61.3 | 73.4 |
| **DMPCoOp (ZS)** | **74.5** | **77.8** |
| BPL (Trained) | 76.1 | 75.1 |

CoCoOp, CoPrompt, and TAC methods. The table also shows that on certain datasets, the gain can be as high as **+7.8/5.4%** on Eurosat/UCF101 respectively. Note that for methods other than CoOp, DMP only synthesizes the text prompts attached to the input of CLIP and do not generate the weight or projection matrices of the deeper prompts used by those methods. The learned weights from the original prompting method is used together with the synthesized prompts from DMP. Despite only modifying the shallow prompts, DMP obtains **+0.4-1.5%** average accuracy improvement for these methods. These findings suggest that the original learned prompts may be somewhat overfitted to the base classes, whereas DMP prompts induce a classifier of stronger generalization. Over all classes (both seen and unseen) and datasets, DMP has an accuracy gain of upto **+1.3%**. It can be concluded that *meta-prompting with a diffusion model outperforms the existing prompting techniques that are used to train it*.

We conduct additional studies using DMPCoOp to evaluate generalization of DMP across tasks. First, we consider domain generalization. Table 3 demonstrates the robustness of a DMP model trained on ImageNet, by evaluating the performance of its prompts on four out-of-distribution (OOD) ImageNet datasets. The DMPCoOp model consistently improves over CoOp across all four ImageNet variants, with **1.16**% average gain in classification accuracy. Figure 4 summarizes the gains of cross-dataset generalization. Prompts learned on ImageNet are used for classification of the 10 other datasets considered in Table 4. DMPCoOp outperforms the baseline CoOp prompts in **7/10** datasets, achieving a significant overall average gain of **1%**. The gains are quite large for datasets with less common classes, such as "FGVC aircraft" **(+5.3%)**, or involving very fine-grained classes, such as "Oxford Flowers" **(+2.5%)**, which are not likely to appear in ImageNet. Conversely, CoOp does slightly better on datasets like Caltech101, whose classes have large overlap with ImageNet.

We next consider cross-task generalization in the more challenging task of taxonomic classification (Wu et al., 2024), which tests the ability of the classifier to classify images with respect to different class subsets in a class hierarchy, using metrics of Mean Treecut Accuracy (MTA) over 25 treecuts and Hierarchical Consistency Accuracy (HCA). Figure 5 summarizes the MTA gains of DMPCoOp over CoOp on Imagenet variants and the SUN dataset. DMPCoOp prompts obtains a significant **5.8-11.7%** average improvement over CoOp, even though no prompts are ever trained for hierarchical classification. These results align with the findings from the class, domain, and dataset generalization experiments above, confirming that the DMP model learns to generate prompts that are more robust than the original learned prompts on which it is trained. See Table 12 for the full results.

To evaluate the generalization ability of DMP, we introduce a composite classification task that require generalization across datasets and explain the procedure in detail as follows. As mentioned above, we used prompt learning to train prompts over the 11 datasets $\mathcal{D}_i, i \in \{1, \ldots, 11\}$ considered in the paper. In all cases, we used the Base/New decomposition, where some classes (New) were left out of training to test generalization. We created a prompt repository with the prompts $\mathcal{P}_i, i \in \{1, \ldots, 11\}$ learned from the 11 datasets and used it to train the DMP model. We then created all 55 datasets $\mathcal{T}_i, i \in \{1, \ldots, 55\}$ containing pairs from these 11 datasets, to test generalization performance. For each dataset $\mathcal{T}_i$, we performed classification with (1) the prompt sampled from DMP, (2) the two prompts $\mathcal{P}_i$ associated with the two datasets $\mathcal{D}_i$ in the pair. For the baseline method (2), we then

Table 8: Comparison of CoOp vs DMPCoOp across All, Base, and New splits for 55 dataset combination pairs.

| Split | Avg CoOp (%) | Avg DMPCoOp (%) | Avg Gain (%) | Top-3 Pairs: CoOp → DMPCoOp, Gain |
|---|---|---|---|---|
| All | 57.9 | 62.2 | 4.3 | EuroSAT&OxfordFlowers  35.5→61.2 (g=25.7)
OxfordFlowers&OxfordPets 63.4→80.9 g=17.5)
EuroSAT&OxfordPets:  51.2→65.7 (g=14.5) |
| Base | 63.4 | 68.1 | 4.7 | ImageNet&OxfordFlowers 32.6→70.0 (g=37.4)
FGVCAircraft&ImageNet  36.1→65.9 (g=29.8)
EuroSAT&OxfordPets  58.0→76.1 (g=18.1) |
| New | 66.4 | 69.7 | 3.3 | OxfordPets&SUN397  47.6→69.7 (g=22.1)
EuroSAT&OxfordFlowers 59.4→77.8 (g=18.4)
EuroSAT&OxfordPets  67.6→81.1 (g=13.5) |

measured the performance of the two prompts $\mathcal{P}_i$ on the test classes of $\mathcal{T}_i$ and chose the best for each dataset. We also tested the average of the two prompts $\mathcal{P}_i$ as the alternative to this but it did not perform better. Note that this is a best-case scenario for prompt learning, which would not be feasible in practice, where test labels are not available. This was then compared to the performance of the DMP prompt, with the results shown in Table 8.

The table shows the average accuracy of baseline CoOp prompts and DMPCoOp prompts across all 55 dataset combination pairs. The last column reports the top-3 highest individual gains observed in specific combinations mentioned which are significantly higher than the CoOp prompts. Table 8 shows that DMPCoOp outperforms the CoOp baseline by **+4.7%** for Base,**+3.3%** for New classes and **+4.3%** for All classes with individual gains for specific pairs as high as **+37.4%/+22.1%/+25.7%** for Base/New/All classes respectively. These results demonstrate that DMP generalizes across tasks and datasets in ways that the base prompt learner cannot, despite being trained only on its prompt embeddings. Additional results for TAC are included in Appendix A.3. We also compare DMPCoOp with Bayesian Prompt Learning (BPL), which requires explicit training for each dataset pair. As shown in Table 7, DMP outperforms the performance of a fully trained BPL model without any task-specific training by **+2.7%** and exceeds BPL's zero-shot baseline by **+13.2/+4.4%** for base/new classes respectively on EuroSAT and Flowers dataset pair. We also note that BPL is computationally expensive, requiring approximately 10 hours on a 40GB GPU even for a single dataset pair, which limits its scalability compared to DMP.

Additional ablation studies on comparison with VAE, autoregressive methods (Table 11), impact of text conditions (Table 18) and noisy prompts (Table 21) are discussed in Appendix A.6.1.

## 4.2 MULTITASK DMP

**Sliders.** Table 2 compares the performance of three slider approaches: the concept sliders of (Gandikota et al., 2023b), the baseline prompt sliders of (Sridhar & Vasconcelos, 2024), and those produced by DMPMulti. CLIP-scores are computed between the images synthesized by the downstream SD-XL model, prompted with sliders for 20 attributes, and the attribute names. Both CLIP and LPIPS scores are evaluated on a set of 100 custom prompts (see appendix A.10.2 for details) with 5 prompts per concept. Concept Sliders often fail to induce noticeable changes in the generated images (Fig. 16), resulting in strong LPIPS scores but weak CLIP scores. In contrast, DMPMulti achieves a CLIP score similar to the Prompt Slider upper bound, and is more that 1 point better than Concept Sliders. It also corrects a tendency of Prompt Sliders to produce exaggerated attributes, which result in poor LPIPS scores. Overall, the sliders synthesized by DMPMulti achieve the best balance, among the three approaches, between image quality and attribute manipulation. This is also illustrated in Fig. 16. Figure 6 shows qualitative results of images synthesized by manipulating the strength of prompts (for attributes 'smiling,' and 'age,') generated by the DMPMulti model. See appendix Figures 13, 14, and 15 for more qualitative results.

**Semantic Negative Sliders.** Prompt Sliders are restricted to positive semantic directions due to their training setup. In contrast, DMPMulti learns a unified text-conditioned prompt space that also supports negative directions by setting the positive prompt to empty text and the negative prompt to the target concept in (3). This enables attenuation of attributes *even though DMPMulti is trained only with positive concepts*. Figure 7 demonstrates this capability; using LEDITS++ (Brack et al., 2023) inversion, a real image is initialized in SDXL and edited with DMPMulti's negative prompts for "age" and "smiling". Compared to InstructPix2Pix and LEDITS++, DMPMulti achieves superior edits, effectively capturing negative semantics such as the absence of "age" or "smiling". Together, Figures 6 and 7 highlight DMPMulti's ability to produce semantically consistent prompts for both positive and negative concept directions.

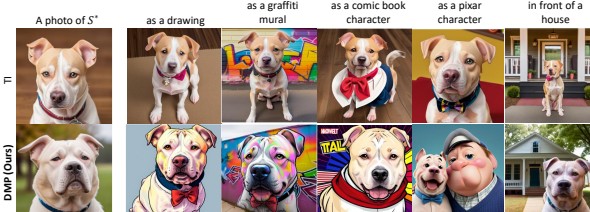

Figure 7: **Editing real images** to reduce age (top) and remove smiling (bottom) attribute from real images using LEDITS++ image inversion and prompt sliders generated with a negative prompt using DMPMulti and compared with InstructPix2Pix.

Figure 8: **Generalization of DMPVariation model**. Images generated using prompts synthesized by TI (top) and DMP-Variation (bottom) for various downstream model text prompts (shown on top). DMPVariation prompts are more robust than TI prompts, which impair the ability of the downstream model to generalize. See Fig. 20 for additional results.

**Identities.** Table 5 compares Identity prompts from DMPMulti with Stylus and Textual Inversion across face recognition accuracy, image-to-image similarity, and prompt fidelity. DMPMulti achieves higher identity scores, maintains prompt compliance comparable to TI/Stylus, and shows lower similarity to training images, indicating stronger generalization. In contrast, TI tends to overfit, consistent with prior findings (Yuan et al., 2023). The last row of Table 5 shows an ablation study of using fewer samples (20 vs 40) per identity for training DMP. Notably, DMPMulti (20) achieves similar identity fidelity and even higher prompt compliance (↑17%) than the 40-sample variant.

### 4.3 DMPVARIATION

Figure 8 presents qualitative results comparing DMPVariation and Textual Inversion (TI) in generating personalized images across diverse contexts. In this example, the downstream model personalized with prompts produced by TI or DMPVariation, is asked to generate a new image of the subject in a different context, specified as a text prompt atop each image. The variation prompts produced by DMPVariation demonstrate greater robustness and generalization, effectively adapting to different contexts while preserving subject identity. This is unlike TI prompting, which tends to overfit to the subject, leading to poor generalization across contexts. In result, it fails to generate suitable images of the dog for all contexts other than *"in front of a house"*. The figure shows that DMPVariation produces successful variation prompts for general objects as diverse as dogs *despite being trained only on CelebA-face identities* (See Fig. 18 for other objects such as bird and statue). Fig. 20 in Appendix shows additional results where TI completely fails as opposed to DMPVariation.

To evaluate generalization, we compare DMPVariation prompts with learned TI prompts on CelebA dataset using Face-ID and CLIP-Text similarity across 28 diverse prompts (Table 19), as well as using the HPSv2 metric on 100 prompt concepts downloaded from Huggingface repository. As shown in Table 6, DMPVariation achieves lower Face-ID similarity, indicating diverse yet identity-preserving prompts, and improves CLIP-Text similarity by **1.4%**, demonstrating better generalization to novel prompts compared to TI embeddings. The table further shows that DMP-generated prompts generalize better, achieving a **+0.7%** gain over TI on the HPSv2 metric, consistent with the clip score. However, as is typical in generative modeling, HPSv2 does not fully reflect the substantial qualitative gap, as illustrated in Figure 20 in Appendix. While TI prompts produce images that *simply do not comply* with the instruction, DMP prompts consistently produce images that satisfy the instructions.

See Appendix A.9 for a discussion on the limitations of DMP and scope for future works.

## 5 CONCLUSION

In this work, we propose a diffusion meta-learning framework for synthesizing versatile and generalizable prompts across classification, personalization, concept manipulation, and subject variation tasks. Our DMP models demonstrate robust prompt generation that avoids overfitting and achieves strong cross-dataset generalization, effective semantic manipulation, and high identity fidelity. DMPClass shows superior accuracy across diverse tasks (upto **3/12%**) and out-of-distribution datasets (**+1.2%**), DMPMulti enables seamless negative prompt generation, and and DMPVariation synthesizes diverse subject variations with a **+1.4** increase in prompt compliance for generalization compared to the baseline. These results affirm the potential of our diffusion based meta-prompting approach as a powerful and adaptable tool for enhancing prompt quality and generalization in vision applications.

## REPRODUCIBILITY STATEMENT

We are committed to reproducibility of the results presented in the paper and have included all necessary details including the algorithm experimental setup 4, hyperparameters A.10 used for our method. We will release the code and trained models publicly after acceptance of the paper for benefit of the community and drive further research in this area.

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

# A  APPENDIX

## A.1  THEORETICAL ANALYSIS

Let $\mathcal{C}$ denote a distribution over tasks or concepts $c \sim \mathcal{C}$. For each concept $c$, we assume a textual description $y(c)$ (e.g., "a personalization prompt for Jennifer Aniston"), a downstream task with loss $\mathcal{L}_{\text{task}}(S; c)$ that evaluates the performance of a candidate prompt $S$ in prompt space $\mathcal{P}$ when applied to a frozen foundation model $F$, and a repository $\mathcal{R}$ of exemplar prompts $S(c)$ obtained from existing prompt-learning techniques.

Our objective in this section is to bound the *expected downstream loss* when sampling prompts from the pretrained generator:

$$\mathcal{J}_{\text{pre}}(\theta) := \mathbb{E}_{c \sim \mathcal{C}} \Big[ \mathbb{E}_{S \sim p_\theta(\cdot | y(c))} \big[ \mathcal{L}_{\text{task}}(S; c) \big] \Big].$$

**High-level intuition.** When the trained diffusion model $p_\theta(\cdot \mid y)$ closely matches the repository/data distribution $p_{\text{data}}(\cdot \mid y)$ in distributional distance, the expected task loss under $p_\theta$ is close to the expected *repository* task loss under $p_{\text{data}}$. If the repository was constructed to contain useful prompts for downstream tasks (i.e., $p_{\text{data}}$ has low expected task loss), then a small distributional discrepancy implies low expected task loss for $p_\theta$ as well. We make this statement precise with the following bounds.

### A.1.1  DISTRIBUTIONAL DISCREPANCY BOUND

We begin with a straightforward decomposition and use standard total-variation and Pinsker inequalities (Cover & Thomas, 2006; Tsybakov, 2008).

**Proposition 1** (Distributional discrepancy bound)**.** *Let $p_{\text{data}}(\cdot \mid y)$ and $p_\theta(\cdot \mid y)$ be two distributions on prompt space $\mathcal{P}$ for a fixed condition $y$. Assume the task loss $\mathcal{L}_{\text{task}}(S)$ is bounded in $[0, L_{\max}]$. Then*

$$\Big| \mathbb{E}_{S \sim p_\theta}[\mathcal{L}_{\text{task}}(S)] - \mathbb{E}_{S \sim p_{\text{data}}}[\mathcal{L}_{\text{task}}(S)] \Big| \leq 2 L_{\max} \, \text{TV}(p_\theta, p_{\text{data}}), \tag{6}$$

*and by Pinsker's inequality,*

$$\text{TV}(p_\theta, p_{\text{data}}) \leq \sqrt{\tfrac{1}{2} \text{KL}(p_{\text{data}} \| p_\theta)}. \tag{7}$$

*Consequently,*

$$\mathbb{E}_{S \sim p_\theta}[\mathcal{L}_{\text{task}}(S)] \leq \mathbb{E}_{S \sim p_{\text{data}}}[\mathcal{L}_{\text{task}}(S)] + L_{\max} \sqrt{2 \text{KL}(p_{\text{data}} \| p_\theta)}. \tag{8}$$

*Proof.* Let $\ell(S) = \mathcal{L}_{\text{task}}(S)$ and denote $\Delta := \mathbb{E}_{p_\theta}[\ell] - \mathbb{E}_{p_{\text{data}}}[\ell]$. By the definition of total variation (and the fact that $0 \leq \ell \leq L_{\max}$),

$$|\Delta| = \Big| \int \ell(S) \, (p_\theta - p_{\text{data}})(dS) \Big| \leq \int |\ell(S)| \, |p_\theta - p_{\text{data}}|(dS) \leq L_{\max} \int |p_\theta - p_{\text{data}}|(dS).$$

By definition $\text{TV}(p_\theta, p_{\text{data}}) = \frac{1}{2} \int |p_\theta - p_{\text{data}}|$, hence equation 6 holds.

Pinsker's inequality (see e.g. (Cover & Thomas, 2006)) gives equation 7. Combining the two inequalities yields equation 8. $\qquad\qquad\square$

**Remarks.** Inequality equation 8 reduces expected downstream loss under the model to two terms: the expected loss of the repository distribution (which is a function of data collection quality) and the KL divergence between the dataset distribution and the pretrained generator. The latter is controlled by how well the denoiser is trained (see next subsection).

### A.1.2  RELATING DENOISING LOSS TO MODEL-DATA KL

We now recall a standard link between the denoising objective used in DDPMs and a divergence between the model and data distributions (see, e.g., (Vincent, 2011; Song et al., 2020)). Under standard DDPM assumptions and appropriate variance schedule, minimizing the simplified denoising

loss equation 2 is equivalent (up to constants and time discretization effects) to score matching / denoising score-matching which estimates the score function $\nabla_x \log p_{\text{data}}(x)$. A well-trained denoiser implies an accurate score estimator, which in turn implies a small KL divergence between the model and data distributions in $x_0$-space (the space of clean prompts). Formally, one can show:

**Lemma 1** (Denoising loss controls KL (informal)). *Under the standard DDPM/score-matching correspondence and mild regularity conditions, if the denoising risk satisfies*

$$\mathcal{L}_{\text{denoise}}(\theta) \leq \varepsilon,$$

*then the KL divergence between $p_{\text{data}}(x_0 \mid y)$ and $p_\theta(x_0 \mid y)$ admits the bound*

$$\text{KL}\big(p_{\text{data}}(\cdot \mid y)\big\|p_\theta(\cdot \mid y)\big) \leq C\,\varepsilon + o(1),$$

*where $C > 0$ is a constant that depends on the variance schedule, the discretization, and model parametrization; the $o(1)$ term vanishes as the diffusion discretization becomes finer.*

**Remarks.** The lemma is qualitative: precise constants follow from score-matching and likelihood bounds in the DDPM literature (e.g., (Ho et al., 2020b; Song et al., 2020)). The key message is that a small denoising loss implies a small divergence between the learned and data distributions.

Combining Lemma 1 with Proposition 1 yields a bound of the form

$$\mathbb{E}_{S \sim p_\theta}[\mathcal{L}_{\text{task}}(S)] \leq \mathbb{E}_{S \sim p_{\text{data}}}[\mathcal{L}_{\text{task}}(S)] + L_{\max}\sqrt{2C\,\varepsilon} + o(1).$$

### A.1.3 CONCENTRATION FROM FINITE REPOSITORY

So far we have related the model expectation to the (population) data distribution. In practice we only train on finite $\mathcal{R}$ with $n$ samples per condition $y$. Let $\widehat{p}_{\text{data}}$ denote the empirical distribution formed by the repository. By Hoeffding's inequality (or McDiarmid) (Hoeffding, 1963; McDiarmid, 1989), with probability at least $1 - \delta$ over the draw of the repository,

$$\left|\mathbb{E}_{S \sim \widehat{p}_{\text{data}}}[\mathcal{L}_{\text{task}}(S)] - \mathbb{E}_{S \sim p_{\text{data}}}[\mathcal{L}_{\text{task}}(S)]\right| \leq L_{\max}\sqrt{\tfrac{\log(2/\delta)}{2n}}. \tag{9}$$

### A.1.4 DMP PERFORMANCE GUARANTEE

Combining the above pieces yields the following performance guarantee for DMP.

**Proposition 2** (DMP performance guarantee). *Assume the denoising loss satisfies $\mathcal{L}_{\text{denoise}}(\theta) \leq \varepsilon$ and the repository contains $n$ i.i.d. prompts for the condition $y$. Then with probability at least $1 - \delta$ over the repository sample,*

$$\mathbb{E}_{S \sim p_\theta(\cdot|y)}\big[\mathcal{L}_{\text{task}}(S)\big] \leq \mathbb{E}_{S \sim \widehat{p}_{\text{data}}(\cdot|y)}\big[\mathcal{L}_{\text{task}}(S)\big]$$
$$+ L_{\max}\sqrt{2C\,\varepsilon} + L_{\max}\sqrt{\tfrac{\log(2/\delta)}{2n}} + o(1), \tag{10}$$

*where $C$ is the constant from Lemma 1 and the $o(1)$ term accounts for discretization error in the diffusion approximation.*

*Proof.* Start from Proposition 1 applied to $p_\theta$ and $p_{\text{data}}$:

$$\mathbb{E}_{p_\theta}[\ell] \leq \mathbb{E}_{p_{\text{data}}}[\ell] + 2L_{\max}\sqrt{\tfrac{1}{2}\,\text{KL}(p_{\text{data}}\|p_\theta)}.$$

Apply Lemma 1 to bound the KL by $C\varepsilon + o(1)$; hence

$$\mathbb{E}_{p_\theta}[\ell] \leq \mathbb{E}_{p_{\text{data}}}[\ell] + L_{\max}\sqrt{2C\,\varepsilon} + o(1).$$

Now replace the population expectation $\mathbb{E}_{p_{\text{data}}}[\ell]$ by the empirical expectation $\mathbb{E}_{\widehat{p}_{\text{data}}}[\ell]$ and apply Proposition 9 (Hoeffding) which with probability at least $1 - \delta$ yields the stated sampling error term $L_{\max}\sqrt{\tfrac{\log(2/\delta)}{2n}}$. Combining these terms gives equation 10. $\square$

**Interpretation.** Bound equation 10 decomposes the expected task loss under the pretrained generator into (i) the empirical repository loss (quality of collected prompts), (ii) an approximation term controlled by the denoising risk (how well DMP models the repository), and (iii) a sampling term that vanishes as the repository size $n$ grows.

Table 9: **Conceptual differences between DMP and prior prompt learning methods.** Unlike prior works that refine prompts using data for a specific task or example, DMP treats prompts as a *distribution* and trains a diffusion meta-model that amortizes prompt generation across many tasks and repositories. DMP does not require access to the original data used to train the prompts (L159–L161) and supports one-shot sampling, composition, and negative prompting without per-task optimization.

| Method | Task Data/ Loss Free | Supports Multiple Tasks | Learns across Tasks | Stores Prompts | Core Idea |
|---|---|---|---|---|---|
| Stylus / Retrieval-based | ✓ | ✗ | ✓ | ✓ | Searches large repositories; limited generalization. |
| *Diffusion-based Methods* | | | | | |
| Prompt Diffusion | ✗ | ✗ | ✗ | ✓ | Refines prompts for a specific task using data. |
| Diff-Prompt | ✗ | ✗ | ✗ | ✓ | Mask-supervised diffusion for a single task. |
| Neural Network Diffusion | ✗ | ✗ | ✗ | ✗ | Diffusion to generate model weights for a particular task/dataset (weight-generation). |
| Conditional LoRA Param Gen. | ✗ | ✓ | ✓ | ✗ | Diffusion/hypernetwork that generates LoRA adapters conditioned on task (Cond P-Diff / CondLoRA). |
| DiffLoRA | ✗ | ✓ | ✓ | ✗ | Diffusion model predicts personalized low-rank (LoRA) weights at inference (zero-shot personalization). |
| *Prompt Learning Methods* | | | | | |
| Hierarchical Variational TTP | ✗ | ✗ | ✗ | ✓ | Test-time variational prompt generator relying on task-specific data features. |
| Language-Aware Soft Prompting (LASP) | ✗ | ✗ | ✗ | ✓ | Text-conditioned optimization for soft prompts, task-specific. |
| Consistency-guided PL (CoPrompt) | ✗ | ✗ | ✗ | ✓ | Consistency loss improves prompt robustness; task-specific. |
| PromptKD | ✗ | ✗ | ✗ | ✓ | Distills prompts from teacher to student in unsupervised setting; task-specific. |
| Dual-Prompt Collaboration (DPC) | ✗ | ✗ | ✗ | ✓ | Dual-prompt collaboration requiring tuned prompt parameters; task-specific. |
| Bayesian Prompt Learning (BPL) | ✗ | ✗ | ✗ | ✓ | Bayesian uncertainty modeling over task-specific prompts. |
| Patch-Prompt Aligned Bayesian Prompt Tuning | ✗ | ✗ | ✗ | ✓ | Bayesian hierarchical prompt generation (label-specific stochastic prompts); per-task tuning. |
| *Meta-Learning Methods* | | | | | |
| Prompt Learning via Meta-Regularization (ProMetaR) | ✗ | ✓ | ✓ | ✓ | Meta-regularization to improve prompt generalization across tasks; requires training data. |
| Gradient-Regulated Meta-Prompt (GRAM) | ✗ | ✓ | ✓ | ✓ | Meta-learned prompt init + gradient regulator for few-shot cross-domain generalization. |
| AWT (Augment, Weight, Transport) | ✗ | ✓ | ✗ | ✗ | Augment inputs, dynamically weight them, and use optimal transport to adapt VLMs without extra training. |
| PRewrite (Prompt Rewriting w/ RL) | ✗ | ✓ | ✗ | ✓ | LLM-based prompt rewriter trained with RL to improve downstream task performance. |
| **DMP (Ours)** | ✓ | ✓ | ✓ | ✗ | **Learns a prompt distribution; one-shot sampling, composition, negative prompts.** |

## A.2 RELATION TO PRIOR METHODS

Table 9 illustrates how DMP differs from prior prompt-learning approaches across key axes such as requiring task-specific data or loss, supporting multiple tasks in a zero-shot manner, learning across multiple tasks and the need for storage of prompts. Classical prompt learning techniques are *task and model specific*, and require *task specific data and losses*. DMP instead learns the distribution of prompts from a prompt repository (produced by these prompt learning techniques). It can then be used to sample prompts for *many tasks*. Note that *DMP is not task specific and does not require task specific losses or data, just a prompt repository*. This makes DMP as a general prompt generator, free from the data and optimization constraints that characterize prior prompt-learning techniques. We show that, in many cases, DMP sampled prompts even outperform prompt learning prompts. For example, the classification results of Table 11 shows that DMP outperforms the base method on the unseen classes of the very same dataset, despite never accessing a single image.

## A.3 DMP FOR CLASSIFICATION

**Generalization to Novel Tasks.** We assessed the compositionality of DMP in Table 8, motivated by our hypothesis that conditioning on class label names naturally facilitates compositional generalization. We evaluated this for CoOp prompts where we showed that DMPCoOp achieved consistent and often large gains as noted below:

All split: +4.3% average improvement, with top combination pairs showing very large gains (e.g., EuroSAT & OxfordFlowers: 35.5 to 61.2, +25.7).

Table 10: Comparison of TAC vs DMPTAC across All, Base, and New splits for 55 dataset combination pairs.

| Split | Avg TAC (%) | Avg DMPTAC (%) | Avg Gain (%) | Top-3 Pairs: TAC → DMPTAC, Gain | | |
|-------|-------------|----------------|--------------|------------------|------------------|-----------|
| All | 63.2 | 63.3 | 0.1 | OxfordFlowers&OxfordPets | 78.4→80.2 | (g=1.8) |
| | | | | ImageNet&OxfordFlowers | 51.4→53.1 | (g=1.7) |
| | | | | OxfordFlowers&SUN397 | 48.2→49.9 | (g=1.7) |
| Base | 69.6 | 69.8 | 0.2 | OxfordFlowers&OxfordPets | 85.0→88.0 | (g=3.0) |
| | | | | FGVCAircraft&SUN397 | 54.6→56.7 | (g=2.1) |
| | | | | SUN397&StanfordCars | 65.1→66.8 | (g=1.7) |
| New | 69.7 | 70.2 | 0.4 | OxfordFlowers&OxfordPets | 79.6→82.6 | (g=3.0) |
| | | | | OxfordFlowers&SUN397 | 57.9→60.2 | (g=2.3) |
| | | | | EuroSAT&FGVCAircraft | 51.1→53.3 | (g=2.2) |

Base split: +4.7% average improvement (e.g., ImageNet & Flowers: 32.6 to 70.0, +37.4).

New split: +3.3% average improvement (e.g., Pets & SUN397: 47.6 to 69.7, +22.1).

We now extend this evaluation to TAC prompts. Table 10 shows the results of this experiment. For TAC-based models, as noted in the paper, it includes deeper prompt components such as projection and head matrices, which DMP does not model as DMP only synthesizes the shallow text prompts. Despite this structural limitation, DMPTAC still achieves positive improvements (e.g., +0.4% average in new split for 55 dataset combination pairs) and large gains on individual pairs (e.g., Flowers & Pets with +3.0% gain).

This demonstrates that even under simple, synthetic identifiers, DMP enhances generalization and maintains strong performance in multi-dataset settings.

**Base to New generalization.** Figure 9 shows the zoomed version of the DMP framework for ease of viewing. Table 11 shows the complete results of DMPClass for CoOp, CoCoOp, CoPrompt, Maple,

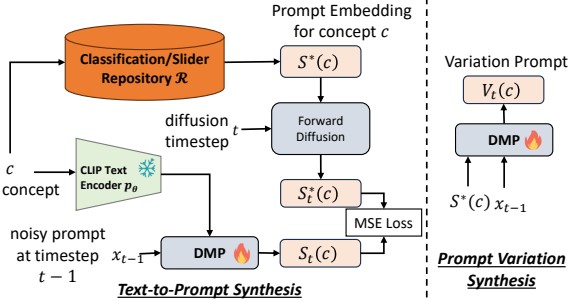

Figure 9: **(Zoomed Version) Left:** Diffusion Meta-Prompt framework for Text-to-Prompt synthesis. **Right:** Prompt Variation synthesis conditioned on learned Textual Inversion Prompts.

and TAC methods over the baselines across 11 datasets for Base-to-novel class generalization setting. The reported results are the average over three seed runs. **DMP consistently improves average accuracy across three independent random seeds on both all and novel classes**, outperforming existing prompt learning methods. Specifically, DMP yields gains of **+1.3% (All) and +3.0% (New)** over CoOp, **+0.6%/+0.8%** over CoPrompt, **+0.1%/+1.5%** over CoCoOp, **+0.4%/+0.8%** over MaPLe, and **+0.4%/+0.4%** over TAC. Importantly, these improvements are achieved while tuning only the text prompts, leaving projection matrices and model weights fixed.

The benefits of DMP are especially pronounced on datasets underrepresented in CLIP pretraining, where adaptation is more challenging. On **EuroSAT**, DMP achieves relative gains of **+7.5%/+2.8%/+1.0%/+5.1%/+1.3%** over CoOp, CoPrompt, CoCoOp, MaPLe, and TAC, respectively. On **DTD**, the improvements are **+4.0%/+4.5%/+3.7%/+1.1%/+0.9%**, and on **FGVC**, they are **+4.5%/+0.8%/+1.9%/+0.9%/+0.4%**. These results highlight that DMP not only delivers consistent gains across methods and datasets but also shows **stronger generalization on novel and less common domains**, demonstrating its superiority as a prompt learning approach.

Figure 11 and Figure 12 show the zoomed version (for ease of viewing) of cross-dataset and cross-task generalization results of DMPCoOp prompts over the baseline CoOp prompts, respectively.

Table 11: Base2new generalization per dataset: performance for All, Base, and New classes, and HM (Harmonic Mean). The results reported are the average over three seed runs. * denotes our implementation as the code is not publicly available.

| Method | (a) Average | | | | (b) ImageNet | | | | (c) Caltech101 | | | | (d) OxfordPets | | | |
|---|---|---|---|---|---|---|---|---|---|---|---|---|---|---|---|---|
| | All | Base | New | HM | All | Base | New | HM | All | Base | New | HM | All | Base | New | HM |
| VAE (Kingma & Welling, 2014) | 58.6 | 62.7 | 68.3 | 65.0 | 57.0 | 64.1 | 57.8 | 60.8 | 89.1 | 91.7 | 94.3 | 92.9 | 90.1 | 92.0 | 97.1 | 94.5 |
| Top-1 Retrieval (Luo et al., 2024) | 67.5 | 80.7 | 72.8 | 76.5 | 50.5 | 78.4 | 47.9 | 59.5 | 37.3 | 76.9 | 46.4 | 57.9 | 69.9 | 85.8 | 66.8 | 75.1 |
| Transformer (Vaswani et al., 2017) | 68.3 | 82.3 | 69.4 | 75.3 | 68.1 | 76.4 | 66.8 | 71.3 | 94.8 | 98.2 | 94.8 | 96.5 | 89.9 | 94.6 | 95.4 | 95.0 |
| GPT-2 (Radford et al., 2019) | 68.1 | 81.8 | 69.1 | 74.9 | 68.0 | 76.2 | 66.8 | 71.2 | 94.8 | 98.3 | 95.0 | 96.6 | 90.1 | 94.7 | 95.9 | 95.3 |
| Prompt Diffusion* (Du et al., 2024) | 67.0 | 69.9 | 73.0 | 71.4 | 54.7 | 60.6 | 56.4 | 58.4 | 91.3 | 94.4 | 93.2 | 93.8 | 92.3 | 94.3 | **97.5** | 95.9 |
| CoOp (Zhou et al., 2022b) | 68.8 | **82.3** | 70.4 | 75.9 | 68.5 | **76.5** | 67.2 | 71.5 | 94.8 | 98.3 | 95.0 | 96.6 | 90.1 | 94.7 | 95.9 | 95.3 |
| DMPCoOp | **70.1** | 80.3 | **73.4** | **76.5** | **68.7** | 75.4 | **68.8** | **72.0** | 94.8 | 98.3 | 95.3 | 96.8 | 91.7 | 95.4 | 97.3 | 96.3 |
| Δ | +1.3 | -2.0 | +3.0 | +0.6 | +0.2 | -1.1 | +1.6 | +0.5 | 0.0 | 0.0 | +0.3 | +0.2 | +1.6 | +0.7 | +1.4 | +1.0 |
| CoCoOp (Zhou et al., 2022a) | **70.1** | 80.7 | 72.5 | 76.0 | 69.9 | **75.8** | 70.8 | 73.2 | 93.7 | **97.8** | 93.2 | 95.4 | 91.2 | **95.1** | 97.6 | 96.3 |
| DMPCoCoOp | 70.2 | 79.5 | **74.0** | 76.4 | 70.1 | 75.8 | 71.2 | 73.4 | 93.8 | 97.7 | 93.7 | 95.7 | 92.1 | 95.0 | 97.8 | 96.4 |
| Δ | +0.1 | -1.2 | +1.5 | +0.4 | +0.2 | 0.0 | +0.4 | +0.2 | +0.1 | -0.1 | +0.5 | +0.3 | +0.9 | -0.1 | +0.2 | +0.1 |
| CoPrompt (Roy & Etemad, 2024) | 72.3 | **83.1** | 74.6 | 78.3 | **70.7** | 76.7 | 71.4 | **73.9** | 95.8 | 98.7 | 95.3 | 97.0 | 91.1 | **95.3** | 97.0 | 96.1 |
| DMPCoPrompt | **72.9** | 82.5 | **75.4** | **78.5** | 70.7 | 76.6 | 71.5 | 73.9 | 95.7 | 98.7 | 95.4 | 97.0 | 91.6 | 95.2 | 97.2 | 96.2 |
| Δ | +0.6 | -0.6 | +0.8 | +0.2 | 0.0 | -0.1 | +0.1 | 0.0 | -0.1 | 0.0 | +0.1 | 0.0 | +0.5 | -0.1 | +0.2 | +0.1 |
| Maple (khattak et al., 2023) | 72.0 | **82.2** | 75.1 | 78.2 | 70.2 | 76.7 | 70.5 | 73.5 | 94.5 | **98.0** | 94.3 | 96.1 | **92.3** | 95.4 | **97.8** | **96.6** |
| DMPMaple | **72.4** | 82.0 | **75.9** | **78.6** | 70.3 | 76.8 | 70.6 | 73.6 | 94.8 | 98.0 | 95.5 | 96.7 | 92.3 | 95.4 | 97.6 | 96.5 |
| Δ | +0.4 | -0.2 | +0.8 | +0.4 | +0.1 | +0.1 | +0.1 | +0.1 | +0.3 | 0.0 | +1.2 | +0.6 | 0.0 | 0.0 | -0.2 | -0.1 |
| TAC (Hao et al., 2025) | 74.6 | **85.2** | 77.1 | 80.8 | 71.3 | **78.5** | 71.0 | **74.6** | **95.2** | 98.6 | 95.0 | 96.7 | **93.1** | **96.0** | 98.0 | **97.0** |
| DMPTAC | **75.0** | 85.1 | **77.5** | **80.9** | **71.4** | 78.5 | 71.2 | 74.6 | 95.2 | 98.6 | 95.0 | 96.8 | 93.1 | 95.9 | **98.2** | 97.0 |
| Δ | +0.4 | -0.1 | +0.4 | +0.1 | +0.1 | 0.0 | +0.2 | 0.0 | 0.0 | 0.0 | 0.0 | +0.1 | 0.0 | -0.1 | +0.2 | 0.0 |

| Method | (e) StanfordCars | | | | (f) Flowers102 | | | | (g) Food101 | | | | (h) FGVC Aircraft | | | |
|---|---|---|---|---|---|---|---|---|---|---|---|---|---|---|---|---|
| | All | Base | New | HM | All | Base | New | HM | All | Base | New | HM | All | Base | New | HM |
| CoOp (Zhou et al., 2022b) | 68.7 | **76.7** | 68.2 | 72.2 | 74.5 | **96.7** | 68.3 | 80.1 | 84.9 | **90.0** | 89.9 | **89.9** | 25.1 | **36.9** | 27.1 | 31.2 |
| DMPCoOp | **69.3** | 74.5 | **72.0** | **73.2** | **75.6** | 93.4 | **72.2** | **81.4** | **86.4** | 89.6 | 89.9 | 89.7 | **26.3** | 35.7 | **31.6** | **33.5** |
| Δ | +0.6 | -2.2 | +3.8 | +1.0 | +1.1 | -3.3 | +3.9 | +1.3 | +1.5 | -0.4 | 0.0 | -0.2 | +1.2 | -1.2 | +4.5 | +2.3 |
| CoCoOp (Zhou et al., 2022a) | **68.9** | 71.2 | 73.2 | **72.2** | 74.4 | **94.7** | 70.1 | 80.6 | 85.8 | 90.6 | 91.3 | 90.9 | **26.0** | **35.5** | 32.1 | 33.7 |
| DMPCoCoOp | 68.6 | 69.9 | **74.4** | 72.1 | **74.6** | 90.6 | **72.8** | **80.8** | **86.6** | 90.7 | 91.5 | 91.1 | 25.9 | 32.6 | **34.0** | 33.2 |
| Δ | -0.3 | -1.3 | +1.2 | -0.1 | +0.2 | -4.1 | +2.7 | +0.2 | +0.8 | +0.1 | +0.2 | +0.2 | -0.1 | -2.9 | +1.9 | -0.5 |
| CoPrompt (Roy & Etemad, 2024) | 68.3 | 74.0 | 71.0 | 72.5 | 81.4 | **96.5** | 75.8 | 84.9 | 86.5 | 90.3 | 91.6 | 90.9 | **28.9** | **37.5** | 35.5 | **36.4** |
| DMPCoPrompt | 68.3 | 73.9 | **71.1** | 72.5 | **81.5** | 96.0 | **76.2** | **85.0** | **86.6** | 90.3 | 91.7 | 91.0 | 28.9 | 36.3 | **36.3** | 36.3 |
| Δ | 0.0 | -0.1 | +0.1 | 0.0 | +0.1 | -0.5 | +0.4 | +0.1 | +0.1 | 0.0 | +0.1 | +0.1 | 0.0 | -1.2 | +0.8 | -0.1 |
| Maple (khattak et al., 2023) | **69.8** | 72.9 | 74.0 | 73.4 | 76.9 | **95.9** | 72.3 | 82.5 | 86.9 | 90.7 | 92.1 | 91.4 | 28.4 | **37.5** | 35.5 | 36.4 |
| DMPMaple | 69.7 | 72.7 | **74.1** | 73.4 | **77.5** | 95.5 | **73.1** | **82.8** | **87.1** | 90.8 | 92.1 | 91.5 | **28.8** | 37.0 | **36.4** | **36.7** |
| Δ | -0.1 | -0.2 | +0.1 | 0.0 | +0.6 | -0.4 | +0.8 | +0.3 | +0.2 | +0.1 | 0.0 | +0.1 | +0.4 | -0.5 | +0.9 | +0.3 |
| TAC (Hao et al., 2025) | **74.2** | **81.2** | 74.8 | **77.9** | 81.2 | **98.0** | 75.9 | 85.6 | **86.9** | **91.0** | 91.9 | 91.4 | **34.3** | 45.1 | 38.0 | 41.2 |
| DMPTAC | 74.2 | 81.0 | **74.9** | 77.8 | **81.4** | 98.0 | 76.1 | **85.7** | 86.9 | 90.9 | 91.9 | 91.4 | 34.2 | **45.3** | **38.4** | **41.5** |
| Δ | 0.0 | -0.2 | +0.1 | -0.1 | +0.2 | 0.0 | +0.2 | +0.1 | 0.0 | -0.1 | 0.0 | 0.0 | -0.1 | +0.2 | +0.4 | +0.3 |

| Method | (i) SUN397 | | | | (j) DTD | | | | (k) EuroSAT | | | | (l) UCF101 | | | |
|---|---|---|---|---|---|---|---|---|---|---|---|---|---|---|---|---|
| | All | Base | New | HM | All | Base | New | HM | All | Base | New | HM | All | Base | New | HM |
| CoOp (Zhou et al., 2022b) | **67.6** | **80.8** | 72.6 | **76.5** | 51.9 | **80.8** | 51.8 | 63.1 | 63.2 | **90.8** | 72.9 | 80.9 | 67.6 | **83.4** | 65.3 | 73.2 |
| DMPCoOp | 67.3 | 80.4 | **72.9** | 76.5 | **53.7** | 75.1 | **55.8** | **64.0** | **65.9** | 85.6 | **80.7** | **83.1** | **71.7** | 79.6 | **70.7** | **74.9** |
| Δ | -0.3 | -0.4 | +0.3 | 0.0 | +1.8 | -5.7 | +4.0 | +0.9 | +2.7 | -5.2 | +7.8 | +2.2 | +4.1 | -3.8 | +5.4 | +1.7 |
| CoCoOp (Zhou et al., 2022a) | **69.8** | **79.7** | 76.6 | **78.1** | 52.3 | **77.5** | 54.8 | 64.2 | **66.0** | **87.9** | 65.6 | 74.9 | **72.7** | **82.2** | 72.1 | 76.8 |
| DMPCoCoOp | 69.5 | 78.4 | **77.5** | 78.0 | **52.9** | 75.8 | **58.5** | **66.0** | 65.8 | 87.2 | **66.6** | **75.4** | 72.5 | 80.6 | **76.0** | **78.2** |
| Δ | -0.3 | -1.3 | +0.9 | -0.1 | +0.6 | -1.7 | +3.7 | +1.8 | -0.2 | -0.7 | +1.0 | +0.5 | -0.2 | -1.6 | +3.9 | +1.4 |
| CoPrompt (Roy & Etemad, 2024) | 72.5 | **82.3** | 79.6 | 80.9 | 56.3 | **82.1** | 57.6 | 67.6 | 67.1 | **94.3** | 66.8 | 78.0 | **76.5** | **86.8** | 78.7 | **82.5** |
| DMPCoPrompt | **72.6** | 82.2 | **79.8** | **81.0** | **57.4** | 80.5 | **62.1** | **70.1** | **70.7** | 91.1 | **69.6** | **78.6** | 76.5 | 86.4 | **78.8** | 82.4 |
| Δ | +0.1 | -0.1 | +0.2 | +0.1 | +1.1 | -1.6 | +4.5 | +2.5 | +3.6 | -3.2 | +2.8 | +0.6 | 0.0 | -0.4 | +0.1 | -0.1 |
| Maple (khattak et al., 2023) | **71.2** | **80.8** | 78.7 | **79.7** | 55.8 | **80.2** | 59.2 | 68.1 | 72.2 | **93.7** | 72.9 | 81.9 | **73.8** | **82.9** | 78.6 | **80.7** |
| DMPMaple | 70.9 | 80.2 | 78.7 | 79.4 | **55.9** | 79.2 | **60.3** | **68.4** | **74.4** | 93.1 | **78.0** | **84.9** | 73.6 | 82.9 | 78.4 | 80.6 |
| Δ | -0.3 | -0.6 | 0.0 | -0.3 | +0.1 | -1.0 | +1.1 | +0.3 | +2.2 | -0.6 | +5.1 | +3.0 | -0.2 | 0.0 | -0.2 | -0.1 |
| TAC (Hao et al., 2025) | **73.1** | **83.6** | 79.7 | 81.6 | 59.1 | **83.6** | 62.7 | 71.6 | 76.4 | **94.3** | 80.2 | 86.6 | **78.2** | **87.2** | 81.1 | 84.1 |
| DMPTAC | 73.1 | 83.6 | **79.9** | **81.7** | **59.3** | 83.3 | **63.6** | **72.1** | **76.7** | 93.9 | **81.5** | **87.2** | 78.2 | 87.2 | **81.4** | **84.2** |
| Δ | 0.0 | 0.0 | +0.2 | +0.1 | +0.2 | -0.3 | +0.9 | +0.5 | +0.3 | -0.4 | +1.3 | +0.6 | 0.0 | 0.0 | +0.3 | +0.1 |

Table 12 shows the full results of cross-task generalization experiment described in section 4.1. We note that DMPCoOp obtains higher accuracies consistently across both the Mean Treecut Accuracy and Hierarchical Consistency Accuracy metrics on all the datasets considered. Notably, DMPCoOp

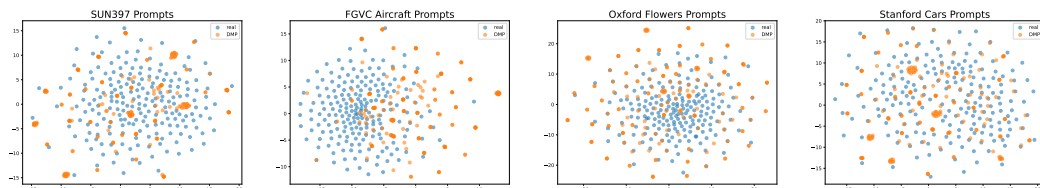

Figure 10: **t-SNE visualizations of prompt embeddings.** Each figure shows real prompts (blue) and DMP-generated prompts (orange) with different noise seeds for different datasets: (a) SUN397, (b) FGVC-Aircraft, (c) Oxford Flowers, (d) Stanford Cars. Generated prompts broadly overlap with the real prompt manifolds while exhibiting greater spread, demonstrating both fidelity and diversity across domains.

obtains **+11.4%** and **+5%** improvement on SUN dataset for MTA and HCA respectively. This shows that DMPCoOp prompts are more robust and generalize well beyond the task on which it was trained.

We further provide the results of cross-dataset generalization for TAC model in Table 15, cross-domain experiments in Table 16, and cross-task generalization in Table 17. The gains over baseline TAC are larger for cross-task generalization where DMPTAC prompts obtains **+10% on SUN and +9% on Imagenet-Sketch** datasets for hierarchical classification.

**t-SNE visualization.** To assess the fidelity and diversity of prompts synthesized by the DMP framework, we conducted an embedding space analysis comparing real prompts from the repository $\mathcal{R}$ with DMP-generated prompts sampled using different random noise seeds. Figure 10 presents a two-dimensional t-SNE projection of both sets of embeddings, with real prompts in blue and generated prompts in orange for SUN397, FGVC, Oxford Flowers and Stanford Cars datasets respectively from left to right.

The visualization reveals that generated prompts broadly overlap with the real prompt manifold, while also exhibiting a greater spread, indicative of higher variability. This suggests that the model captures the underlying structure of prompt space without resorting to memorization, while also producing novel variations.

To quantify these observations, we computed the fraction of a prompt's 5 nearest neighbors (in embedding space) that share the same class label between real and generated prompts across 11 datasets. On average, 70.7% of generated prompts share the same nearest-neighbor labels as their real counterparts, showing strong semantic alignment between real and generated embeddings while ensuring diversity.

Table 12: Cross-task generalization per dataset: performance for Mean Treecut Accuracy (MTA) (Wu et al., 2024), and Hierarchical Consistency Accuracy (HCA) (Wu et al., 2024).

| Method | ImageNet | | -V2 | | -S | | -R | | -A | | SUN397 | |
|---|---|---|---|---|---|---|---|---|---|---|---|---|
| | MTA | HCA | MTA | HCA | MTA | HCA | MTA | HCA | MTA | HCA | MTA | HCA |
| CoOp | 40.7 | 0.8 | 38.9 | 0.8 | 34.4 | 0.5 | **57.7** | 9.9 | 43.8 | 4.1 | 19.0 | 31.3 |
| DMPCoOp | **47.4** | **2.6** | **45.0** | **2.1** | **40.2** | **2.0** | 57.3 | **18.7** | **55.5** | **5.5** | **30.4** | **36.3** |
| Δ | +6.7 | +1.8 | +6.1 | +1.3 | +5.8 | +1.5 | -0.4 | +8.8 | +11.7 | +1.3 | +11.4 | +5.0 |

**Impact of Classifier-Free Guidance (CFG) scales.** The performance of DMP with different guidance values is shown in Table 13. The trends show that the average accuracy across all datasets decreases with increasing guidance scales.

Table 13: Comparison with different classifier free guidance scales. Base2new generalization per dataset: performance for All, Base, and New classes, and HM (Harmonic Mean).

| Method | (a) Average (scale 4.5) | | | | (b) Average (scale 5.5) | | | | (c) Average (scale 6.5) | | | | (d) Average (scale 7.5) | | | |
|---|---|---|---|---|---|---|---|---|---|---|---|---|---|---|---|---|
| | All | Base | New | HM | All | Base | New | HM | All | Base | New | HM | All | Base | New | HM |
| DMPCoOp | 70.1 | 80.3 | 73.4 | 76.5 | 69.6 | 80.3 | 70.6 | 76.3 | 69.6 | 79.3 | 70.4 | 75.0 | 68.1 | 79.4 | 70.1 | 75.0 |

**Variational Autoencoder Reconstruction.** Table 14 shows the reconstruction accuracy of the CoOp prompts for the trained Variational Autoencoder (VAE) across different datasets. The autoencoder reconstructs the prompts almost perfectly with only 0.1% difference on average across all the datasets.

Table 14: Quantitative results of VAE reconstruction: Comparison against CoOp prompts across various datasets. The VAE reconstructs the CoOp prompts baseline almost perfectly with only 0.1% difference on average.

| Model | ImageNet | Oxford flowers | Oxford pets | Stanford cars | Caltech 101 | Food101 | FGVC Aircraft | SUN397 | DTD | EuroSAT | UCF101 | Average |
|---|---|---|---|---|---|---|---|---|---|---|---|---|
| CoOp | **68.5** | **72.6** | **90.1** | **68.7** | **94.8** | 84.9 | **25.1** | **67.6** | 51.9 | 58.9 | **66.2** | **68.1** |
| Autoencoder | 68.5 | 72.0 | 89.6 | 68.6 | 94.6 | **85.1** | 25.0 | 67.0 | **52.0** | **59.5** | 66.1 | 68.0 |
| △ | 0.0 | -0.6 | -0.5 | -0.1 | -0.2 | +0.2 | -0.1 | -0.6 | +0.1 | +0.6 | -0.1 | -0.1 |

Table 15: Ablation study on Cross-dataset generalization of DMPTAC Imagenet prompts

| Model | ImageNet | Oxford flowers | Oxford pets | Stanford cars | Caltech 101 | Food101 | FGVC Aircraft | SUN397 | DTD | EuroSAT | UCF101 | Average |
|---|---|---|---|---|---|---|---|---|---|---|---|---|
| TAC | 71.3 | 70.2 | 90.6 | 63.9 | 94.2 | 84.5 | 23.9 | 66.9 | 45.1 | 43.0 | 66.5 | 65.4 |
| DMPTAC | **71.4** | 71.0 | 91.0 | 63.8 | 94.0 | 84.7 | 24.6 | 67.0 | 45.8 | 44.4 | 66.8 | 65.9 |
| △ | +0.1 | +0.8 | +0.4 | -0.1 | -0.2 | +0.2 | +0.7 | +0.1 | +0.7 | +1.4 | +0.3 | +0.5 |

## A.4 DMP FOR SLIDERS

Figure 15 shows additional qualitative results of slider prompts generated by DMP for the attributes "long hair" and "chubby".

Figure 16 presents a qualitative comparison of images generated by Concept Sliders (using the author-provided models), Prompt Sliders, and DMP. The results show that Concept Sliders struggle to induce the intended attributes, even at higher scales, due to their sensitivity to training hyperparameters, which requires careful tuning as noted in (Sridhar & Vasconcelos, 2024). We observe that DMP-generated prompts produce images that are qualitatively similar to those from Prompt Sliders. However, the original Prompt Sliders method has limitations in maintaining subject identity at higher scales, as discussed in (Sridhar & Vasconcelos, 2024). In contrast, DMP—despite being trained with the Prompt Sliders embeddings—demonstrates greater robustness, effectively preserving subject identity even at relatively higher scales. These result corroborate the results observed in Table 2 of the paper.

Figure 22 illustrates the results of pure negative prompts, where the sliders yield images with attributes opposite to the specified concepts, such as shorter hair or a neutral (non-smiling) expression.

## A.5 DMP FOR VARIATIONS

**Generalization.** Figure 17 shows images synthesized for the variation prompts generated by DMP-Variation model, for a common seed. Note that the variation prompts are conditioned by the Textual inversion embedding for the target concept, which is illustrated by a GT image in the figure. The figure shows that DMPVariation produces successful variation prompts for general objects as diverse as birds and statues *despite being trained only on CelebA-face identities*.

**Qualitative Results.** Figure 18 shows additional qualitative results for variations generated by DMPVariation model.

Figure 19 shows the synthesized prompts that produce variations of a person, conditioned on the textual inversion embedding of the person. Note that, while all images are synthesized with the same SDXL seed, they exhibit a diversity of background scenes, hair patterns, clothing, etc. This is an additional benefit of the natural prompt variability of DMP-based personalization: to increase the diversity of the synthesized images.

Figure 20 presents qualitative results comparing DMP and TI in generating personalized subject images across diverse contexts. The variation prompts produced by DMP demonstrate greater robustness and generalization, effectively adapting to different contexts while preserving subject identity. In contrast, Textual Inversion tends to overfit to the subject, leading to poor generalization.

Table 16: Domain generalization accuracy (%) for TAC prompts sampled on ImageNet.

| | Source | Target | | | | |
|---|---|---|---|---|---|---|
| Method | ImageNet | -V2 | -S | -A | -R | Average |
| TAC | 71.3 | 64.6 | 48.3 | 48.6 | 76.6 | 61.9 |
| DMPTAC | **71.4** | **64.7** | 48.5 | **48.8** | **76.7** | **62.0** |

Table 17: Cross-task generalization per dataset: performance for Mean Treecut Accuracy (MTA).

| Method | ImageNet | -V2 | -S | -R | -A | SUN397 |
|---|---|---|---|---|---|---|
| TAC | 71.3 | 34.1 | 28.3 | **5.5** | 36.5 | 30.0 |
| DMPTAC | **71.4** | **39.6** | **37.3** | **5.7** | **38.9** | **40.0** |
| △ | +0.1 | +5.5 | +9.0 | +0.2 | +2.4 | +10.0 |

Table 18: Ablation study on Cross-dataset generalization of DMPCoOp Imagenet prompts: Comparison of classnames against dataset names as prompt inputs to the DMPCoOp model.

| Model | ImageNet | Oxford Flowers | Oxford pets | Stanford cars | Caltech 101 | Food101 | FGVC Aircraft | SUN397 | DTD | EuroSAT | UCF101 | Average |
|---|---|---|---|---|---|---|---|---|---|---|---|---|
| CoOp | 68.5 | 66.5 | 88.5 | 61.9 | **92.7** | 84.8 | 15.2 | 60.7 | 40.9 | **46.9** | 65.3 | 62.9 |
| DMPCoOp (Dataset Names) | 68.3 | 65.2 | 88.0 | 63.4 | 92.7 | 83.9 | 16.8 | 62.6 | **41.0** | 46.3 | 66.6 | 63.2 |
| Δ (Dataset) | -0.2 | -1.3 | -0.5 | +1.5 | 0.0 | -0.9 | +1.6 | +1.9 | +0.1 | -0.6 | +1.3 | +0.3 |
| DMPCoOp (Class Names) | **68.7** | **69.0** | **89.2** | 62.4 | 91.4 | **85.7** | 20.5 | **62.9** | 40.4 | 46.2 | **66.7** | **63.9** |
| Δ (Class) | +0.2 | +2.5 | +0.7 | +0.5 | -1.3 | +0.9 | +5.3 | +2.2 | -0.5 | -0.7 | +1.4 | +1.0 |

Figure 11: **(Zoomed Version) Cross-dataset gener-** Figure 12: **(Zoomed Version) Cross-task generaliza-**
**alization**: Comparison of DMPCoOp against CoOp **tion**: Comparison against CoOp prompts across vari-
prompts sampled for ImageNet. DMPCoOp general- ous datasets. DMPCoOp generalizes better than CoOp
izes better than CoOp with a 1% average accuracy gain. baseline with a 5-12% average accuracy gain.

For instance, it completely fails to generate correct images for the Buddha statue and succeeds in only a single scenario for the duck and dog subjects. Table 24 shows the detailed HPSv2 scores for the results presented in Table 6 of the paper.

The observed generalization of DMP beyond its training domain can be explained by two complementary principles: (1) Latent Structure of Prompt Space. (2) Distributional Robustness of Diffusion Models.

1. Prompt embeddings encode semantic concepts in a continuous, compositional latent space (Wang et al., 2023). Even though TI is trained on faces, the learned repository $\mathcal{R}$ spans a manifold of semantic representations that share structural similarities with other concepts (e.g., animal attributes, artistic styles). Diffusion in this space does not memorize individual prompts but learns a generative prior over semantic transformations, enabling extrapolation to novel concepts.

2. Diffusion models trained on corrupted versions of $S(c)$ implicitly learn a score function that approximates the gradient of the log data distribution (Song et al., 2020). Since score matching enforces local smoothness in high-dimensional space, the denoiser learns to interpolate meaningfully between prompt embeddings, even outside the training distribution. This mechanism explains why the model can adaptively synthesize coherent prompts for unseen categories.

Together, these mechanisms suggest that DMP does not merely replicate memorized prompts but learns a domain-agnostic generative prior over prompt space, providing a rationale for its robust generalization.

**Identity Composition.** Figure 21 demonstrates additional results of combining two identities, displayed on the left, using DMPMulti model to synthesize identity prompts with Eq. 8. The synthesized identity clearly incorporates prominent features from both original faces, such as the nose and chin, resulting in a cohesive blend of attributes.

**Subject Composition.** Figure 23 illustrates the ability of DMPVariation to generate prompts for combined concepts. In this examples, the prompts elicit the downstream model to produce images that combine the two subjects displayed on the left. This is done by sampling subject prompts using the DMPVariation model and Eq. (3). These are then fed to the SDXL model to produce the images on the right, for increasing guidance scale $\eta$. The synthesized subject clearly incorporates prominent features from both, such as the nose and chin, resulting in a cohesive blend of attributes.

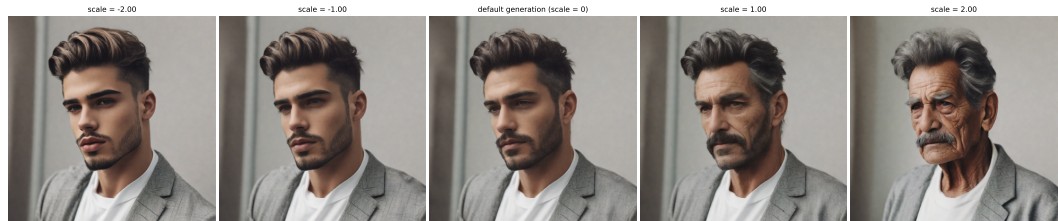

Figure 13: **Prompt Sliders** images synthesized with sliders sampled by DMPSlider when prompted for age. The prompt used for the SD-XL model is *"A photo of a beautiful man"*.

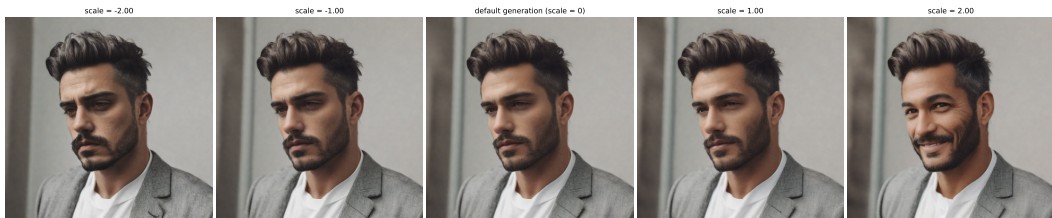

Figure 14: **Prompt Sliders** images synthesized with sliders sampled by DMPSlider when prompted for smiling. The prompt used for the SD-XL model is *"A photo of a beautiful man"*.

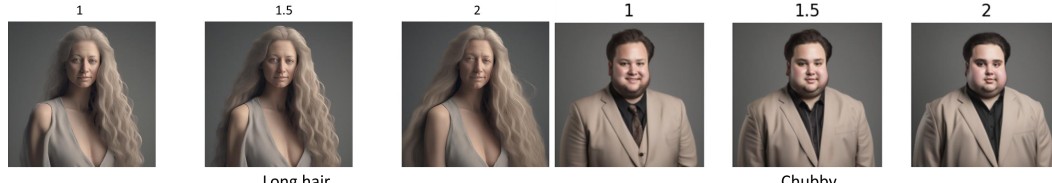

Figure 15: **Qualitative results of DMPSlider prompts** depicting the concepts "long hair" and "chubby". The prompts used for the SD-XL model for the images shown from left to right are as follows. *"A closeup photo of a person", "Professional headshot of a person"*.

**Interpreting Variation Prompts.** We computed the top-5 nearest-neighbor tokens in the CLIP vocabulary for the prompts sampled by DMPVariation model conditioned on a Textual Inversion embedding of a subject. For a random subject not in the training dataset, the TI prompt embedding returns `<w>karanjohar</w>`, `<w>conclude</w>`, `<w>leaked</w>`, `<w>prohibition</w>`, `<w>vijaysethu</w>`. The DMPVariation prompt returns `<w>karanjohar</w>`, `<w>pandoramusic</w>`, `episo`, `<w>leaked</w>`, `<w>refriger</w>`. It overlaps with two words out of five showing that the prompt is indeed a variation of the conditioned Textual Inversion prompt. Extending this analysis over 66 unseen identities, we found an average of **2.7 common words in the top-5 and 5.6 in the top-10** nearest-neighbor tokens, demonstrating that the DMP model effectively generalizes while retaining some of the subject-specific characteristics.

Table 19: **Prompts used for evaluating generalization.** Prompts were designed to explore style and concept variations of the subject `sks` and borrowed from DreamBooth (Ruiz et al., 2023).

| Prompts | | | |
|---|---|---|---|
| a sks on the beach | sks flower arrangement | sks stained glass window | sks as a witcher |
| A photo of two sks on a boat | sks Funko Pop | sks latte art | A cubism painting of sks person |
| Manga drawing of sks | Pointillism painting of sks | Ukiyo-e painting of sks | A sks as a knight in plate armor |
| sks as a knight in plate | Banksy art of sks | sks piloting a fighter jet | Greek sculpture of sks |
| Fauvism painting of sks | Cave mural depicting sks | sks by Andy Warhol | sks in the style of Archer |
| Colorful graffiti of sks | sks as Ziggy Stardust | sks in a comic book | Watercolor painting of sks |
| a sand sculpture of sks | sks in a Santa hat | sks as a wizard | a photo of sks |

## A.6 ABLATION STUDIES

### A.6.1 ABLATION STUDY ON ALTERNATIVE METHODS FOR META-PROMPTING.

The first two rows of Table 11 show the ablation study of using Transformer and GPT-2 for modeling the distribution of CoOp prompts across the 11 datasets. We include the Transformer as a non-

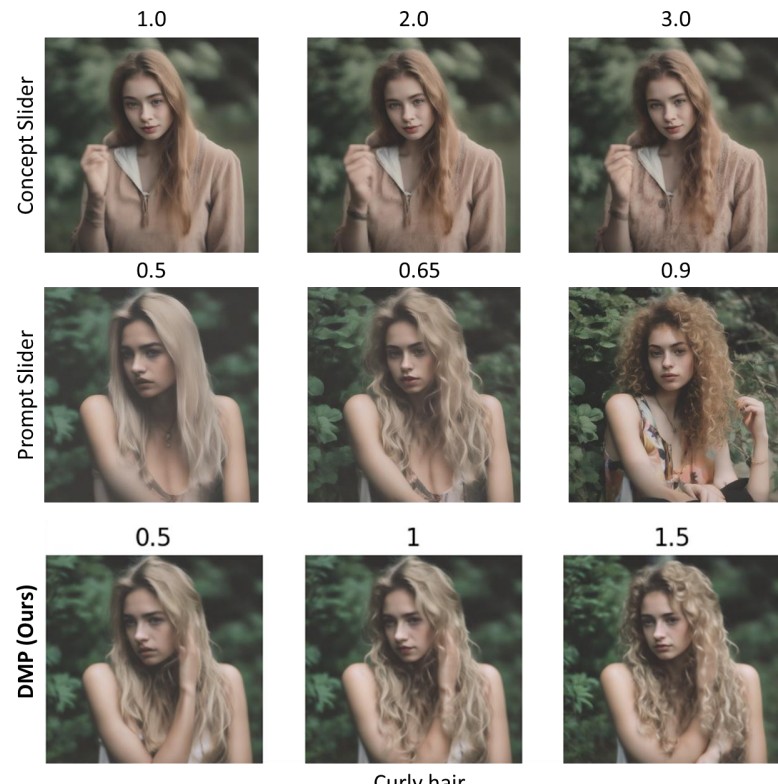

Figure 16: **Qualitative comparison** of images synthesized with baseline prompt sliders and concept sliders against DMPSlider. The prompt used for the SD-XL model is *"A photo of a girl"* for the concept **"curlyhair"**.

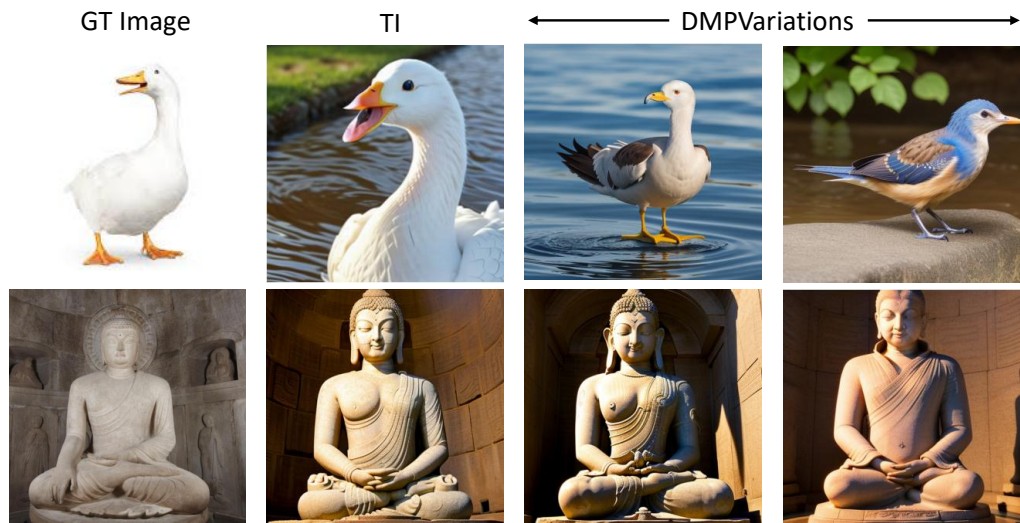

Figure 17: **Generalization of DMPVariation model**. Left: groundtruth images. Second: images generated by SD for the original TI prompts and the last two columns are the images for variation prompts sampled by DMP. See Fig. 18 for additional results.

generative baseline and GPT-2 as an autoregressive generative model. For the Transformer, we use a pretrained RoBERTa-Base model, which is finetuned using LoRA to predict prompt embeddings from text conditions. A linear layer is added on top of the final layer to produce output embeddings of the target dimension (2048 for CoOp/CoPrompt), and training is done with MSE loss. The table shows that transformer tends to overfit to the base classes as it is able to closely match the performance of the baseline CoOp on the base classes while under-performing on the novel classes leading to a decrease in the overall performance. For GPT-2, we discretize the continuous prompt embeddings by

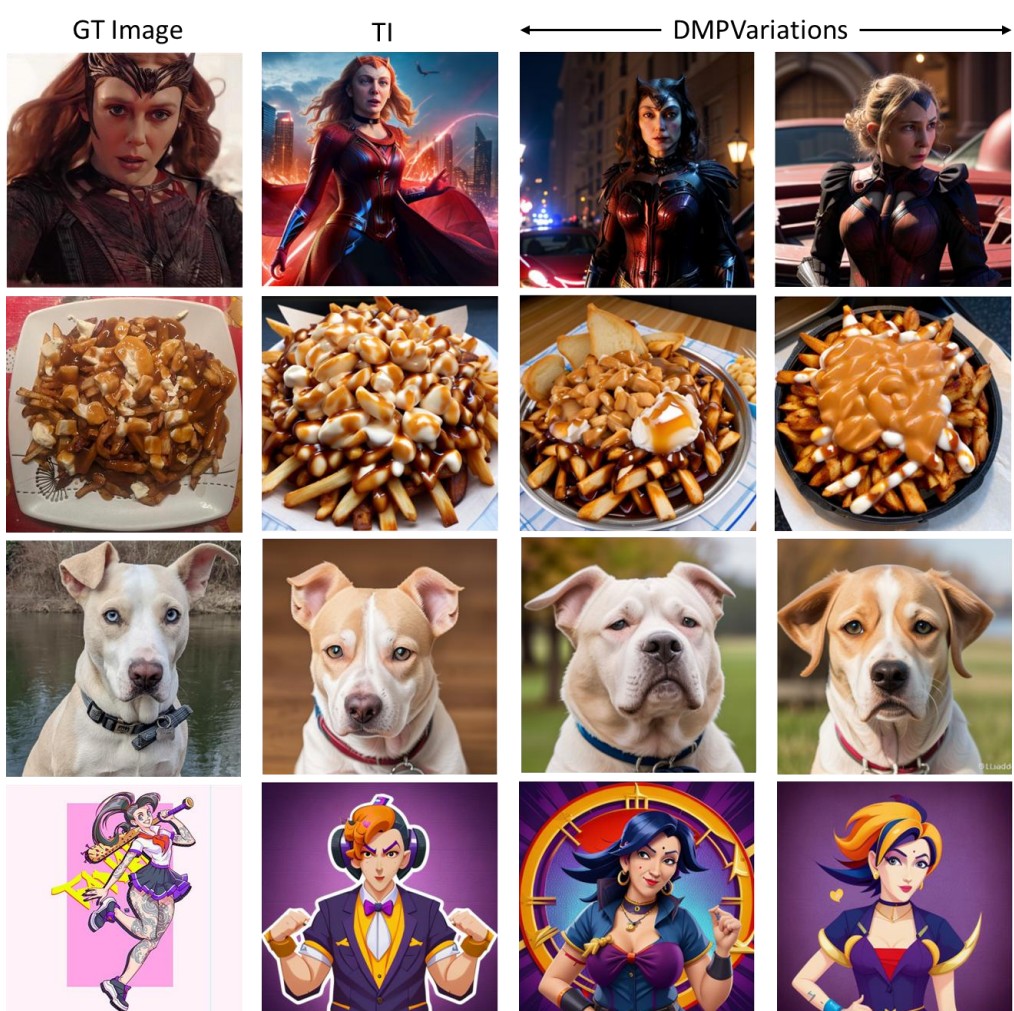

Figure 18: **Additional Qualitative results of generalization of DMPVariation model**. Left: groundtruth images. Second: images generated by SD for the original TI prompts and the last two columns are the images for variation prompts sampled by DMP.

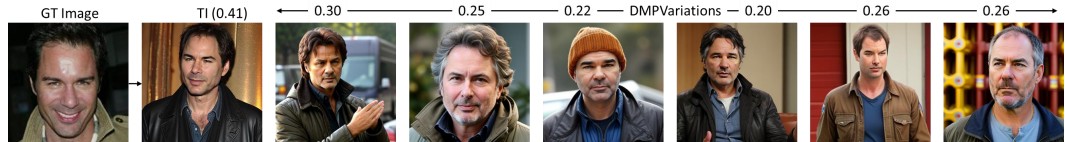

Figure 19: **DMP Variations of Textual Inversion (TI) Prompts**: The leftmost image is the real groundtruth, the second is generated by TI, and the rest are DMP variations (each image represents a new identity) conditioned on the TI embedding. All images are generated with a fixed seed to the stable diffusion model. The FaceID similarity to the groundtruth image is listed on top of each image.

identifying their top-5 nearest tokens in the CLIP embedding space. These tokens are then converted back to text and used to finetune a pretrained GPT-2 model with LoRA, trained to predict the top-5 tokens using standard cross-entropy loss. The model is conditioned on the same text inputs as used in the diffusion counterpart. Although each text condition has 40 associated prompts from different initializations (as described in Section 3.1), the resulting tokens after discretization are almost identical across seeds. This indicates that the variation captured in the continuous embedding space is lost during the discretization process-a known limitation, as discretization inherently reduces information. The table reflects this observation and shows that GPT-2 based modeling is inferior to diffusion since diffusion is much better for modeling continuous distribution of prompts. Moreover, unlike autoregressive methods, diffusion offers multiple benefits such as classifier-free guidance, negative prompting, inversion, editing and composition that are challenging or infeasible with models

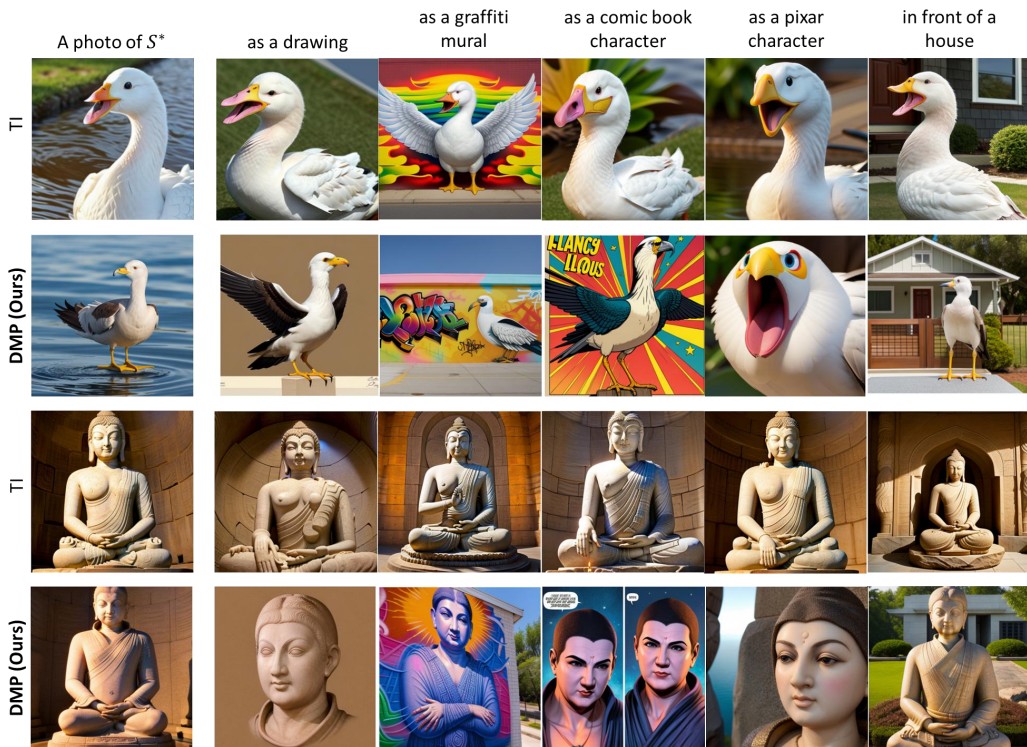

Figure 20: **Qualitative results** of images generated from prompts synthesized by DMPVariation. Our Diffusion Meta-Prompts are more robust and less overfitted than the baseline textual inversion which fails to generalize.

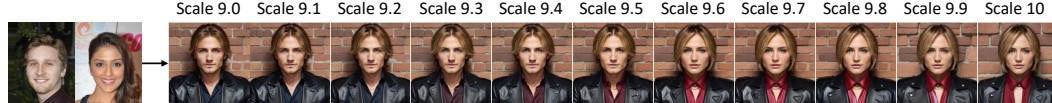

Figure 21: **Identity composition:** images generated with DMPVariation model prompts (right) for increasing guidance scales from 9 to 10 for the composition of the two identities shown on the left.

like GPT-2. These results show that modeling the prompt distribution with diffusion is more effective and flexible than using autoregressive methods.

### A.6.2  ABLATION ON THE NUMBER OF PROMPTS PER CONCEPT

We already include the ablation study when using only 20 prompts per concept for identity synthesis in Table 5 of the paper. Here, we include an additional ablation study on the number of prompts for classification tasks in Table 20. It shows that DMP works well even with as few as 5 or 10 prompts per task/concept. For 5 prompts per class, DMPCoOp obtains +1.2% gain on new classes and it increases as $n$ increases (+1.6/+2.5/+3.0 for $n =$10/20/40 prompts respectively). The results are consistent with our theoretical guarantee discussed in Appendix A.1 where higher $n$ corresponds to lower downstream risk and better generalization.

### A.6.3  ABLATION ON THE TEXT INPUTS TO DMPCoOp MODEL

We conducted an ablation study using the dataset names as the prompt or text condition to the DMPCoOp model instead of the classnames. Table 18 shows that the performance of the model using dataset names is better than the baseline CoOp prompts by 0.3% on average while it is 0.7% lower as compared to the model using classnames. This shows that using class-specific names generates prompts with robust generalization than just using a single dataset name as the text condition.

### A.6.4  ABLATION ON THE QUALITY OF TRAINING DATA

To investigate this, we assessed the impact of the quality of training data by training DMP with noisy prompts. We apply additive Gaussian noise $(n_i)$ with a standard deviation of $\sigma = 0.01$ to the $i^{\text{th}}$

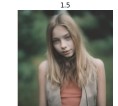 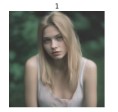 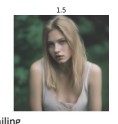 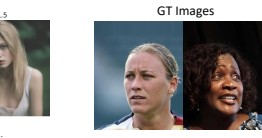 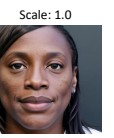 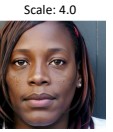 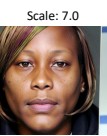

Figure 22: **Negative Prompt Sliders** images synthesized with sliders sampled by DMPSlider when negatively prompted for different concepts. The prompt used for the SD-XL model is *"A photo of a girl"*.

Figure 23: **Subject composition:** images generated with DMPVariation model prompts (right) for the composition of the two identities shown on the left with the guidance scale denoted at the top. See Fig. 21 for additional results at finer scale.

Table 20: Ablation study with different number of prompts per concept. Base2new generalization per dataset: performance for All, Base, and New classes, and HM (Harmonic Mean). The results reported are the average over three seed runs.

| Method | (a) Average All | Base | New | HM | (b) ImageNet All | Base | New | HM | (c) Caltech101 All | Base | New | HM | (d) OxfordPets All | Base | New | HM |
|---|---|---|---|---|---|---|---|---|---|---|---|---|---|---|---|---|
| CoOp | 68.8 | **82.3** | 70.4 | 75.9 | 68.5 | 76.5 | 67.2 | 71.5 | **94.8** | **98.3** | 95.0 | 96.6 | 90.1 | 94.7 | 95.9 | 95.3 |
| DMPCoOp | **70.1** | 80.3 | **73.4** | 76.5 | 68.7 | 75.4 | **68.8** | 72.0 | **94.8** | **98.3** | 95.3 | **96.8** | 91.7 | 95.4 | 97.3 | 96.3 |
| DMPCoOp (5 prompts) | 69.1 | 81.4 | 71.6 | 75.9 | 68.9 | 76.6 | 68.0 | 72.0 | **94.8** | 97.9 | 95.2 | 96.5 | 91.1 | 95.4 | 97.1 | 96.2 |
| DMPCoOp (10 prompts) | 69.3 | 81.8 | 72.0 | 76.3 | 69.0 | **76.6** | 68.5 | 72.3 | 94.5 | 98.2 | 95.2 | 96.7 | 92.9 | **96.0** | **97.5** | **96.7** |
| DMPCoOp (20 prompts) | 69.1 | 81.2 | 72.9 | **76.6** | **69.3** | 76.4 | **68.8** | **72.4** | 94.3 | 98.2 | 95.1 | 96.6 | **93.0** | **96.0** | **97.5** | **96.7** |

| Method | (e) StanfordCars All | Base | New | HM | (f) Flowers102 All | Base | New | HM | (g) Food101 All | Base | New | HM | (h) FGVC Aircraft All | Base | New | HM |
|---|---|---|---|---|---|---|---|---|---|---|---|---|---|---|---|---|
| CoOp | 68.7 | **76.7** | 68.2 | 72.2 | 74.5 | **96.7** | 68.3 | 80.1 | 84.9 | **90.0** | 89.9 | 89.9 | 25.1 | 36.9 | 27.1 | 31.2 |
| DMPCoOp | **69.3** | 74.5 | **72.0** | **73.2** | 75.6 | 93.4 | **72.2** | 81.4 | **86.4** | 89.6 | 89.9 | 89.7 | 26.3 | 35.7 | 31.6 | 33.5 |
| DMPCoOp (5 prompts) | 68.8 | 75.2 | 70.8 | 72.9 | 74.5 | 96.0 | 71.1 | 81.7 | 84.5 | 89.5 | 89.1 | 89.3 | **27.9** | **37.5** | **33.3** | **35.3** |
| DMPCoOp (10 prompts) | 68.5 | 75.8 | 69.9 | 72.7 | 73.7 | 95.7 | 69.8 | 80.7 | 85.5 | 89.8 | **91.2** | **90.5** | 27.8 | 37.0 | 32.0 | 34.3 |
| DMPCoOp (20 prompts) | 68.7 | 75.0 | 70.8 | 72.8 | 75.9 | 96.4 | 72.1 | **82.5** | 85.4 | **90.0** | 91.1 | **90.5** | **27.9** | 37.3 | 31.6 | 34.2 |

| Method | (i) SUN397 All | Base | New | HM | (j) DTD All | Base | New | HM | (k) EuroSAT All | Base | New | HM | (l) UCF101 All | Base | New | HM |
|---|---|---|---|---|---|---|---|---|---|---|---|---|---|---|---|---|
| CoOp | 67.6 | **80.8** | 72.6 | 76.5 | 51.9 | **80.8** | 51.8 | 63.1 | 63.2 | **90.8** | 72.9 | 80.9 | 67.6 | 83.4 | 65.3 | 73.2 |
| DMPCoOp | 67.3 | 80.4 | 72.9 | 76.5 | **53.7** | 75.1 | **55.8** | **64.0** | **65.9** | 85.6 | **80.7** | **83.1** | 71.7 | 79.6 | 70.7 | 74.9 |
| DMPCoOp (5 prompts) | 66.8 | 79.8 | 71.7 | 75.5 | 52.1 | 76.4 | 54.3 | 63.5 | 59.1 | 85.7 | 67.2 | 75.3 | 71.1 | **85.6** | 69.6 | 76.8 |
| DMPCoOp (10 prompts) | 67.6 | 80.3 | **73.4** | **76.7** | 52.1 | 75.6 | 53.4 | 62.6 | 60.1 | 89.0 | 71.7 | 79.4 | 70.7 | 85.3 | 69.2 | 76.4 |
| DMPCoOp (20 prompts) | 67.6 | 80.4 | 73.3 | **76.7** | 51.4 | 75.6 | 54.1 | 63.1 | 54.6 | 84.8 | 74.4 | 79.3 | **71.8** | 82.9 | **72.6** | **77.4** |

prompt $S(c)_i$ as

$$S(c)_i = (1 - \sigma)S(c)_i + \sigma n_i.$$

The noisy prompt is applied to $x\%$ of the training dataset, where $x \in \{10, 40\}$.

Table 21 summarizes the experiment where random noise is added to the prompts in the training repository. Meta-prompting is observed to be robust to noise levels ranging from 10% to 40% of the training prompts. DMPCoOp trained with 10% noisy prompts still obtains +1.9% improvement over the baseline. Further, the average H.M for the DMP model with 10% noisy prompts is slightly better (76.7 vs 76.5 for DMPCoOp) than the DMPCoOp model trained on clean prompts suggesting that a small amount of noise can also help with generalization. This is similar to image diffusion models, which are also known to be robust to noise added during training.

## A.7 ABLATION ON DMPMULTI

We trained separate DMP models for identity synthesis and slider synthesis to compare their performance with the DMPMulti model, which was trained to generate both prompt types simultaneously. Table 22 presents the results of the DMPSlider model, trained solely for slider prompt generation. Table 23 presents the results of the DMPIdentity model, trained solely for identity prompt generation. The results indicate that its performance is comparable to that of the DMPMulti model, demonstrating that multi-task training in DMPMulti does not compromise its effectiveness.

### A.7.1 ABLATION ON IDENTITY COMPOSITION WITH TEXTUAL INVERSION

For identity composition, we perform an ablation study using Stable Diffusion with Equation 3, generating new identities by combining prompts such as "a photo of id-1" and "a photo of id-2." Figure 24 illustrates the results of this process using Textual Inversion prompts with the Stable Diffusion v1.5 model. The generated images are often noisy, distorted, and tend to replicate the

Table 21: Base2new generalization per dataset: performance for All, Base, and New classes, and HM (Harmonic Mean). The results reported are the average over three seed runs.

| Method | (a) Average | | | | (b) ImageNet | | | | (c) Caltech101 | | | | (d) OxfordPets | | | |
|---|---|---|---|---|---|---|---|---|---|---|---|---|---|---|---|---|
| | All | Base | New | HM | All | Base | New | HM | All | Base | New | HM | All | Base | New | HM |
| CoOp (Zhou et al., 2022b) | 68.8 | **82.3** | 70.4 | 75.9 | 68.5 | 76.5 | 67.2 | 71.5 | 94.8 | 98.3 | 95.0 | 96.6 | 90.1 | 94.7 | 95.9 | 95.3 |
| **DMPCoOp** | **70.1** | 80.3 | **73.4** | 76.5 | 68.7 | 75.4 | 68.8 | 72.0 | **94.8** | **98.3** | **95.3** | **96.8** | 91.7 | 95.4 | 97.3 | 96.3 |
| **DMPCoOp (10% noise)** | 69.6 | 81.7 | 72.3 | **76.7** | **69.2** | 76.5 | **69.0** | **72.6** | 94.0 | 98.1 | 94.2 | 96.1 | 91.5 | 95.0 | 96.9 | 95.9 |
| **DMPCoOp (40% noise)** | 68.9 | 81.2 | 70.3 | 75.0 | 68.5 | **76.6** | 67.3 | 71.6 | 94.1 | 97.9 | 95.2 | 96.5 | **92.8** | **95.7** | **97.9** | **96.8** |

| Method | (e) StanfordCars | | | | (f) Flowers102 | | | | (g) Food101 | | | | (h) FGVC Aircraft | | | |
|---|---|---|---|---|---|---|---|---|---|---|---|---|---|---|---|---|
| | All | Base | New | HM | All | Base | New | HM | All | Base | New | HM | All | Base | New | HM |
| CoOp (Zhou et al., 2022b) | 68.7 | **76.7** | 68.2 | 72.2 | 74.5 | **96.7** | 68.3 | 80.1 | 84.9 | 90.0 | 89.9 | 89.9 | 25.1 | 36.9 | 27.1 | 31.2 |
| **DMPCoOp** | **69.3** | 74.5 | **72.0** | **73.2** | 75.6 | 93.4 | 72.2 | 81.4 | **86.4** | 89.6 | 89.9 | 89.7 | 26.3 | 35.7 | 31.6 | 33.5 |
| **DMPCoOp (10% noise)** | 69.1 | **76.7** | 69.4 | 72.9 | 77.1 | 96.4 | 72.8 | 83.0 | 85.6 | 89.9 | **91.1** | **90.5** | **27.7** | **37.6** | **33.8** | **35.6** |
| **DMPCoOp (40% noise)** | 68.3 | 76.6 | 69.1 | 72.7 | **79.4** | 95.9 | **75.0** | **84.2** | 84.6 | 89.4 | 90.3 | 89.8 | 26.4 | 36.6 | 32.8 | 34.6 |

| Method | (i) SUN397 | | | | (j) DTD | | | | (k) EuroSAT | | | | (l) UCF101 | | | |
|---|---|---|---|---|---|---|---|---|---|---|---|---|---|---|---|---|
| | All | Base | New | HM | All | Base | New | HM | All | Base | New | HM | All | Base | New | HM |
| CoOp (Zhou et al., 2022b) | **67.6** | **80.8** | 72.6 | **76.5** | 51.9 | **80.8** | 51.8 | 63.1 | 63.2 | **90.8** | 72.9 | 80.9 | 67.6 | 83.4 | 65.3 | 73.2 |
| **DMPCoOp** | 67.3 | 80.4 | **72.9** | **76.5** | **53.7** | 75.1 | **55.8** | **64.0** | **65.9** | 85.6 | **80.7** | **83.1** | 71.7 | 79.6 | 70.7 | 74.9 |
| **DMPCoOp (10% noise)** | 66.2 | 80.7 | 70.5 | 75.3 | 53.5 | 79.9 | 51.9 | 62.9 | 59.3 | 84.5 | 71.6 | 77.5 | **72.5** | **83.0** | **74.0** | **78.2** |
| **DMPCoOp (40% noise)** | 65.9 | 79.7 | 69.8 | 74.4 | 49.8 | 78.4 | 48.6 | 60.0 | 59.6 | 84.0 | 62.8 | 71.9 | 68.4 | 82.3 | 64.1 | 72.1 |

Table 22: Comparison of slider prompts generated by a separate DMP (DMPSlider) against DMPMulti.

Table 23: Comparison of separately trained DMP (DMPIdentity) with Textual Inversion and DMPMulti for identity synthesis.

| Method | CLIP-s ↑ | LPIPS↓ |
|---|---|---|
| Prompt Slider | 30.00 | 0.219 |
| DMPMulti | 29.86 | 0.126 |
| DMPSlider (separate) | 29.88 | 0.121 |

| Method (SD-RV) | Face-ID↑ | DINO ↓ | CLIP-I ↓ | CLIP-T ↑ |
|---|---|---|---|---|
| Textual Inversion | 0.428 | 0.627 | 0.696 | 0.244 |
| DMPMulti | **0.434** | **0.558** | **0.653** | 0.245 |
| DMPIdentity (separate) | **0.435** | **0.550** | 0.659 | **0.246** |

input identities rather than effectively merging their attributes. Additionally, running a full forward diffusion process with multiple identities doubles the inference time from 4 to 8 seconds per image. In contrast, our DMPMulti achieves high-quality identity compositions in approximately 5 seconds, introducing only a 1-second overhead compared to standard Stable Diffusion.

## A.8 DMP FOR PERSONALIZATION

The DMPVariation model enables the generation of prompt variations conditioned on a given Textual Inversion prompt, making it possible to train a text-to-prompt meta-diffusion model as a replacement for personalization prompt repositories. Users only require access to existing prompt repositories, as the DMPVariation model can directly generate the necessary intermediate embeddings. We choose the top-k embeddings generated from the DMPVariation model based on the cosine similarity with the available TI prompts. These embeddings can serve as training data for generating personalized prompts based on textual input. Once trained, this approach eliminates the need to search and retrieve prompts from a database, allowing for on-the-fly prompt generation.

Figure 25 presents a comparison of the images generated by DMPMulti model for the identity labeled "id-38" and "id-96" respectively. The figure presents three classes of images: groundtruth on the left, synthesized by SD prompted by the original TI prompts in the first and third row of the right side, and synthesized by SD prompted by the DMPMulti model, itself prompted for "identity-c." The images synthesized using DMPMulti model have quality comparable to those synthesized with TI prompts.

**Negative Text Guidance.** Figure 26 illustrates the effect of negative prompting by displaying images synthesized when DMPMulti model is prompted with identity-21 (leftmost) as positive and identity-24 (second from left) as negative prompt. The generated identity exhibits contrasting characteristics, such as fuller cheeks, smaller eyes, and a broader nose—features to those of the negative identity (id-24).

**Novel Identities.** Figure 27 shows additional qualitative results of novel identities sampled by DMP and their closest training images.

**Identity Prompt Diffusion.** Figure 28 presents the qualitative results of various identities sampled by DMPMulti model. The figure contains the groundtruth on the left, and synthesized by SD prompted

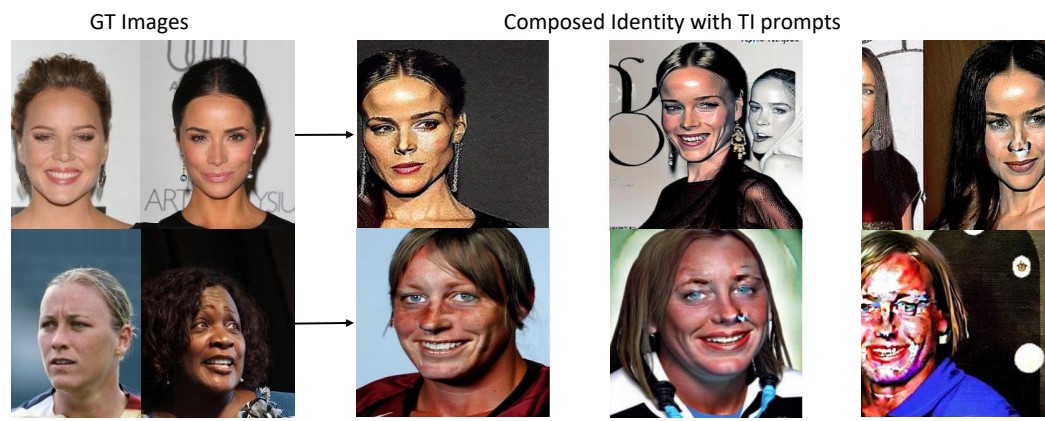

GT Images                    Composed Identity with TI prompts

Figure 24: **Ablation study on Identity composition with Textual Inversion:** images generated with TI prompts by composing the Stable Diffusion outputs for the composition of the two identities shown on the left for each row. TI composition produces distorted identities or repeats the same identities.

by the DMPMulti, itself prompted for"identity-c" on the right. The images synthesized using both models reflect the original identities in the groundtruth images.

**Identity Composition.** Figure 29 demonstrates additional results of combining two identities, displayed on the left, using DMPMulti to synthesize identity prompts with (3). The synthesized identity clearly incorporates prominent features from both original faces, such as the nose and chin, resulting in a cohesive blend of attributes.

Figure 30 shows the results of identity composition using SDv1.5 checkpoint that uses the same CLIP text encoder as SD-Realistic Vision checkpoint. It shows that DMP performs effectively without requiring retraining for this version. Since DMP was trained in CLIP text space, it generalizes to all models sharing the CLIP text encoder, eliminating the need for retraining on specific model versions.

**Interpolation.** Figure 31 shows the qualitative results of interpolating between two faces using the DMPMulti model with classifier-free guidance scale between 0 to 5. The results show that DMPMulti enables fine-grained interpolation by simply manipulating the guidance scale.

## A.9 LIMITATIONS AND FUTURE WORK

While the DMP framework unifies and improves prompt generation and generalization, simplifying deployment, the effectiveness of DMP is fundamentally constrained by the quality and expressiveness of the underlying prompt learning method used to construct the training repository. Second, DMP inherits the limitations of text prompts such as lack of fine-grained control and does not support parameter-efficient fine-tuning techniques that rely on large adapter weights, such as LoRA.

Future research directions can address these limitations by exploring joint training of the meta-model and the downstream foundation models, potentially overcoming the performance ceiling imposed by existing prompt learning techniques. Other directions for future work can explore ways for distilling LoRA (Hu et al., 2022) adapters into prompts and the design of a Meta-LoRA model, which synthesizes weight matrices instead of prompts (a more complex problem due to the large parameter cardinality of LoRA weights).

## A.10 IMPLEMENTATION DETAILS

In this section, we decribe the evaluation setup followed in our experiments and the rationale behind choosing the setup. We then describe the hyperparameter settings used to train all DMP models and finally the prompt format used in DMPMulti (DMPSlider) model.

**Evaluation Setup.** For downstream model, we use stable diffusion (Rombach et al., 2022) `Realistic-Vision-v4` checkpoint using classifier-free guidance with a scale of 4.5 and 30 DDIM steps for the image synthesis. For SD-XL (HuggingFace, 2023) model, we use a scale of 7.5 with 20 DDIM steps. Note that, because DMP prompts are introduced in the CLIP text encoder, they

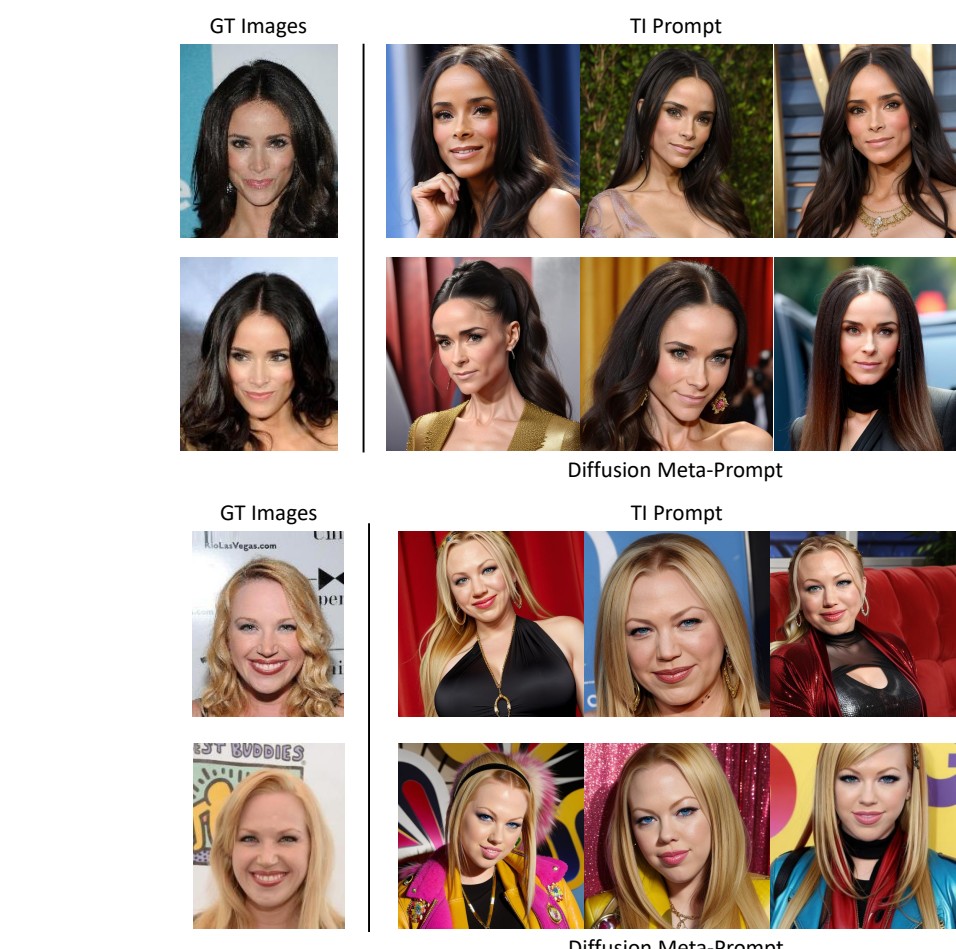

GT Images      TI Prompt

Diffusion Meta-Prompt

GT Images      TI Prompt

Diffusion Meta-Prompt

Figure 25: **Qualitative comparison** of image synthesis with TI and DMPMulti prompts. Left: groundtruth images for Identity-38 and Identity-96 in the training set. Right: images synthesized by SD for the original TI prompts (first and third row) and prompts sampled from the DMPVariation meta-diffusion model (second and fourth row):

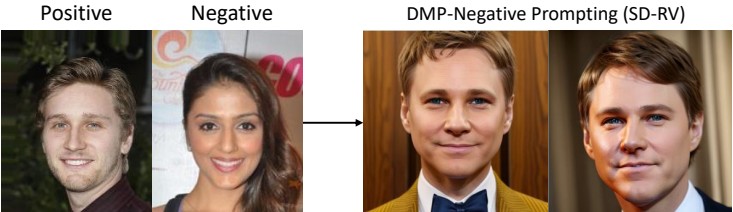

Positive    Negative      DMP-Negative Prompting (SD-RV)

Figure 26: **Negative prompting** samples from the SD model when prompted by the DMPMulti, itself prompted with id-21 (leftmost) as positive and id-24 (second from left) as negative prompt. The generated identity has features opposing to id-24 (chubby cheeks, small eyes, wide nose, etc.)

can be interchangeably used with any diffusion model using this encoder. Our choice of downstream diffusion model follows the original prompting methods.

For models other than CoOp, additional weights or head layers are optimized to prompt the deeper layers of the CLIP encoders. Due to the large number of parameters associated with these weights, it is infeasible to train a diffusion model to synthesize these parameters. For example, the projection matrices of MaPLe have 3.55 million parameters while CoPrompt and TAC have 4.65 million parameters each. This is much larger than even the images produced by Stable diffusion (65536 parameter latent). In contrast, all text prompts have only 2048 or fewer parameters (only 256 parameters in the latent space). So, we only synthesize the text prompts attached to the input of CLIP

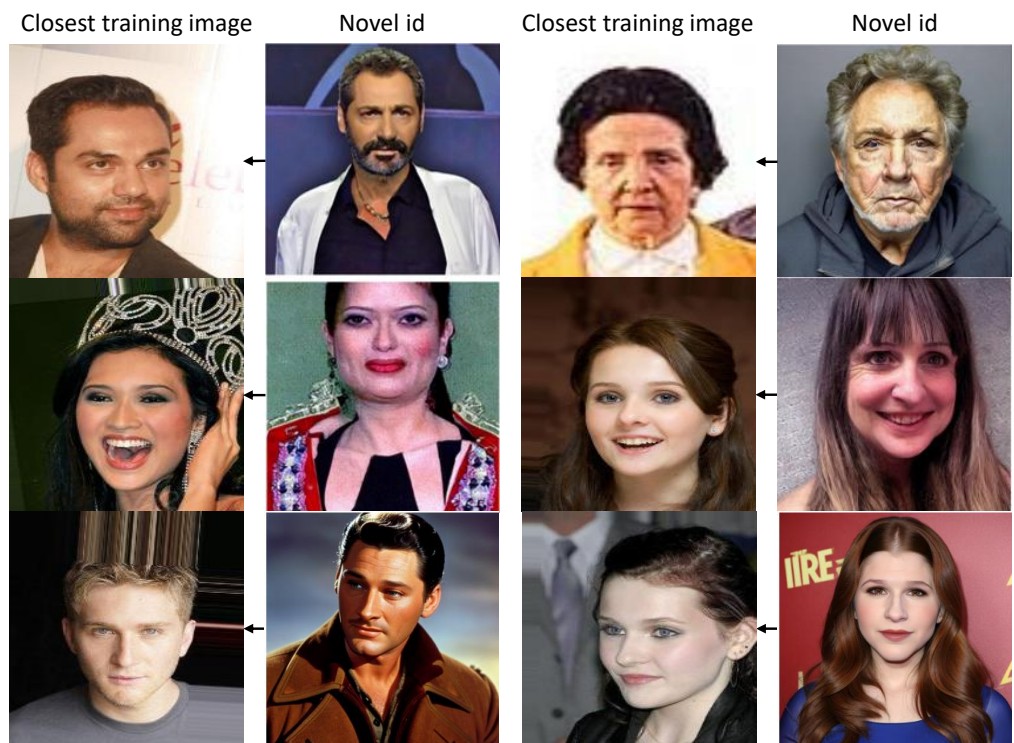

Figure 27: **Additional Qualitative results:** Nearest neighbors in the training set shown on the left for novel ids sampled by DMPMulti model to the right.

model with DMP. During inference, we replace the learned textual prompt from the baseline method with the prompt sampled with DMP while keeping the other weights of the baseline method fixed. As followed in prompt learning literature, we pick three random seeds and report the average results. We note that different initialization seeds result in minor performance differences which explains the performance difference from the original paper reported results.

**Baseline Prompt Learning Methods.** This section provides a consolidated description of all baseline prompt-learning methods and CLIP variants used in our experiments. For every method, we specify the original paper, the core algorithmic idea, the training protocol we follow, and any deviations from the original implementation. All hyperparameters, dataset splits, and CLIP backbone choices were selected to match the official implementations as closely as possible unless otherwise noted.

CoOp (Context Optimization)

We follow CoOp (Zhou et al., 2022b), which introduces learnable context tokens while keeping CLIP frozen. The method optimizes a fixed set of soft prompt vectors shared across all classes. We use the officially released configuration for few-shot learning and adopt the same CLIP backbone settings. No architectural modifications are made relative to the original paper.

CoCoOp (Conditional Context Optimization)

For CoCoOp, we follow (Zhou et al., 2022a), who extend CoOp by generating input-conditioned prompts via a lightweight meta-network. We use the authors' recommended few-shot protocol, including the same data augmentation settings and meta-network structure. The dynamic prompt generator is kept unchanged, and we adhere to the Base/New class evaluation protocol described in the original paper.

CoPrompt (Consistency-Guided Prompt Tuning)

CoPrompt (Roy & Etemad, 2024) adds a consistency regularizer and auxiliary perturbation mechanism to stabilize few-shot prompt learning. We follow the official implementation by applying the published

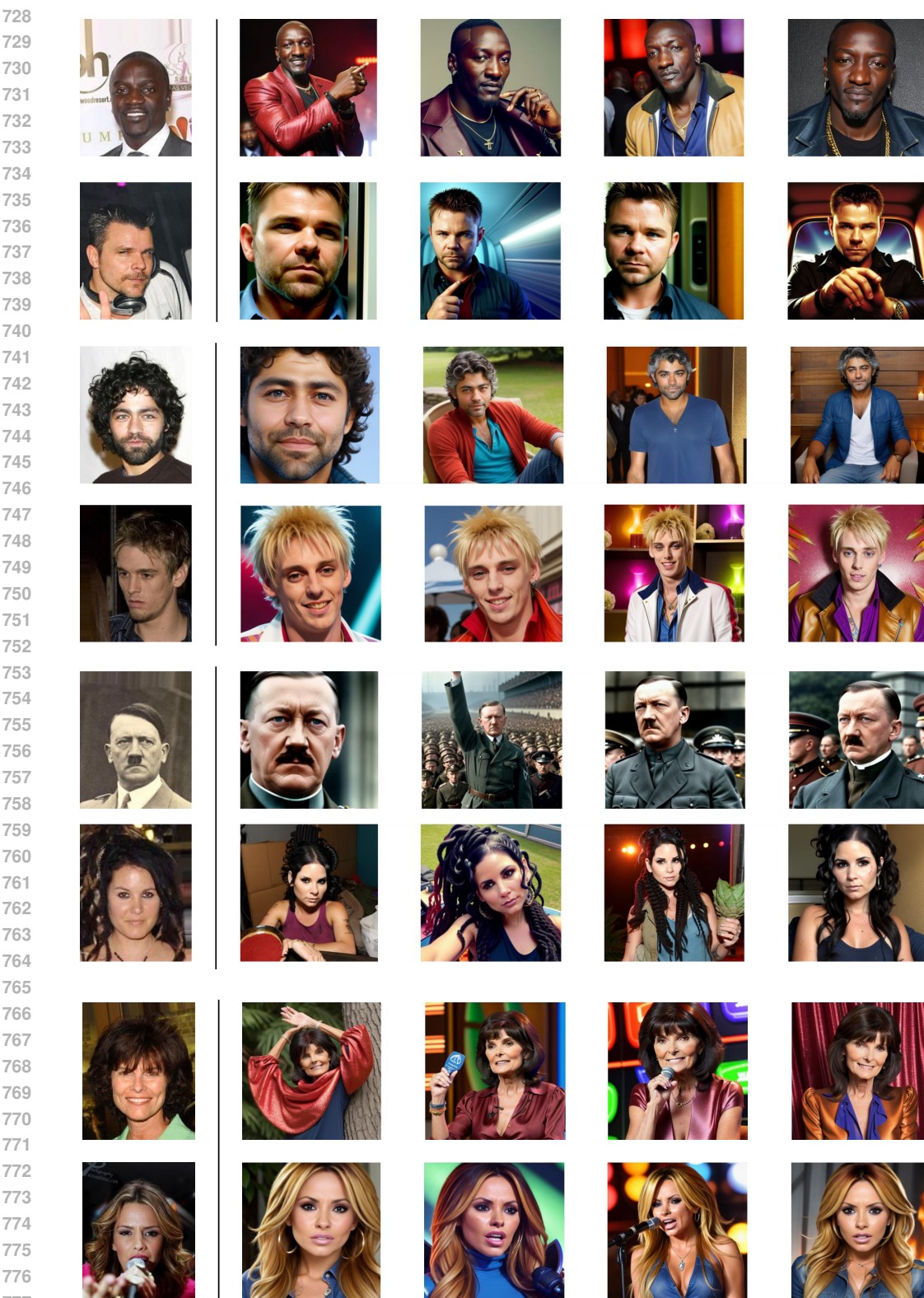

Figure 28: **Qualitative results** of image synthesis with DMPMulti prompts. Left: groundtruth images. Right: images synthesized by SD with prompts sampled from the DMPMulti model.

regularization objectives, prompt-tuning strategy, and model initialization. All perturbation and consistency components are used as defined in the original work.

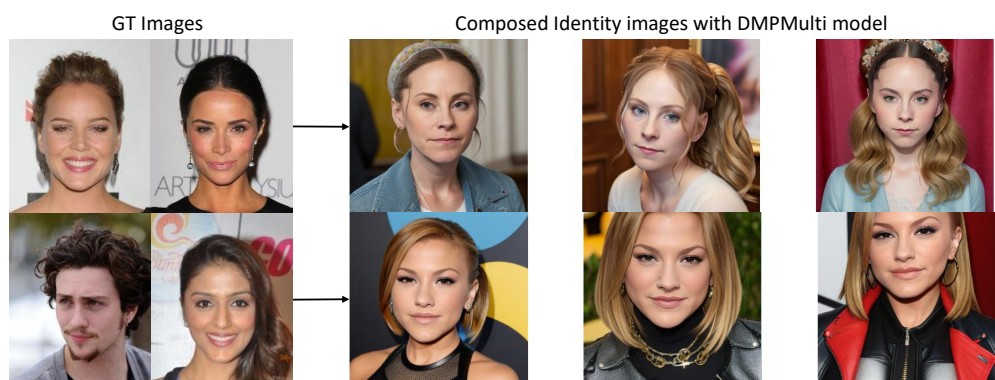

Figure 29: **DMPMulti composition:** images generated with DMPMulti model prompts (right) for the composition of the two identities shown on the left.

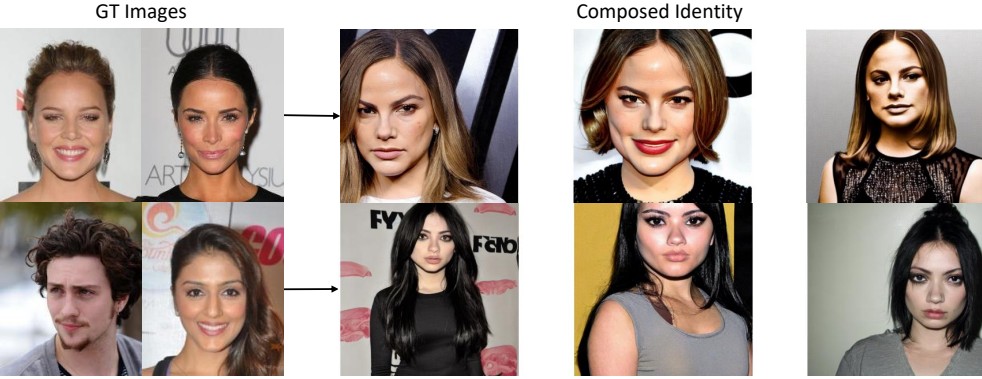

Figure 30: **DMPMulti composition with SDv1.5 model:** Prompts synthesized by DMP generalize well to different downstream models sharing the same CLIP text encoder without any re-training.

### MaPLe (Multimodal Prompt Learning)

MaPLe (khattak et al., 2023) introduces multimodal prompts applied to both the vision and text encoders, along with stage-wise prompt tokens in the vision transformer. We follow the standard MaPLe training recipe: multimodal prompts, stage-wise insertion points, and the default initialization scheme. The architecture is used without modification, and we adopt the dataset-specific training settings recommended by the authors.

### TAC (Task-Aware Prompting / Task-Aware Pre-Context)

For TAC, we follow Hao et al. (2025), who generate pre-context vectors by clustering class embeddings to encode task structure before learning the final prompts. We implement this clustering-based pre-context exactly as described and follow the same shot counts, training loops, and evaluation protocol. We make no changes to the prompt-learning architecture.

### CLIP Variants and Training Protocol

Across all baselines, we use the same CLIP backbones (e.g., ViT-B/32 or ViT-B/16) to ensure comparability. CLIP weights are frozen in all settings, consistent with the original prompt-learning papers. For each method, we match the authors' training configuration—optimizer type, learning-rate schedule, number of iterations/epochs, and batch size—based on their official releases. Where multiple public implementations exist, we default to the authors' reference repository.

#### A.10.1 HYPERPARAMETER SETTINGS

Table 25 summarizes the detailed hyperparameter settings of the DMP models trained from scratch reported in the main paper.

Figure 31: **Interpolating** between two faces using DMPMulti with concept composition.

Table 25: Hyperparameter Settings for the DMP models trained across three different tasks namely personalization, concepts and classification.

| | DMPVariation | DMPMulti | DMPCoOp | Autoencoder |
|---|---|---|---|---|
| $z$-shape | - | - | 16 x 8 | 16 x 8 |
| $x$-shape | 768 x 1 | 768 x 1 | 2048 x 1 | 2048 x 1 |
| $|Z|$ | - | - | 128 | 128 |
| $|X|$ | 768 | 768 | 2048 | 768 |
| Diffusion steps | 1000 | 1000 | 1000 | 1000 |
| Optimizer | AdamW | AdamW | AdamW | AdamW |
| Noise Schedule | linear | linear | linear | linear |
| Nparams | 33M | 33M | 33M | 1M |
| Channels | 128 | 128 | 128 | 128 |
| Depth | 1 | 1 | 1 | 1 |
| Channel Multiplier | 1,2,2,2 | 1,2,2,2 | 1,2,2,2 | 1,1,1,1,2,2,2,2 |
| Attention resolutions | 16, 8, 4 | 16, 8, 4 | 16, 8, 4 | - |
| Head Channels | 8 | 8 | 8 | 8 |
| Batch Size | 320 | 320 | 256 | 128 |
| Iterations | 100k | 100k | 100k | 100k |
| Learning Rate | 1e-6 | 1e-6 | 2.0e-7 | 4.5e-6 |

### A.10.2  EVALUATION PROMPTS FOR DMPSLIDER

The example format of the prompts used for evaluation is shown below for the concept "Age",

- A portrait of a woman with a warm smile, {}

- A person's face, {}

- A man sitting on a park bench, reminiscing about his youth, {}

- A couple of friends enjoying a picnic together, {}

- A photo of a person, {}

**The full list of prompts used for evaluation will be made public along with the code and trained models.**

### A.10.3  CODE AND TRAINED MODELS

**Code is attached in the supplementary material. Code and trained models will be released publicly upon acceptance of the paper.**

### A.11  BACKGROUND

Here, we discuss the background on prompt tuning approaches used in CoOp/CoPrompt (Roy & Etemad, 2024) methods. We learn these CoOp/CoPrompt prompts to train the DMPCoOp/CoPrompt model, which unifies the synthesis of prompts for multiple downstream classifiers. **Prompt Tuning.** Foundation visual-language models like CLIP (Radford et al., 2021) are trained with contrastive learning and a large dataset of image-text pairs to align image-text representations in a shared semantic space, created by an image $f_I$ and a text $g_T$ encoder. After pre-training, open-set zero-shot image recognition is implemented by specifying class names with a pre-defined prompt template (e.g. $T_i =$, "a photo of a [CLASS]$_i$") and determining the class $i$ whose text feature $g_T(T_i)$ has maximum cosine similarity with image feature $f_I(I)$. While powerful, this zero-shot classifier implementation frequently fails to match the performance of classifiers trained for specific class sets.

Prompt-tuning methods bridge this gap by learning soft-prompts from a few samples, to improve the performance of the foundation model. CoOp (Zhou et al., 2022b) introduces and refines a set of $M$ continuous context vectors $V = \{v_1, v_2, \ldots, v_M\}$ as the learnable prompt. The prompt $T_i = \{v_1, v_2, \ldots, v_M, c_i\}$ concatenates these vectors and the class token embedding $c_i$. CoOp learns

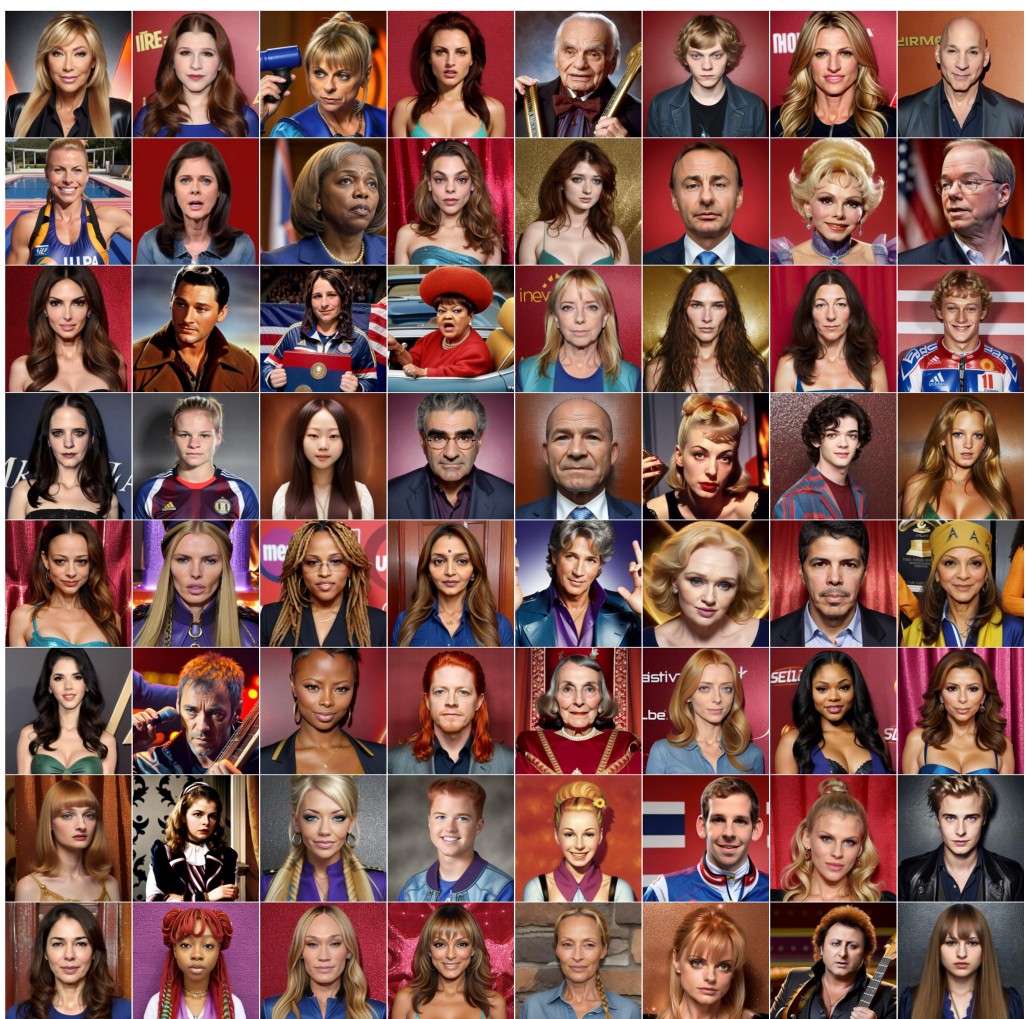

Figure 32: **Qualitative results of generating novel identities during inference** using random identity conditioning with DMPMulti model. The prompt used for the Stable Diffusion Realistic Vision model is *"A photo of a id-x"* where x is the id not present in the training set. The identities do not overfit (**mean face ID similarity of 0.0102 across training images**). Note that all the images use a fixed seed to the diffusion model.

the static context vectors $V$ by minimizing the negative log-likelihood of the correct class token

$$L_{CE}(V) = -\sum_i y_i \log p(T_i|I), \tag{11}$$

where $y_i$ is the one-hot ground-truth label for class $i$. Since the foundation model parameters are frozen, the learnable prompt $V$ can be efficiently optimized with few training samples.

## A.12 BROADER IMPACT

We introduce a new meta-learning framework for generating prompts for foundation models. While it offers the benefits of storage and runtime efficiency, it uses existing pretrained models which are shown to contain harmful biases that maybe elicited by the prompts, it can also be potentially misused to propagate harmful, unlawful or unethical information with the personalization of celebrities. Since, the framework is meta-learning, any harmful prompts can be identified before the image generation step where the embeddings can be inspected with nearest neighbor tokens in the text space. Additionally, recent advancements in image watermarking (Luo et al., 2022) can help to identify generated image contents to protect against these risks.

## A.13 LLM USAGE

LLM was used to polish the writing (e.g., clarity, grammar). It was not used in any other stage.

| Concept | DMP | TI | Delta |
|---|---|---|---|
| ettblackteapot | 0.2905 | 0.2778 | +0.0127 |
| dicoo | 0.2837 | 0.2700 | +0.0137 |
| goku | 0.2830 | 0.2800 | +0.0030 |
| degodsheavy | 0.2827 | 0.2754 | +0.0073 |
| spider-gwen | 0.2825 | 0.2810 | +0.0015 |
| johnny-silverhand | 0.2812 | 0.2760 | +0.0052 |
| blue-haired-boy | 0.2812 | 0.2800 | +0.0012 |
| bullvbear | 0.2798 | 0.2683 | +0.0115 |
| black-waifu | 0.2790 | 0.2715 | +0.0075 |
| a-female-hero-from-the-legend-of-mir | 0.2788 | 0.2710 | +0.0078 |
| freddy-fazbear | 0.2786 | 0.2750 | +0.0036 |
| ldrs | 0.2783 | 0.2642 | +0.0141 |
| chonkfrog | 0.2778 | 0.2756 | +0.0022 |
| concept-art | 0.2770 | 0.2715 | +0.0055 |
| degods | 0.2769 | 0.2686 | +0.0083 |
| hanfu-anime-style | 0.2766 | 0.2703 | +0.0063 |
| joemad | 0.2764 | 0.2673 | +0.0091 |
| dog | 0.2761 | 0.2780 | -0.0019 |
| stuffed-penguin-toy | 0.2760 | 0.2756 | +0.0004 |
| fox-purple | 0.2760 | 0.2734 | +0.0026 |
| colossus | 0.2756 | 0.2634 | +0.0122 |
| chungus-poodl-pet | 0.2754 | 0.2637 | +0.0117 |
| anya-forger | 0.2754 | 0.2737 | +0.0017 |
| furrpopasthetic | 0.2751 | 0.2737 | +0.0014 |
| bob-dobbs | 0.2751 | 0.2664 | +0.0087 |
| eddie | 0.2751 | 0.2607 | +0.0144 |
| arthur1 | 0.2750 | 0.2666 | +0.0084 |
| dragonborn | 0.2750 | 0.2556 | +0.0194 |
| kay | 0.2747 | 0.2605 | +0.0142 |
| tesla-bot | 0.2747 | 0.2634 | +0.0113 |
| borderlands | 0.2747 | 0.2637 | +0.0110 |
| lavko | 0.2744 | 0.2588 | +0.0156 |
| gim | 0.2740 | 0.2659 | +0.0081 |
| hubris-oshri | 0.2740 | 0.2659 | +0.0081 |
| tubby | 0.2737 | 0.2673 | +0.0064 |
| finn-token | 0.2737 | 0.2734 | +0.0003 |
| moxxi | 0.2730 | 0.2722 | +0.0008 |
| altvent | 0.2730 | 0.2676 | +0.0054 |
| omlettehaai | 0.2730 | 0.2600 | +0.0130 |
| jos-de-kat | 0.2730 | 0.2651 | +0.0079 |
| loab-style | 0.2730 | 0.2573 | +0.0157 |
| crinos-form-garou | 0.2727 | 0.2693 | +0.0034 |
| blue-zombie | 0.2727 | 0.2693 | +0.0034 |
| cgdonny1 | 0.2725 | 0.2610 | +0.0115 |
| amogus | 0.2725 | 0.2551 | +0.0174 |
| lucky-luke | 0.2725 | 0.2751 | -0.0026 |
| captain-haddock | 0.2725 | 0.2725 | +0.0000 |
| manga-nov-23 | 0.2725 | 0.2551 | +0.0174 |
| button-eyes | 0.2722 | 0.2683 | +0.0039 |
| bruma | 0.2722 | 0.2632 | +0.0090 |
| ouroboros | 0.2720 | 0.2734 | -0.0014 |
| fursona | 0.2720 | 0.2646 | +0.0074 |
| kanovt | 0.2720 | 0.2522 | +0.0198 |
| doc | 0.2717 | 0.2693 | +0.0024 |
| warhammer-40k-drawing-style | 0.2715 | 0.2727 | -0.0012 |
| nard-style | 0.2715 | 0.2698 | +0.0017 |
| baluchitherian | 0.2715 | 0.2617 | +0.0098 |
| insidewhale | 0.2715 | 0.2630 | +0.0085 |
| irasutoya | 0.2715 | 0.2598 | +0.0117 |
| devonm | 0.2712 | 0.2617 | +0.0095 |
| edgerunners-style-v2 | 0.2712 | 0.2660 | +0.0052 |
| nixeu | 0.2710 | 0.2660 | +0.0050 |
| shek-9-12-opening | 0.2710 | 0.2708 | +0.0002 |
| loab-character | 0.2710 | 0.2656 | +0.0054 |
| ldr | 0.2710 | 0.2666 | +0.0044 |
| malika-favre-art-style | 0.2710 | 0.2632 | +0.0078 |
| drive-scorpion-jacket | 0.2710 | 0.2686 | +0.0024 |
| apulian-rooster-v0-1 | 0.2710 | 0.2351 | +0.0359 |
| fftstyle | 0.2708 | 0.2593 | +0.0115 |
| alf | 0.2708 | 0.2664 | +0.0044 |
| wheelchair | 0.2708 | 0.2730 | -0.0022 |
| obama-self-2 | 0.2705 | 0.2705 | +0.0000 |
| dog-chip | 0.2703 | 0.2676 | +0.0027 |
| cheburashka | 0.2700 | 0.2705 | -0.0005 |
| ihylc | 0.2700 | 0.2580 | +0.0120 |
| cat-toy | 0.2695 | 0.2600 | +0.0095 |
| **Average** | 0.2731 | 0.2663 | +0.0068 |

Table 24: HPSv2 comparison for 75 random concepts where each concept is evaluated for 6 different prompts.

