# OpenReview forum: "Diffusion Meta-Prompts for Foundation Models"
_ICLR.cc/2026/Conference — Submitted to ICLR 2026_

### Official Review · Reviewer_fRKq · 2025-10-27

**Soundness:** 3
**Presentation:** 3
**Contribution:** 2
**Rating:** 4
**Confidence:** 3

**Summary:**

In this work, the authors propose Diffusion Meta-Prompts (DMP), a diffusion-based framework that models the distribution of learned prompts to generate task-conditioned prompts for foundation models, aiming to enhance generalization and efficiency across diverse tasks. Instead of fine-tuning model weights or training task-specific adapters, DMP learns a set of meta prompts in the latent or textual embedding space that guide the diffusion process during sampling.

**Strengths:**

1. The paper is well-organized and easy to follow.

2. The proposed DMP framework is presented as a general approach applicable to multiple modalities and tasks, including classification, personalization, and image synthesis, which demonstrates versatility.

3. The code is attached, making the method reproducible.

**Weaknesses:**

1. The idea of modeling prompt distributions via generative models is not truly new. Prior works [1-3] have already explored meta-learning or generative approaches for prompts. The paper’s contribution over these is unclear and incremental.

2. The paper’s related work and experiments fail to include comparisons with existing generative or meta-learning–based prompt tuning methods, such as [1–3]. limiting the evaluation of the proposed DMP framework against relevant state-of-the-art approaches.

3. The initialization strategy of the prompt repository is not clearly described.

4. The paper does not include an ablation study on the number of prompts stored in the prompt repository. It remains unclear how sensitive the method is to this design choice or how to balance generation quality against computational cost.

5. Although the paper includes a discussion of limitations, it does not present concrete failure cases or qualitative examples, making it difficult to understand the method’s robustness boundaries and potential failure modes.

[1] Prompt Learning via Meta-Regularization. Park, J. et al., CVPR 2024.
[2] Patch-Prompt Aligned Bayesian Prompt Tuning for Vision-Language Models. Liu, X. et al., UAI 2024.
[3] Gradient-regulated meta-prompt learning for generalizable vision-language models. Wang, J. et al., CVPR 2023.

**Questions:**

1. Proposition 1 assumes that the repository contains *n* i.i.d. prompts for each condition *y*. Given that prompts are typically correlated or template-based, how realistic is this assumption, and how would the theoretical guarantee change if the prompts are not truly i.i.d.?

2. In Figure 9, the prompts generated by DMP appear to form clustered patterns in the embedding space. Could the authors explain what these clusters represent?

---

> ### Author Response · Authors · 2025-11-27
>
> We thank the reviewer for their time and immensely thoughtful comments to improve the paper. We appreciate the positive comments on our proposed method such as versatility across tasks, paper is well-organized and on reproducibility.
>
> **1) Novelty**
>
> - We thank you the reviewer for these pointers. Please see the \textbf{general comment}. These approaches are somewhere in between prompt learning methods and DMP. They essentially perform prompt learning over a set of tasks, using ideas like meta learning, where an inner loop performs prompt learning for a dataset or task and an outer loop learns some meta-parameters that allow the transfer of information between tasks. Some aspects of these methods are like DMP, e.g the fact that at the end  of the process the meta learning process can perform all tasks.
>
> - On the other hand there are several important differences. The first is scalability. These methods still learn using the image data from all tasks, which makes them much more compute intensive. For example, training over the 11 datasets considered in this paper requires more than 100 GB VRAM in GPU memory for the bayesian approach which is not possible on a single GPU and requires multiple GPUs or nodes. The three step procedure of Figure 3 is key to avoiding this complexity. The prompts are learned per task, e.g. using one dataset at a time and any of prompt learning method, and the $\textit{global}$ training of the DMP model only requires the prompt repository, which is orders of magnitude smaller than the image repository of the associated tasks. This decomposes the complexity of learning the prompt distribution into a sequence of low complexity training steps. The approaches cited by the reviewer lack this property, which makes them much less scalable in the number of tasks.
>
> - The second is that these techniques $\textit{do not address the problem targeted by this paper}.$
> DMP is meant to address the problem that repositories of prompts learned by prompt learning methods $\textit{already exist}$ but are cumbersome to use and do not generalize to other tasks. The fundamental question is how a model learned from such a repository performs both on the datasets on which the prompts are learned $\textit{and}$ beyond those datasets. The references cited by the reviewer are simply not applicable to this problem, since the images of the original problems $\textit{are not even available}$, only the prompt repository. In fact, they can't even be easily modified to learn from prompts, since the bulk of these methods is the learning of the prompts themselves.
>
>  - Third, these methods have no ability to generalize in the way that DMP does. They cannot replicate the ability of DMP to enable one-shot composition, and negative prompts without per-task optimization (see sections 4.2 and 4.3 of the paper), which are properties specific to the use of diffusion models. In fact, these methods cannot even be applied to  tasks like personalization of diffusion models, or any other diffusion model tasks, without major modifications, since they require task-specific losses and training. It is not even clear that a meta-learning algorithm that performs well over a set of classification tasks trained with cross-entropy (as these methods do), will still work when applied over a set of image synthesis tasks trained with diffusion losses. The three step decomposition of DMP avoids this problem, since the diffusion model is simply tasked with learning a distribution of already trained prompts, $\textit{independently of what image data, losses, and training methods were used to learn them}.$
>
> - Finally, the methods cited by the reviewer have no ability to support the generalization to new tasks shown in the right side of Figure 3. This generalization requires a powerful model of the prompt distributions, capable of learning the structure of the prompt space and able to sample new prompts. Simply learning some meta-parameters for information transfer across tasks is not going to accomplish this.

---

> > ### Author Response · Authors · 2025-11-27
> >
> > **2) Comparisons with prior methods**
> >
> > - From the previous comment, it should be clear that these methods are not comparable. For example, it is not even clear how the methods cited by the reviewer could be used to generate prompts for a diffusion model.  These methods are the state of the art for a narrowly defined set of tasks that does not even begin to approach the generality of those covered by DMP. DMP should not be expected to beat these methods on the specific tasks where they are trained. It is not surprising that a task specific model will beat a generalist method, like DMP, on the specialized datasets where it was trained. For example, ImageNet trained CNNs still beat CLIP on ImageNet. This does not detract from the fact that CLIP is a much more important model than the state-of-the-art ImageNet CNNs.  Similarly, these methods would not be expected to generalized beyond the datasets where they are trained. Rather than an extensive comparison to all these methods, we believe that it is much more important to show the ability of DMP to generate good prompts for the completely different tasks of diffusion model personalization, controlling concept generation with prompt sliders, negative prompting, and all other applications demonstrated on Figures 4-8 of the paper and throughout the appendix. Furthermore, as stated in the general comment, for methods that require access to data (e.g., meta-regularization approaches that learn from raw images or produce fine-tuned weights), direct performance comparisons are not strictly fair because those methods exploit image/data signals that DMP purposely does not use. The paper’s motivation is precisely to avoid searching large prompt/LoRA repositories or re-training on images: DMP learns from prompts only, not images.
> >
> > - We do compare DMP against multiple recent prompting methods by training DMP on their learned prompts and evaluating the synthesized prompts in the downstream tasks (CoOp, CoCoOp, MaPLe, CoPrompt, TAC). See Sec. 4 and Table 11 (appendix lists full results).  These comparisons are just to check that DMP $\textit{retains competitive performance}$ despite operating under a far more challenging and general setting. We show that, in many cases, DMP sampled prompts even outperform the prompt learning prompts on which DMP is trained. For example, the classification results of Table 4 in the paper show that **DMP outperforms the base method on the unseen classes of the very same dataset, despite never accessing a single image**. We show that DMP generalizes better than these methods for novel tasks (Figure 5), cross-domain tasks (Table 3, Figure 4) and in variations of the specific tasks (Table 7) on which they are trained. We believe that this sufficient. An exhaustive comparison to all variants of prompt learning is beyond the scope of the paper. After all, DMP could be applied to the prompts learned by any prompt learning method, including meta-learning procedures.
> >
> > **3) The initialization strategy of the prompt repository is not clearly described.**
> >
> > - For classification prompts we use the baseline prompt learning algorithms (CoOp,TAC etc.) following the hyperparameters specified in the respective papers, and save the resulting prompt embeddings. For each dataset we collect 40 different initializations (random seeds) and save the prompt embeddings to form the training tensor.
> >
> > - For textual-inversion variation repositories (faces) we save 40 variations per identity (40 steps sampled from the TI optimization between 200-400 iterations, every 5 steps).
> >
> > - This 40-seed design explicitly increases prompt diversity and is the basis for DMP training; the choice is empirical and robust (see ablations in Table 20). These exact details are in Sections 3.1,3.2,3.3 and the appendix A.10.
> >
> > **4)  ablation study on the number of prompts stored in the prompt repository**
> >
> > - Good point. We included the result for 20 prompts per concept for identity synthesis in Table 5 of the paper. We now show  an additional ablation study on the number of prompts for classification tasks in Table 20. It shows that DMP works well with as few as 5 or 10 prompts per task/concept. **For 5 prompts per class, DMPCoOp obtains +1.2\% gain on new classes over the  baseline. This gain increases with the number of  prompts $n$ (+1.6/+2.5/+3.0 for $n=$10/20/40 prompts respectively).** The results are consistent with our theoretical guarantee discussed in Appendix A.1 where higher $n$ corresponds to lower downstream risk and better generalization.

---

> ### Author Response · Authors · 2025-11-27
>
> **5) Failure cases**
>
> - As with image diffusion models, sampling with DMP is sensitive to the noise seed and its performance varies slightly depending on the seed. However, in our experiments, DMP maintained high accuracy even at different noise seeds with **only a standard deviation of 0.7 % from the mean accuracy**. In contrast, the standard deviation of the baseline prompt learning methods used to build the repository from different initialization seeds show a higher variation of 4.7\% .
>
> **6) How would the theoretical guarantee change if the prompts are not truly i.i.d.?**
>
> - Good point. Although it may be argued that prompts in a repository are often correlated or template-based, this characterization does not accurately apply to our setting. The prompts used to train DMP are not human-authored natural-language templates, but rather learned prompt embeddings that arise from independent optimization trajectories for each concept with different seeds. Empirically, these embeddings exhibit high inter-class variability and do not form the repeating template structures that typically induce strong dependence. Moreover, the only point in our analysis where independence is invoked is the concentration step via Hoeffding's inequality. For this step, strict independence is not required: weak dependence is sufficient and still yields the same $O(1/\sqrt{n})$ rate, effectively replacing $n$ with an effective sample size $n_{\mathrm{eff}} \approx n$ in practice. Finally, samples drawn from the diffusion model are conditionally independent by construction, regardless of dependencies that may exist within the training set. Therefore, the i.i.d. assumption in Proposition 1 is a standard and reasonable approximation in the context of learned prompt repositories and does not materially affect the validity of the theoretical guarantee.
>
> - Practically, we mitigate correlation by collecting prompts from 40 different initializations (per dataset/condition) and by saving prompts across multiple optimization steps (TI case). We also empirically show robustness to noisy or partially corrupted repositories (A.6.4)
>
> **7) In Figure 9, the prompts generated by DMP appear to form clustered patterns in the embedding space. Could the authors explain what these clusters represent?**
>
> - Figure 10’s t-SNE visualizations (in the revised version) show real prompt embeddings (from the repository) vs DMP-generated prompts (multiple noise seeds). The clusters show that the prompts sampled by DMP are more **consistent** than those originally learned by the prompt learning method. This could explain the improved generalization of these prompts and the lower variance of the performance of the prompted model discussed above.

---

### Official Review · Reviewer_jZN5 · 2025-10-30

**Soundness:** 3
**Presentation:** 2
**Contribution:** 2
**Rating:** 4
**Confidence:** 4

**Summary:**

This work introduces a diffusion-based approach to model the distribution of learned foundation models prompts. The approach generates
prompts conditioned on text or prompt embeddings, and is suitable for both VLM and image generation models. The method is evaluated across a plethora of tasks, showing improvements compared to the baseline.

**Strengths:**

- Straightforward but interesting approach for prompting
- Evaluation for multiple type of tasks
- Generally, good improvements against the baselines and included approaches
- The included theoretical backing is a plus

**Weaknesses:**

- Novelty wise, what are the conceptual difference compared to similar works that perform diffusion for prompting: "Prompt Diffusion Robustifies Any-Modality Prompt Learning, 2024", "Diff-prompt: Diffusion-driven prompt generator with mask supervision, 2025" etc ?
- Many of the comparison made are against CoOp, which while a pioneering work, its performance is now surpassed by many other approaches.  How would the results from Table 3,7,Fig. 3,4 etc on top of TAC?
- The spacing across the paper is aggressively adjusted, making it harder to read (ex: Fig. 3-5 and Tab. 5-7 are nearly overlaping and and conjoined with the rest to the text, same on top of section 4 etc).
- Missing comparisons with state-of-the-art: [A,B,C,D,e etc], with some of these methods significantly surpassing the present baseline either on a dataset level, or globally (e.g: [D]).
- Similarly, for diffusion models (e.g: [F])

[A] Hierarchical Variational Test-Time Prompt Generation for Zero-Shot Generalization, Wu et al, CVPR 2025
[B] LASP: Text-to-Text Optimization for Language-Aware Soft Prompting of Vision & Language Models, Bulat et al, CVPR 2023
[C] Consistency-guided prompt learning for vision-language models, Roy et al, ICLR 2024
[D] PromptKD: Unsupervised Prompt Distillation for Vision-Language Models, Li et al, CVPR 2024
[E] DPC: Dual-Prompt Collaboration for Tuning Vision-Language Models, Li et al, CVPR 2025
[F] Contrastive Test-Time Composition of Multiple LoRA Models for Image Generation, Meral et al, CVPR 2025

**Questions:**

- What is the training of the presented method vs prior work (e.g. CoOp, TAC etc) ?
- The visual results quality looks somewhat poor, perhaps because the diffusion model used? are there any combination with better generators to understand how well it works?

---

> ### Author Response · Authors · 2025-11-27
>
> We thank the reviewer for their time and insightful comments to improve the paper. We appreciate the positive comments on our proposed method on extensive evaluation, good improvements and theoretical guarantee.
>
> **1) Novelty**
>
> - Please refer to the $\textbf{general comment}$ for a discussion on the novelty of the proposed method over prior approaches. Figure 3 and Table 9 provides the conceptual differences between prior meta-learning or prompt tuning methods and DMP.
>
> As should be clear from Figure 3, the setting of DMP (center/right side of the figure) is fundamentally different from that of the methods referred by the reviewer, which are prompt learning methods (left side). As such, these methods learn prompts for a single example or dataset, $\textit{using data}$ from that dataset and $\textit{task specific}$ losses. DMP addresses the problem of how to deal with the $\textit{outcomes of these methods}$, i.e. the prompts they produce, over a large set of tasks. It is meant to address the problem that repositories of such prompts $\textit{already exist}$ but are cumbersome to use and do not generalize to other tasks. The fundamental question is how a model learned from such a repository performs both on the datasets on which the prompts are learned $\textit{and}$ beyond those datasets. This question is discussed in Fig 4-8, and sections 4.2 and 4.3 of the paper, where it is shown that this generalization enables one-shot sampling, composition, and negative prompts without per-task optimization. Note that, for  prompt learning methods, the question of how they perform $\textit{beyond}$ the dataset where they are trained does not even make sense. The prompts learned by these methods have some ability to generalize in the neighborhood of the regime (dataset and task) where they are learned, e.g. unseen classes of the same dataset, but even there they already underperform DMP. Tables 8 and 10 show that they have little ability to generalize even  across pairs of datasets of similar tasks. We provide a theoretical guarantee and comprehensive empirical results across multiple prompt families and tasks demonstrating that DMP has this generalization. Note also that DMP does not require access to the original data used to train the prompt learning methods. This is critical to ensure scalability of learning in the number of tasks.   The  methods cited by the reviewer lack this property. They must be trained one-task-at a time.
>
> **2) Comparison with TAC models**
>
> Please see the new results section in the general comment. We provide the additional results of dataset combination pairs for TAC and DMPTAC in Table 10, cross-dataset generalization in Table 15, cross-domain in Table 16, cross-task generalization in Table 17. For the simpler problem of domain generalization on the simple classification task (ImageNet to ImageNet-V2/S/A/R) DMPTAC matches or slightly surpasses TAC. However, in the more challenging problem of domain and task generalization, where the model has to perform hierarchical classification across various domains, DMPTAC yields substantial gains **(e.g., ImageNet-S: +9.0%, SUN397: +10.0%)**. These results collectively show that DMP is not merely “dependent” on the base prompts and it actively generalizes beyond them, achieving performance levels unattainable by simply reusing or averaging the base method’s prompts.
>
> For TAC-based models, as noted in the paper, the comparison is not fair, since these methods include trained components other than prompts, such as projection and head matrices, which can contain the bulk of the adaptation parameters. Since DMP only synthesizes prompts, we reuse the parameters of the original method for anything else. This obviously limits the gains of DMP, since the prompts are just a component of the adaptation. Despite this structural limitation, Table 10 shows that DMPTAC still achieves positive improvements (e.g., +0.4% average in new split for 55 dataset combination pairs) and large gains on individual pairs **(e.g., Flowers \& Pets with +3.0\% gain)** over TAC.
>
> **3) Spacing and formatting issues**
>
> We apologize for the inconvenience caused due to the formatting which was a result of the page limit. We thank the reviewer for the pointer and will adjust the formatting for improved figure/table spacing and zoomed insets (we already provide higher-resolution figures and zoomed versions in the Appendix).

---

> ### Author Response · Authors · 2025-11-27
>
> **4) Missing Comparisons**
>
> -  Please refer to the $\textbf{general comment}$ for discussion on comparison with prior works. We emphasize that all method cited by the reviewer are prompt learning methods (left side of Figure 3). They are not competitors to DMP, because they all assume access to image-label data or test-time visual inputs during optimization, whereas DMP intentionally assumes no access to image data. They are instead complementary, in the sense that DMP can be applied to prompt repositories produced by these methods. But because, for prompt learning methods, prompts are learned from a single task, they are not expected to generalize across tasks. While prompt learning methods could potentially be trained across several tasks, this has not been shown and the complexity would be large, since the training would have to be over the image datasets of all tasks. For example, training over the 11 datasets considered in this paper would require more than 100 GB VRAM in GPU memory which is not possible on a single GPU and requires multiple GPUs or nodes. The three step procedure of Figure 3 is key to avoiding this complexity. The prompts are learned per task, e.g. using one dataset at a time and any of the prompt learning methods cited by the reviewer, and the $\textit{global}$ training of the DMP model only requires the prompt repository, which is orders of magnitude smaller than the image repository of the associated tasks. This decomposes the complexity of learning the prompt distribution into a sequence of low complexity training steps.
>
> - An extensive discussion of how DMP compares to all prompt learning methods is beyond the scope of the paper. The paper addresses the problem that prompt repositories $\textit{already exist}$. The question is how they can be leveraged to learn a model that allows obtaining a prompt for $\textit{any}$ task by simply providing a natural language specification of that task, as shown on the right of Figure 3. The methods cited by the reviewer cannot be used for this, since $\textit{there is not even image data available for training.}$ From this point of view, the exact techniques used on the left of Figure 3 are irrelevant. The only question is whether DMP maintains comparable performance on those tasks, as is commonly done for the evaluation of foundation models. We already show that this is true for some methods (CoOp, MaPLe, CoPrompt, and TAC), an extensive comparison to all methods is, in our opinion, not needed or interesting. None of these methods have, for example, been shown capable of generating prompts to personalize a diffusion model for image synthesis, an objective that DMP can achieve as easily as synthesizing prompts for classification. So, the methods cited by the reviewers are the state of the art for a narrowly defined task that does not even begin to approach the generality of DMP. DMP should not be expected to beat these methods on the specific tasks where they are trained. It is not surprising that a dataset specific model will beat a generalist method, like DMP, on the specialized dataset where it was trained. For example, ImageNet trained CNNs still beat CLIP on ImageNet. This does not detract from the fact that CLIP is a much more important model than the state-of-the-art ImageNet CNNs. Rather than an extensive comparison to all these methods, we believe that it is much more important to shown the ability of DMP to generate good prompts for the completely different tasks of diffusion model personalization, controlling concept generation with prompt sliders, negative prompting, and all other applications demonstrated on Figures 4-8 of the paper and throughout the appendix.
>
> - In summary, DMP and the methods cited by the reviewer are complementary. Any of these methods could produce the prompts on which DMP is trained. We **already provide comparisons to CoPrompt [C] in Table 4 and Table 11 of the paper**, as well as several other prompt learning methods (CoOp, MaPLe, CoPrompt, and TAC). In all cases, DMP has comparable or improved performance, even though it is only trained on the prompts learned by these methods and does not even use image data. Furthermore, **DMP generalizes to other tasks (e.g prompting of image synthesis models) to which these methods cannot even be applied without modifications, and enables sampling, composition, and negative prompts without ever revisiting the underlying data again.** Finally, DMP is scalable in terms of task learning, due to the three step procedure of Figure 3, something that prompt learning methods do not even aspire to. We have added these points to the discussion to emphasize complementarity rather than competition.

---

> ### Author Response · Authors · 2025-11-27
>
> **5) What is the training of the presented method vs prior work (e.g. CoOp, TAC etc) ?**
>
> - CoOp / TAC / MaPLe / CoPrompt and all other prompt learning methods are task, model, and dataset specific,  learning  continuous prompt vectors (and in some methods additional projection matrices) by gradient descent. They require per-task training and task-specific losses and produce a set of learned prompt vectors/weights per combination of task, dataset, and downstream model.
>
> - DMP training starts from a repository of these prompts, which is used to train a diffusion model, as shown in Figure 3. Training DMP requires first collecting a prompt repository (we obtained it by running the baseline methods under their original training regimes but in practice the prompts can be produced by thousands of independent practitioners or using the DMPVariation model as discussed in Appendix A.8) and then training a 1-D diffusion U-Net (or VAE+diffusion model when prompts are high-dimensional) conditioned on a CLIP text embedding of the prompt specification, e.g. "an SD-XL prompt for the synthesis of anime characters" in Figure 3. At inference, DMP samples prompts from the learned distribution, conditioned on a natural language specification of the task, e.g. ``an SD-XL prompt for the synthesis of abstract art"  without per-task re-optimization, reducing storage and enabling prompt composition/negation. Hyperparameters and model sizes are discussed in Appendix A.10/Table 25.
>
> **6) The visual results quality looks somewhat poor, perhaps because the diffusion model used?**
>
> We used Stable Diffusion v1.5 for our experiments and so the quality of the generated images is bounded by the model. We provide additional results with SD-XL models in Figure 13 and Figure 14, where the quality is much higher. This shows that the problem identified by the reviewer is due to the diffusion model, not DMP.

---

### Official Review · Reviewer_Vpu4 · 2025-10-31

**Soundness:** 2
**Presentation:** 2
**Contribution:** 2
**Rating:** 4
**Confidence:** 3

**Summary:**

The paper presents a generative approach for prompt-learning, whereby a sampling distribution is learned and used to sample a prompt from a query. The authors rely on training a diffusion model that can decode a prompt from noise, conditioned on a particular query, and which is learned using a large pool of prompts available for different methods. Contrary to existing prompt-tuning techniques, in the proposed DMP the generative model is learned from a pool of available prompts rather than from fine-tuning the prompt embeddings to a particular task.

**Strengths:**

The idea of training a generative method for prompt generation in the text domain rather than in the continuous domain of embeddings as in existing methods is appealing from the perspective that generated prompts are semantically complete and can hint on how the models need prompting from a human perspective.

The results are competitive, showing that the proposed approach is supported by the results.

**Weaknesses:**

The paper is loosely written and lacks consistency. The authors refer to the literature in a rather unprofessional way with sentences like "we train CLIP following the respective papers", what papers? this needs to be more specific.

The authors refer to the choice of diffusion in the introduction but this choice is not justified experimentally. Why not other generative approaches? Some simpler alternatives can outperform diffusion for constrained tasks like prompt generation.

I am not sure if I understood this so please correct me in the rebuttal if I did so. The method aims at sampling prompts from a prompt distribution that is learned over a set of discrete prompts. In other words, while existing prompt learning approaches learn continuous representations for particular models, the DMP learns from the pool of prompts, without regard to how these perform on the downstream tasks. This amounts to generating combinations of prompt that are likely to generalize less than the continuous counterparts to new tasks. For example, in Bayesian Prompt Learning (a reference the authors missed in the paper: "Derakhshani et al. Bayesian prompt learning for image-language model generalization, ICCV'23") one can sample new prompts after being learned through backpropagation from the model's error in a set of specific datasets. BPL shows to generalize to new classes without the need of retraining so I believe the authors should consider comparing both theoretically and experimentally against it.


In Section 3.1 it is mentioned that the models are trained "for several prompt tuning methods such as CoOp". I might have missed that bit, but I didn't get how these methods are combined with the proposed DMP. I would like to ask the authors to explain this clearly.

**Questions:**

I have included my questions above.

---

> ### Author Response · Authors · 2025-11-26
>
> We thank the reviewer for their time and thoughtful comments on the paper. We are encouraged by the positive comments on the proposed method (appealing) and the results.
>
>
> **1) On paper writing, consistency and clarity**
>
> - We thank the reviewer for the suggestion. Some of these omissions were due to space limitations. We have added the methods in the revision. We included a more detailed discussion of all papers used (CoOp/CoCoOp/CoPrompt/MapLe/TAC) in appendix A.10, which contains implementation details for baseline prompt learners and CLIP variants, and a pointer to the appendix plus citations in the paper.  We will further revise the paper carefully to ensure clarity and consistency.
>
> **2) Choice of Diffusion Models**
>
> - We selected diffusion models for prompt generation for three tightly related reasons: (a) they model complex, multimodal continuous distributions well, (b) they afford control mechanisms (classifier-free guidance, negative prompts, conditional sampling, stochastic interpolation) that are directly useful for prompt composition, negative prompting, and sampling diversity, and (c) they can be prompted with natural language, e.g. ``an SD-XL prompt to synthesize abstract art" in Figure 3, using well understood mechanisms that are easy to implement and enable an intuitive user interface.
>
> - Note that diffusion models explicitly learn denoising paths between noise and sample, which naturally supports controlled stochastic sampling and classifier-free guidance (CFG). This has many uses, e.g. negative prompting can be implemented in a principled manner by using CFG to  push samples away from a user-provided (negative) language prompt. Autoregressive or simple VAE models do not offer an equally straightforward mechanism for CFG-style control. These advantages are discussed in Sec. 3 and A.8.
>
> - Beyond this, the choice of diffusion models is justified by better performance. Appendix section A.6.1 presents a comparison to Transformer / LLM-style autoregressive generators and standard VAE baselines. Table 11 of the paper shows that the latter lead to  weaker downstream  model generalization than the diffusion model we use.
>
> - It could be true that a simpler conditional model could be competitive for single-task prompt generation. However, as we hope that it is now clear from the $\textbf{general comment}$ above, the goal of the paper is exactly to go beyond this and learn a model that generalizes across many tasks, as shown in Figure 3. Besides enhanced controlled prompt operations (composition/negation), DMP’s main advantage is in cross-dataset and cross-task generalization, as shown by the large gains in the combination-dataset experiments of Table 8,10 and the MTA cross-task results in Table 12,17 where DMP prompts are evaluated for hierarchical classification with gains for DMPTAC are as large as +9/+10 for Imagenet-sketch/SUN397 datasets respectively. Please see the additional results under general comment section for more details on these results.

---

> ### Author Response · Authors · 2025-11-26
>
> **3) Clarification on the method**
>
> - Please refer to the $\textbf{general comment}$ and Figure 3 for a clarification of these issues. We do not understand the distinction between continuous and discrete prompts referred by the reviewer, as the "discrete prompts'' used to train DMP  are actually the "continuous" prompts learned by prompt learning. The prompt repository used to learn the DMP model is produced by prompt learning methods. Diffusion models for image synthesis are learned from datasets of "discrete" images and can still synthesize "continuous" images that they have never seen, e.g. "an astronaut riding a horse in Mars". Similarly, DMP can produce ``continuous prompts" for new tasks.
>
> - We think that there is a misunderstanding when the reviewer says ``without regard to how these perform on the downstream tasks". Note that the prompt generation is conditioned by natural language, e.g. "an SD-XL prompt to synthesize abstract art" in Figure 3. This induces an organization of the prompt space that enables $\textit{generalization.}$ It is similar to the organization of image space by a model trained for image synthesis with natural language conditioning. Like the image synthesis model learns to generate images of "a cat drawing a banana," DMP  learns to generate prompts for tasks (specified by language) on which it was not specifically trained. So, it is not true that the prompts are learned "without regard to how they perform on the downstream tasks". There is also no evidence that these prompts "are likely to generalize less than the continuous counterparts to new tasks," in the same way that the picture of "a cat drawing a banana" looks real, even if no one has ever seen a cat drawing a banana. All that DMP is doing is $\textit{interpolating}$ in prompt space from the repository of prompts on which it is trained. If the space is well learned, this interpolation should be as good as computing the **"continuous counterpart"**. This is the principle behind image foundation models which is the same for prompt diffusion as well.
>
> - We also provide a theoretical connection to downstream risk: Proposition 1 in Appendix A.1 links the diffusion denoising objective to downstream task loss (the denoising KL upper-bounds the downstream risk under mild assumptions). Thus DMP’s objective is not blind to downstream utility and the paper provides a formal guarantee tying the two. All our results show that the prompts sampled by DMP generalize well to tasks unseen at training. We will add some more explanatory text in Sec. 3 to make this linkage and the framework explicit.

---

> ### Author Response · Authors · 2025-11-26
>
> **4) Comparison with Bayesian Prompt Learning**
>
> Thank you for the pointer. We have included it in the related works. While BPL (Bayesian Prompt Learning) is relevant as method for generative modeling of prompts, there are  major differences.
>
> - BPL performs posterior inference to sample prompt hypotheses, using $\textit{task-specific}$ image-label data and a downstream loss during training. Hence, it should be placed in the prompt learning section of Figure 3. While it is true that, being a Bayesian approach, it can sample new prompts, these are just for the task, dataset, and model, on which it is trained. The best placement of BPL in Figure 3 would be as an $\textit{enhanced prompt learning method}$ that allows sampling a set of prompts in the neighborhood of the prompt produced by standard prompt learning methods for a given model and dataset. But, because the prompts are learned from a single task, they would not generalize beyond a neighborhood.
>
> - Having said this, it could be possible to train BPL over many tasks and models. This, however, has not been shown in that paper. We also believe that it is not practical, since the training would have to be over the image datasets of all tasks and the complexity would be very large. For example, training over the 11 datasets considered in this paper would require more than 100 GB VRAM in GPU memory which is not possible on current GPUs and requires multiple GPUs or nodes. The three step procedure of Figure 3 is key to avoiding this complexity. The prompts are learned per task, e.g. using one dataset at a time, and the $\textit{global}$ training of the DMP only requires the prompt repository, which is orders of magnitude smaller than the image repository of the associated tasks. This decomposes the complexity of learning the prompt distribution into a sequence of low complexity training steps.
>
> - In any case, to investigate how DMP compares to BPL, in terms of ability to generalize across tasks, we performed an experiment using one dataset pair, Oxford Flowers and Eurosat. As a DMP representative, we used the DMPCoOp model trained over the 11 datasets. For BPL, we compared two approaches. In the first, denoted BPL (ZS), BPL was applied to each dataset separately, resulting in two prompt sampling methods specialized to each dataset. We then used the prompts sampled from each model on the two datasets and picked the model that performed best. This is the single-task BPL model that generalizes best to the other task. Finally, we trained BPL over the two tasks, by combining all classes, and measured the performance of the prompts sampled from the resulting model, which is denoted as BPL (Trained). Table 7 compares results of all approaches for the task of classification over all classes from both datasets. As expected, the table shows that the prompts produced by BPL (Trained) are superior to those of BPL (ZS). Note that for BPL (ZS) even the Base classes pose a generalization problem, since the model was trained on only one dataset. In result, the performance is weak. The much more computationally intensive joint training over the two datasets produces significantly better results. Even this, however, which is overfitted to the combination of the two datasets, has weaker performance than DMP, which simply learns from the prompts produced by the CoOp method $\textit{individually}$ for the 11 datasets. Even though DMP was never trained specifically on the combined EuroSAT+Flowers dataset, it generalizes to unseen classes of this dataset with $\textbf{+2.7\%}$ better performance than BPL (Trained), which is trained on the combined dataset.
> We note that BPL is computationally expensive, requiring approximately 10 hours to train on a 40GB GPU (memory requirement is 40GB) even for a single dataset pair, which limits its scalability compared to DMP.
>
>
> - Finally, we would like to emphasize that the paper addresses the problem that prompt repositories \textit{already exist}. The core question is how to leverage them to train a model that can produce a prompt for any task directly from its natural-language description, as illustrated on the right side of Figure 3. BPL cannot serve this purpose because no image data is available for training in our setup. From this point of view, the exact techniques used on the left of Figure 3 are not important. The only question is whether DMP maintains comparable performance on those tasks, as is commonly done for the evaluation of foundation models. There should not be a requirement that DMP beats BPL on a specific task where the latter was trained, because DMP can do so many other tasks that BPL cannot (at least without task-by-task retraining). It is entirely expected that a dataset-specialized model like BPL will outperform a generalist model like DMP on its own training domain. For instance, ImageNet-trained CNNs can still surpass CLIP on ImageNet, but this does not diminish CLIP’s importance.

---

> > ### Author Response · Authors · 2025-11-27
> >
> > **5) In Section 3.1 it is mentioned that the models are trained "for several prompt tuning methods such as CoOp". I might have missed that bit, but I didn't get how these methods are combined with the proposed DMP. I would like to ask the authors to explain this clearly.**
> >
> > - The  procedure is summarized in Figure 3 and discussed in the $\textbf{general comment}$. The following are the detailed steps.
> >
> >   - For each prompt learning method (CoOp, TAC, MaPLe, CoPrompt, TI), downstream model, and dataset, learn the prompts on the dataset training splits, following the published code or hyperparameters and save the learned prompts (we use 40 random seeds / checkpoints per dataset to build diversity);
> >
> >   - Aggregate \textbf{all} prompts into the prompt repository tensor and pair each prompt with the CLIP text-embedding of the associated natural language description, e.g. ``SD-XL prompt for anime" in Figure~\ref{fig:DMPteaserReb};
> >
> >   - Use the prompt repository to train a diffusion model to sample prompts conditioned on the text embedding (classifier-free guidance used during training/inference);
> >
> >   - At inference, sample prompts from DMP conditioned on a text description and use the sampled prompt vectors in the frozen downstream model.
> >
> > We will update the text in Sec. 3.1 with the above detailed description and have already added Figure 1, Table 1 for clarifications (the appendix already contains the full algorithmic pseudocode).

---

### Official Review · Reviewer_zqwL · 2025-11-02

**Soundness:** 3
**Presentation:** 3
**Contribution:** 3
**Rating:** 6
**Confidence:** 4

**Summary:**

This paper focues on the prompt generation and proposes the Diffusion Meta-Prompt (DMP) to train a conditional generator via diffusion models to directly output the prompt embeddings from the description input. One of the core ideas of DMP is to view the prompts of specific tasks as a distribution in the embedding space, and employ a diffusion model to learn such prompt distribution. To do this, DMP first collects the (description, prompt) pairs from the base prompt tuning methods, and then train a conditional generator by the denosing loss in diffusion. The resulting generator can directly output the prompt given a testing description. Results on image clasification and editing tasks show the improvements of DMP over the base models.

**Strengths:**

1), The idea of directly training a diffusion model to generate prompts is novel to the prompt-tuning community, and the results of the proposed model clearly demonstrate the effectiveness of this motivation.

2), DMP is well-motivated by integrating prompt tuning with generative modeling. By viewing prompts as distributions in the embedding space, the learned DMP exhibits strong generalizability and transferability to unseen tasks.

3), The authors conducted extensive experiments to evaluate the proposed model, and all the results consistently support its underlying motivation.

**Weaknesses:**

1), One of the main concerns lies in the relatively small improvements shown in Table 4. Since DMP is trained based on the outputs of the base model, this dependency may limit its overall performance.

2), The authors provide various visualization results to demonstrate the effectiveness of DMP. However, it would be helpful to also include quantitative results to substantiate the improvements in image generation quality.

3), What are the main technical differences between DMP and Neural Network Diffusions [1,2,3]? They appear to share a similar idea of generating parameters through diffusion models.

4), DMP can be viewed as an enhancement stage that refines the outputs of a base model using diffusion processes. Recent studies have leveraged Large Language Models (LLMs) to refine textual prompts in word space [4,5]. The authors are encouraged to discuss in detail how DMP differs from such approaches. Including additional comparative results would also strengthen the paper.

[1] Neural network diffusion

[2] Conditional lora parameter generation.

[3] Difflora: Generating personalized low-rank adaptation weights with diffusion

[4] AWT: Transferring Vision-Language Models via Augmentation, Weighting, and Transportation

[5] PRewrite: Prompt Rewriting with Reinforcement Learning.

**Questions:**

Please see the Weakness above.

---

> ### Author Response · Authors · 2025-11-26
>
> We thank the reviewer for their time and insightful comments on the paper. We appreciate the positive comments on novelty, motivation, strong generalization of our method and extensive experiments in the paper.
>
> **1) Concern on the relatively small improvements shown in Table 4 and dependency on the outputs of the base model**
>
> - In an attempt to provide fair comparisons, we ended up inducing the reviewers to miss the significant properties of DMP. Please see the $\textbf{general comment above}$. Table 4 of the paper reports results on single tasks and single datasets. This is the ``turf" of prompt learning methods, where prompts are $\textit{task and dataset specific}$. These prompts are overfitted to these classes and datasets, which makes them difficult to beat in these classes and datasets. On the other hand, DMP is trained to generate prompts for $\textit{any}$ task and $\textit{any}$ dataset.  Nevertheless, Table 4 of the paper shows that DMP prompts perform comparably even on the tasks/datasets used to train the prompt learning prompts. In fact, DMP achieves small gains for unseen classes of these datasets. This is an important $\textit{check}$ that, even though it is learned from overfitted prompts, $\textit{DMP is less overfitted to the training classes}$. But the gain magnitudes are not that important, because these classes are still very close those where the overfitted prompts are trained.
>
> - The benefits of DMP are much easier to see on evaluations that require cross-dataset and cross-task generalization. Please refer to the **Additional Results comment** under general comment above for the detailed discussion of the generalization results of our method. These results collectively show that DMP is not merely “dependent” on the base prompts and it actively generalizes beyond them, achieving performance levels unattainable by simply reusing or averaging the base method’s prompts.
>
> **2) On the quantitative results for image generation**
>
> -  We note that several such results are already provided in the paper.  Tables 2 and 5 demonstrate measurable improvements over the baselines for identity and slider-based attribute control, Table 6 analyzes the generalization of DMP-generated prompts, showing a **+1.4-point** gain on CLIP prompt alignment score over baseline Textual Inversion (TI). Face identity variation tasks also include quantitative metrics showing that DMPVariation produces more consistent identities and better prompt interpolation quality than TI alone. Additional results are presented in Appendix A.7 (Table 22 and Table 23).
>
> - We now also **quantify the generalization observed in Figure 20 of the paper, using the HPSv2 metric, in Table 6**. We collected 100 random concepts downloaded from huggingface repositories to test the generalization of DMP prompts against the baseline TI with the six different prompt templates shown in Figure 8 and Figure 20 of the paper. Table 6 shows that DMP generated prompts have superior generalization, which is consistent with the qualitative results shown in the paper. However, as is common for generative modeling, the difference in HPSv2 does not fully capture the large difference between TI and DMP, which can be verified in Figure 20 of the paper. While TI produces images that $\textit{simply do not comply}$ with the edit instruction, all **images synthesized with DMP prompts comply with those instructions.**

---

> ### Author Response · Authors · 2025-11-26
>
> **3) technical differences between DMP and Neural Network Diffusions**
>
> - Please refer to General Comment on the novelty of DMP over prior prompt learning approaches. Regarding prior neural-network diffusion (NND) methods, while both these and DMP use diffusion models to synthesize parameters, they differ fundamentally in what is modeled, what training data is required, and how the generated outputs are used. Prior works, such as NND, conditional LoRA diffusion, and Difflora, diffuse neural network weights or LoRA matrices, which are extremely high-dimensional (millions of parameters), require supervision from image-label data or task losses, and synthesize functional model parameters for direct network adaptation. In contrast, DMP diffuses low-dimensional prompt embeddings, i.e. low-dimensional vectors learned by existing prompting methods, and critically, $\textit{does not require images, labels, or gradients from the downstream task.}$ Only the learned prompts themselves are needed. Hence, DMP has vastly lower-complexity, which allows supporting wide arrays of tasks.
>
> - More generally, training models of prompt distributions is orders of magnitude more scalable than training models of distributions of model weights. This allows DMP to successfully use a single model to capture the distribution of prompts from many tasks, unlike prior methods, which have not been shown to work for anything other than single tasks. The prompt space also has significant amounts of structure, which do not necessarily hold for weight spaces. As we show in the paper, this allows prompt sampling, cross-dataset prompt generalization, prompt composition, and negative prompting, which have not been demonstrated for NNDs and we do not believe to be possible. The diversity of tasks and downstream models that we discuss (and present results for) in the paper has never been demonstrated for NND. We believe that it is  simply not possible to achieve with NND, at least with the computational resources we have available. **In summary, the dimensionality of the model weight space creates a barrier to even exploring the idea of learning the distribution of model weights across a large set of tasks, which is the main goal of this work. Furthermore, the connection between prompts and important mechanisms of diffusion models, like classifier free guidance, allow functionalities, e.g. negative prompting, that are not available for weight-based adaptation. We show that DMP inherits these functionalities.**
>
> **4) On relation to prior works on Large Language Models (LLMs) to refine textual prompts in word space**
>
> - LLM-based prompt refinement methods (such as AWT and PRewrite) require data from the downstream task (not just prompts) and operate in language space, rewriting text prompts using linguistic heuristics or RL. They require LLM inference, additional linguistic rules, and often task labels or performance rewards. They are usually applied to a single task, similarly to prompt tuning methods used to produce the prompts that are used to learn DMP. They do not operate on learned embeddings and they cannot sample a distribution of prompt vectors or exploit diffusion-specific capabilities such as classifier-free guidance or negative sampling.
>
> - On the other hand, DMP diffuses embedding-space prompts, not natural language. It learns a continuous distribution of prompts over multiple tasks, enabling sampling diverse prompts, composing prompts from multiple conditions,
> generating negative prompts, and cross-dataset / cross-task generalization. It does not use data from the downstream tasks or datasets, just prompts. The ``LLM prompt refinement equivalent to DMP" would be an LLM trained over a repository of refined language prompts to produce produce LLM refined prompts for new tasks, without any additional language data from those tasks.
>
> - We have added Table 9 in the Appendix clarifying the differences and will add a discussion in the related works clarifying this distinction.

---

### Comment · Area_Chair_AG2p · 2025-11-20
**To AI Review**

Dear authors,

I would like to share an important reminder regarding the review process. Recently, we have noticed that some reviewers may be using AI tools to help generate their reviews. This can lead to low-quality or inaccurate feedback, which is unfair to authors who deserve careful and thoughtful evaluations.

To help maintain fairness, I kindly ask for your assistance: If you believe a review you received was partly or fully generated by AI, and you have some evidence (for example: unusual writing style, clear factual mistakes, AI-detector results, repeated generic sentences, etc.), please feel free to contact me directly.

I will review any evidence you provide and, if appropriate, adjust the weight of the reviewer’s evaluation so that it does not negatively affect your submission. Thank you for helping us keep the review process fair and responsible. Your understanding and cooperation are greatly appreciated.

Best regards,

AC

---

### Author Response · Authors · 2025-11-26
**General Comment**

We thank all the reviewers for their thoughtful and useful feedback to improve the paper. We are encouraged that reviewers found our method novel and effective (zqwL, Vpu4), theoretically sound (jZN5), versatile with extensive evaluation and good improvements (jZN5, fRKq).

We have uploaded a revised manuscript with the changes highlighted in blue for the convenience of the reviewers.

**Relation to prior works**
- Because DMP can subsume many tasks, it is difficult to always keep in mind the difference between DMP and previous prompting works. We feel that this generated some confusion in the reviewer comments. We now include Figure 3 in the revision to make it easier to always keep in mind the differences between approaches. The main point is that classical prompt learning techniques are $\textit{task and model specific}$, and require $\textit{task specific data and losses.}$ DMP instead learns the distribution of prompts from a prompt repository (produced by these prompt learning techniques). It can then be used to sample prompts for $\textit{many tasks}$. Note that **DMP is not task specific and does not require task specific losses or data, just a prompt repository.** When the repository contains only one or few prompts per concept, we can generate the diverse variations of it for training the DMP using the DMPVariation model (Sec 3.2). Appendix A.8 illustrates this where DMPVariation model produces prompts with sufficient intra-concept variability for effective DMP training (Table 22, 23), even without additional data or task-specific optimization. This makes any method that requires task specific data/losses $\textit{non-comparable to DMP}$. While we present comparisons to such methods, these are just to check that DMP $\textit{retains competitive performance}$ despite operating under a far more challenging and general setting. We show that DMP’s sampled prompts often outperform the prompt-learning prompts it was trained on. For example, the classification results of Table 4 in the paper show that DMP outperforms the baseline on the unseen classes of the very same dataset, despite never accessing a single image. We show that DMP generalizes better than these methods for novel tasks (Figure 5), cross-domain tasks (Table 3, Figure 4) and in variations of the specific tasks (Table 7) on which they are trained.

- We emphasize, however, that while satisfying, this is $\textit{not the important point.}$  Showing that DMP outperforms a baseline prompt learning method on a specific classification task is equivalent to showing that a foundation model can do better on ImageNet than an ImageNet trained CNN. This is not the reason why foundation models are important. The importance is that they can generalize to a lot of tasks. Similarly, DMP can produce prompts for many tasks, including tasks as diverse as image generation, classification, etc. In fact, if trained from a large scale repository, DMP becomes a foundation model for prompts. Comparing the  results of such a model to the results of a task specific approach, like prompt learning, is simply a $\textit{check}$. The DMP model should not do much worse than the task specific models on average. But the value of the DMP model is that it can do all tasks. This makes it clearly more useful than the task specific methods. Hence, focusing narrowly on whether DMP matches a task-specific method on its own training dataset misses the fundamental point that
task-specific prompts cannot be used outside their domain, while DMP can be used everywhere. The only requirement should be that the prompts sampled by DMP induce downstream model performance $\textit{comparable}$ to that achieved with prompt learning, which our experiments consistently confirm.

- The reviewers asked for comparisons to many methods, which are listed in Table 9. To see how uncomparable these methods are to DMP, it suffices to see that most (if not all) of them are designed for classification datasets and losses. They cannot even be used to produce prompts for diffusion models, for example, without significant changes. This is unlike DMP, which is totally agnostic to what task the prompts are designed for, or the losses and datasets used to produce them. We refer to this as being ``task data/loss free". This is just one dimension along which different methods can be categorized and differ from DMP. Several others are listed, e.g.  whether the methods can produce prompts for multiple tasks in a zero-shot manner, as shown in the right of Figure 3, if the methods can learn over several tasks or only support learning prompts one-task-at-at-time, and whether the prompts have to be stored to perform inference with the downstream model.
It is clear from the table that DMP differs from most approaches in several of these dimensions and that comparisons between the methods are thus inherently not fair. While DMP is a foundation model for prompts, all these methods train prompts for specific tasks.

---

> ### Author Response · Authors · 2025-11-26
> **Additional Results**
>
> - The benefits of DMP are much easier to see on evaluations that require cross-dataset and cross-task generalization. In the paper, we tried to make these points using 1) Table 7, which shows that DMP **consistently improves across all, new and base classes for combined classification**, and 2) Figure 5, which shows that DMPCoOp obtains **significant gains on hierarchical classification** on ImageNet OOD variants **(e.g., ImageNet-A: +11.7, SUN397: +11.4)**. We now recognize that we should have emphasized this discussion more. To address this, we now present a more extensive set of experiments that require generalization across datasets and explain the procedure in detail as follows. First, we used prompt learning to train prompts over the 11 datasets ${\cal D}_i, i \in \{1, \ldots, 11\}$ considered in the paper (see L346). In all cases, we used the Base/New decomposition, where some classes (New) were left out of training to test generalization. We created a prompt repository with the prompts ${\cal P}_i, i \in \{1, \ldots, 11\}$ learned from the 11 datasets and used it to train the DMP model. We then created all 55 datasets ${\cal T}_i, i \in \{1, \ldots, 55\}$ containing pairs from these 11 datasets, to test generalization performance. For each dataset ${\cal T}_i$, we performed classification with
>   - prompt sampled from DMP
>   - the two prompts ${\cal P}_i$ associated with the two datasets ${\cal D}_i$ in the pair.
>
> - We then measured the performance of the two prompts ${\cal P}_i$ on the test classes of ${\cal T}_i$ and chose the best for each dataset. We also tested the average of the two prompts ${\cal P}_i$ but it did not perform better. Note that this is a best-case scenario for prompt learning, which would not be feasible in practice, where test labels are not available, making it impossible to know which prompt does best. This was compared to the performance of the DMP prompt, with the results shown in Table 8.
>
> - The table shows the average accuracy of baseline CoOp prompts and DMPCoOp prompts across all 55 dataset combination pairs. The last column reports the top-3 highest individual gains observed in specific combinations mentioned which are significantly higher than the CoOp prompts.
>
>
> - In particular, DMPCoOp achieves **consistent and often large gains** as noted below:
>
>   - **All split: +4.3\% average improvement, with top combination pairs showing very large gains (e.g., EuroSAT \& OxfordFlowers: 35.5 to 61.2, +25.7)**.
>
>   - **Base split: +4.7\% average improvement (e.g., ImageNet \& Flowers: 32.6 to 70.0, +37.4).**
>
>   - **New split: +3.3\% average improvement (e.g., Pets \& SUN397: 47.6 to 69.7, +22.1).**
>
> - These results demonstrate that DMP generalizes across tasks and datasets in ways that the base prompt learner cannot, despite being trained only on its prompt embeddings.

---

### Comment · Area_Chair_AG2p · 2025-11-27

Dear Reviewers and Authors,

As we are approaching the rebuttal deadline, I would like to share a gentle reminder with everyone.

For authors:
If you have not yet submitted your rebuttal, please make sure to do so as soon as possible. Submitting very close to the deadline may reduce the chance for reviewers to read and respond in time, which could affect the discussion phase.

For reviewers:
If a rebuttal has already been submitted for your assigned paper, I encourage you to take a moment to read it and, where appropriate, provide a brief response or update your evaluation. Of course, this is not meant to pressure anyone into changing scores, it is simply to ensure that all reviews remain well-informed before final decisions.

Thank you all for your time and effort in keeping the review process smooth and constructive.

Warm regards,
AC

---

### Meta-Review · Area_Chair_MpAD · 2025-12-16

**Summary:**

Summary of each reviewers' core comments and how the authors responded to them:

1. Reviewer zqwL (score 6, certainty 4): The reviewer asks questions on (a) a small improvement in Table 4, (b) technical differences from other existing papers, and (c) the difference from the recent LLM-based methods. In a way, questions (b) and (c) relate to the novelty aspect of the paper. Although the reviewer explicitly wrote that the approach is novel, questions like (b) and (c) made me wonder if the reviewer truly believes the approach is novel. The authors responded to all questions. Some of the responses are useful and helpful for judging the difference between their proposed method and others.

2. Reviewer  Vpu4 (score 4, certainty 3): The reviewer questions (a) presentation issues and (b) the differences from the way Bayesian prompt learning performs. While the authors explained the key differences between their method and BPL, I also agree with the reviewer that there is a key distinction between "discrete" and "continuous" prompt learning approaches, which should be more clearly contrasted.

3. Reviewer  jZN5 (score 4, certainty 4): The reviewer questions (a) the novelty of the work and (b) the missing comparisons. The authors provided additional experimental results to rebut.

4. Reviewer fRKq (score 4, certainty 3): The reviewer questions (a) the novelty of the work and (b) the missing ablation studies. The authors provided additional ablation study results to rebut.


Overall, I find the novelty of the proposed method somewhat questionable. Three reviewers questioned how this proposed method compares with existing work in terms of technical detail. I have absolutely no idea whether the reviewers would find the authors' responses convincing and increase the score, but I don't feel the described differences make this paper particularly novel. Hence, I recommend rejecting the paper.

**Reviewer Concerns:**

See the summary above.

**Reviewer Scores:**

Three reviewers questioned how this proposed method compares with existing work in terms of technical detail. I have absolutely no idea whether the reviewers would find the authors' responses convincing and increase the score, but I don't feel the described differences make this paper particularly novel.

---

### Decision · Program_Chairs · 2026-01-26

Reject